# Imaging the flat bands of magic-angle graphene reshaped by interactions

J. Xiao[1,3], A. Inbar[1,3], J. Birkbeck[1,3], N. Gershon[1], Y. Zamir[1], Y. Vituri[1], T. Taniguchi[2], K. Watanabe[2], E. Berg[1] & S. Ilani[1✉]

Electron interactions in quantum materials fundamentally shape their energy bands and, with them, the material's most intriguing quantum phases. Magic-angle twisted bilayer graphene (MATBG)[1–3] has emerged as a model system in which flat bands lead to a variety of such phases, yet the precise nature of these bands has remained elusive owing to the lack of high-resolution momentum-space probes. Here we use the quantum twisting microscope (QTM) to directly image the interacting energy bands of MATBG with unprecedented momentum and energy resolution. Away from the magic angle, the observed bands closely follow the single-particle theory. At the magic angle, however, we observe bands that are completely transformed by interactions, exhibiting light and heavy electronic character at different parts of momentum space. On doping, the interplay between these light and heavy components leads to a variety of notable phenomena, including interaction-induced bandwidth renormalization, Mott-like cascades of the heavy particles and Dirac revivals of the light particles. We also uncover a persistent low-energy excitation tied to the heavy sector, suggesting a new unaccounted degree of freedom. These results resolve the long-standing puzzle in MATBG—the dual nature of its electrons—by showing that it originates from electrons at different momenta within the same topological heavy-fermion-like flat bands. More broadly, our results establish the QTM as a powerful tool for high-resolution spectroscopic studies of quantum materials previously inaccessible to conventional techniques.

Since the discovery of MATBG[1–3] a central challenge has been to determine whether its quantum states are best understood through a local framework or a topological one. On one hand, its correlated insulating states[2,4–8] observed at integer moiré fillings resemble Mott insulators, which are naturally described by localized electrons. On the other hand, its competing Chern insulating states[9–16] can only be explained through topological arguments. Perhaps the most puzzling observation, however, is that electrons in MATBG seem to exhibit contradictory behaviours even within the same electronic state: compressibility[17] and spectroscopic[18] measurements have revealed cascades of Dirac-like compressible states, characteristic of itinerant electrons. Yet, entropy measurements[19,20] have uncovered a giant magnetic entropy that could only be explained if nearly all electrons behave as local moments. This noteworthy duality[19]—simultaneous signatures of both localized and extended behaviour—remains one of the most fundamental unresolved puzzles in the study of MATBG.

A range of theoretical approaches, including Hartree[21,22], Hartree–Fock[17,23–26], quantum Monte Carlo[27,28], exact diagonalization[29,30], density matrix renormalization group[31–33] and dynamical mean-field theory[34,35], showed that interactions can profoundly affect the flat bands predicted by the Bistritzer–MacDonald (BM) model[1]. A key insight, solidified with the development of the topological heavy fermion (THF)[36–38] and Mott semimetal[39] frameworks, is that the topological nature of MATBG's flat bands may lead to its electronic duality: although the electrons largely behave as heavy, localized 'f-electrons', topology mandates the existence of momentum-space regions in which they act instead as light, delocalized 'c-electrons'. These two distinct electronic characters are thus predicted to coexist within the same flat bands, occupying different regions of momentum space. Although nano angle-resolved photoemission spectroscopy experiments have provided rough evidence for the presence of flat bands[40–44], the absence of a technique capable of resolving their detailed momentum-space structure has, so far, left these fundamental puzzles unanswered.

In this work, we present high-resolution momentum-space images of the interacting energy bands of MATBG, obtained using quantum twisting microscopy at $T = 4$ K (ref. 45). Away from the magic angle, the observed bands closely follow the BM prediction, showing Dirac points at the mini-Brillouin zone (BZ) corners with renormalized Fermi velocity. At the magic angle, however, the bands are substantially reshaped by interactions: across most of momentum space, the bands become extremely flat and largely gapped, hallmarks of heavy f-electron-like behaviour. By contrast, near the Γ point of the mini-BZ, the bands are dispersive and gapless, consistent with the presence of c-like light electrons. The evolution of the band structure with carrier density reveals Mott-like cascades of the heavy electrons and Dirac-like compressibility dominated by the light electrons. We directly observe

[1]Department of Condensed Matter Physics, Weizmann Institute of Science, Rehovot, Israel. [2]National Institute for Materials Science, Tsukuba, Japan. [3]These authors contributed equally: J. Xiao, A. Inbar, J. Birkbeck. ✉e-mail: shahal.ilani@weizmann.ac.il

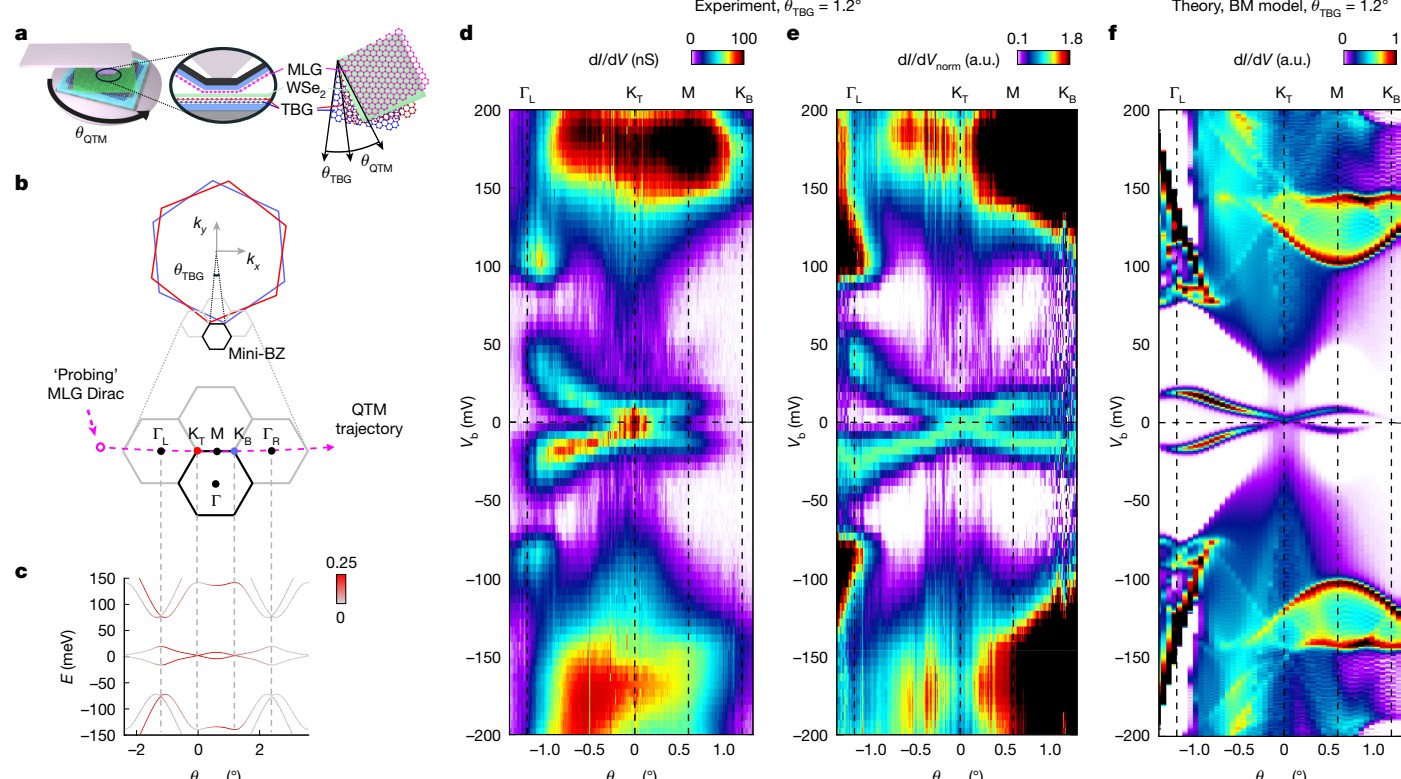

Experiment, $\theta_{TBG}$ = 1.2°                    Theory, BM model, $\theta_{TBG}$ = 1.2°

**Fig. 1 | Measurement set-up and momentum-resolved spectroscopy of TBG with $\theta_{TBG}$ = 1.2°. a**, Schematic of the QTM set-up (left) and cross-section (centre), featuring MLG (purple) on the QTM tip, separated by a bilayer WSe$_2$ tunnelling barrier (green) from back-gated TBG (top and bottom layers in red and blue). We measure differential tunnelling conductance, d$I$/d$V$, between MLG and TBG as a function of bias voltage, $V_b$. The top view (right) defines the TBG twist angle, $\theta_{TBG}$, and the angle between MLG and the top TBG layer, $\theta_{QTM}$. **b**, Momentum-space diagram showing BZs of the TBG's top (red) and bottom (blue) layers, the resulting mini-BZ (black) and neighbouring mini-BZs (grey). Inset, when the tip is rotated with respect to the sample, the MLG Fermi surface (purple circle) traces an arc in momentum space—the 'QTM trajectory' (dashed purple line)—intersecting K$_T$ and K$_B$ and closely approaching the $\Gamma$ points in adjacent mini-BZs. **c**, Calculated single-particle band structure using the non-interacting BM model[1] along the QTM trajectory for $\theta_{TBG}$ = 1.2°. The flat

bands feature Dirac points at K$_T$ and K$_B$ with renormalized Fermi velocity, reaching maximum bandwidth at the $\Gamma_L$ points, at which they are minimally separated from remote bands by energy gaps. The colour scale indicates wavefunction weight on the top TBG layer. **d**, Measured d$I$/d$V$ versus $\theta_{QTM}$ and $V_b$ for $\theta_{TBG}$ = 1.2° (Methods section 'Determine the larger twist angle of $\theta_{TBG}$ = 1.2°'). The top axis marks key momenta along the QTM trajectory: $\Gamma$, K$_T$, M and K$_B$. **e**, Same as **d**, plotted as normalized differential conductance, d$I$/d$V_{norm}$, in which, for each $\theta_{QTM}$, the valence flat-band peak intensity is normalized to unity. **f**, Calculated momentum-resolved tunnelling conductance using the BM model[1], incorporating the MLG–TBG junction electrostatics and MLG probe Dirac dispersion (Methods sections 'Electrostatics of the QTM junction' and 'The tunnelling matrix elements'). For this larger-than-magic-angle TBG, the single-particle model captures rather well the measured data. a.u., arbitrary units.

how the reshuffling of charge between these light and heavy states underlies the previously reported Dirac revivals[17,18]. While resolving the long-standing dual-nature puzzle, our measurement also uncovers a new approximately 15 meV excitation, not captured by present models, hinting at an important, yet unaccounted for, degree of freedom in MATBG.

## Experimental set-up

Our experiment examines momentum-resolved tunnelling across a two-dimensional junction formed between monolayer graphene (MLG) on the QTM tip (Fig. 1a, purple) and twisted bilayer graphene (TBG) on a flat substrate (Fig. 1a, red/blue). The TBG device includes a metallic back gate and is capped with a bilayer WSe$_2$ tunnelling barrier (Fig. 1a, green). The experiment has two relevant angles: $\theta_{TBG}$, the twist angle of the TBG, and $\theta_{QTM}$, the angle between the MLG on the tip and the top layer of the TBG. The QTM enables continuous tuning of $\theta_{QTM}$ during the experiment with millidegree precision. Although $\theta_{TBG}$ is fixed at any location in the sample, it naturally varies from one location to another as a result of inherent twist-angle disorder, allowing us to explore a range of $\theta_{TBG}$ values by performing experiments at different spatial locations.

In momentum space (Fig. 1b), the energy bands of TBG reside within a mini-BZ (black hexagon) whose corners, K$_T$ and K$_B$ (red and blue), correspond to the K points of its top and bottom graphene layers. By rotating the QTM, we rotate the Dirac point of the MLG probe (purple circle) along an arc in momentum space (dashed purple line) that passes through the K$_T$ and K$_B$ points of the TBG and closely approaches the two $\Gamma$ points in adjacent BZs, enabling us to map the TBG energy bands along the $\Gamma$–K$_T$–M–K$_B$–$\Gamma$ trajectory ('QTM trajectory').

Figure 1c presents the canonical single-particle BM energy bands of TBG along the QTM trajectory for a twist angle slightly larger than the magic angle ($\theta_{TBG}$ = 1.2°). Theory predicts that the two low-energy flat bands will exhibit symmetry-protected Dirac points at K$_T$ and K$_B$ and reach the maximal width at the $\Gamma$ points, at which energy gaps separate them from the higher-energy remote bands. The bands are coloured by the weight in momentum space projected on the top TBG layer, to which the electrons tunnel.

## The energy bands at $\theta_{TBG}$ = 1.2°

We begin with QTM measurements of TBG with $\theta_{TBG}$ = 1.2°. Figure 1d shows the tunnelling d$I$/d$V$ measured as a function of $\theta_{QTM}$ and bias voltage, $V_b$, at moiré filling $\nu \approx 0$. Recalling[45,46] that $\theta_{QTM}$ and $V_b$ translate

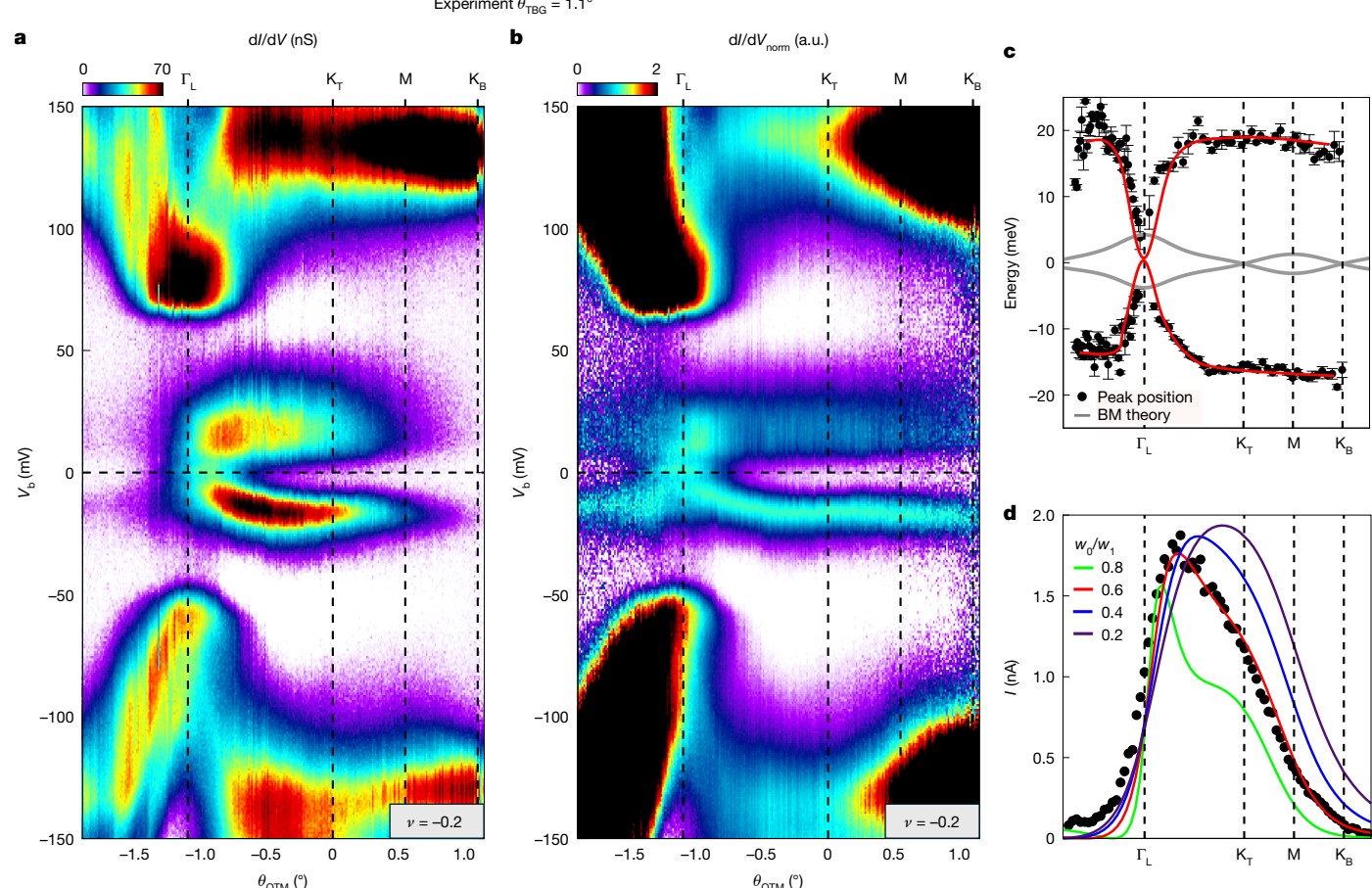

**Fig. 2 | Momentum-resolved spectroscopy of MATBG. a**, Measured d$I$/d$V$ as a function of $\theta_{QTM}$ and $V_b$ at a filling factor of $v = -0.2$ for TBG with $\theta_{TBG} = 1.1°$. Key momenta ($\Gamma_L$, $K_T$, M and $K_B$) are marked on the top axis. **b**, Same data as in panel **a** but for each $\theta_{QTM}$, we normalize the peak intensity of the valence flat band to unity, defining the quantity d$I$/d$V_{norm}$. **c**, Extracted flat-band dispersion from panel **a** (dots), in which we trace the centre of the flat-band peaks versus $\theta_{QTM}$, with a guide to the eye (red). Error bar is obtained by the standard deviation of the peak centre within a 0.05° range. Grey lines plot the single-particle BM bands for $\theta_{TBG} = 1.1°$. **d**, The intensity of the conduction flat band, given by

integrating the area under its peak $I_{peak} = \int dI/dV \cdot dV_b$ in panel **a**, plotted as a function of $\theta_{QTM}$ (dots). We compare this with the theoretically predicted momentum dependence of the intensity, given by $A(k, \omega) \cdot P_t(k) \cdot P_s(k)$, in which $A(k, \omega)$ is the spectral function, $P_t(k)$ is the wavefunction projection on the top TBG layer and $P_s(k) = \langle \sigma_x + I \rangle_k$ is the sublattice projection. The calculation uses the wavefunctions of the BM model and the different lines correspond to different $w_0/w_1$ ratios in this model (Methods section 'The tunnelling matrix elements'). We observe an excellent fit for $w_0/w_1 = 0.6$. a.u., arbitrary units.

to momentum and energy through $k = \theta_{QTM}K_D$ ($K_D$ is the Dirac momentum) and $E = eV_b$, this measurement directly maps the energy bands of TBG along the QTM trajectory (the corresponding k-points are indicated on the top axis). The positions of the peaks in d$I$/d$V$ along the $V_b$ axis reveal the energy of the electronic bands, whereas the amplitudes of the peaks provide insight into the wavefunctions. To better trace the energies of the flat bands, for each $\theta_{QTM}$, we normalize the valence flat-band peak to unity and plot the resulting d$I$/d$V_{norm}$ in Fig. 1e. The very high resolution in the measurement ($\delta k \approx 0.1° \times K_D$ and $\delta E \leq 10$ meV; Methods section 'Energy and momentum resolution') allows us to visualize these bands with unprecedented details. Specifically, we observe a Dirac dispersion with a crossing at $K_T$ but with a Fermi velocity that is seven times smaller than that in MLG. At $\Gamma$, the flat bands attain their maximum width of 70 meV, separated by gaps of 50 meV and 40 meV from the remote conduction and valence bands, respectively. Moreover, we can also identify van Hove singularities in the remote bands at energies of $E = 174$ and $-171$ meV.

Figure 1f presents a theoretical calculation of the momentum-resolved d$I$/d$V$, using the non-interacting BM bands at $\theta_{TBG} = 1.2°$, considering the tunnelling matrix elements (Methods section 'The tunnelling matrix elements') as well as the electrostatics of the MLG–TBG junction and the Dirac dispersion of the MLG probe (Methods section 'Electrostatics

of the QTM junction'). Visibly, for this larger-than-magic-angle TBG, the single-particle model captures well both the band energies and the momentum-dependent tunnelling amplitudes.

## The energy bands at the magic angle

We now turn to the central results of our paper—the energy bands of MATBG. The measurement is performed in a region of the sample in which atomic force microscopy (AFM) measurements taken at room temperature observed a regular moiré pattern with a periodicity of approximately 13 nm (Methods section 'Fabrication and characterization of the QTM tip and van der Waals device'), corresponding to $\theta_{TBG} = 1.1°$. Figure 2a shows d$I$/d$V$ measured with the QTM as a function of $\theta_{QTM}$ and $V_b$, at a carrier density close to charge neutrality ($v = -0.2$). Figure 2b presents the normalized d$I$/d$V$ (d$I$/d$V_{norm}$), following the normalization described earlier.

The bands observed at $\theta_{TBG} = 1.1°$ are very different from those at $\theta_{TBG} = 1.2°$. They do not show a dispersing Dirac point at $K_T$ but instead feature two extremely flat bands, separated by a large energy gap of approximately 35 meV. Notably, the gap appears at almost all momenta along the measured trajectory, except near the $\Gamma$ point, at which the bands are gapless. At this specific filling, the conduction band seems

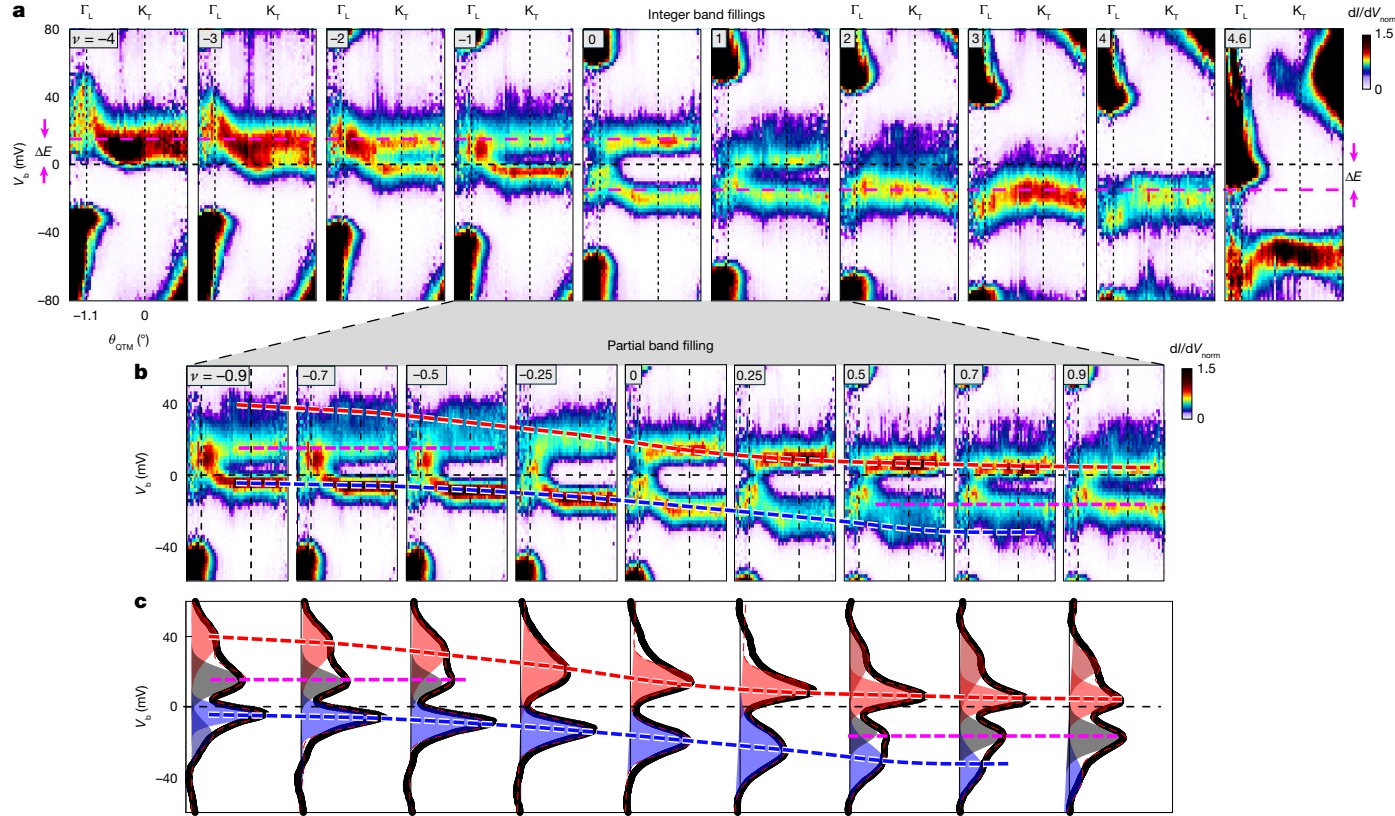

**Fig. 3 | Evolution of the energy bands with filling. a**, Momentum-resolved d$I$/d$V_{norm}$ (normalized as in Figs. 1e and 2b) versus $\theta_{QTM}$ and $V_b$ for all integer fillings $v = -4$ to $v = 4$ and for $v = 4.6$ ($\Gamma_L$ and $K_T$ indicated on the top $x$-axis). At all fillings, the bands are flat throughout, except near the $\Gamma_L$ point. At neutrality, the two bands are symmetric around the Fermi level, $E_F$ ($V_b = 0$), and are separated by about 35 mV, except at $\Gamma_L$, where they touch. At integer electron doping, the conduction flat band is pinned at $E_F$ from above, whereas the valence flat band lies $\Delta E \approx 15$ mV below $E_F$ (dashed purple line). At integer hole doping, the valence flat band pins to $E_F$ from below and the conduction flat band shifts to $\Delta E \approx 15$ meV above it (dashed purple line). At $v = 4.6$, the flat bands are fully occupied and $E_F$ lies within the remote bands near $\Gamma_L$. **b**, d$I$/d$V_{norm}$ versus $\theta_{QTM}$ and $V_b$ at fractional fillings between $v = -0.9$ and $v = 0.9$ (error bar $\delta v \approx 0.1$). The dashed red line marks one band's evolution from a high-energy smeared 'plume' at negative

doping to a narrow energy level pinned above $E_F$ at positive doping. The dashed blue line marks the opposite evolution of another band. The dashed purple line marks two more energy bands that coexist with the former and remain fixed at $V_b = \pm 15$ mV. **c**, Linecuts of d$I$/d$V$ versus $V_b$ (black lines) at the same filling factor as panel **b**, averaged over the flat-band region (Methods section 'Spectroscopy at different momenta: its Gaussians decomposition and spectral function peak and lifetime analysis versus quasiparticle energy'). The linecuts are decomposed into Gaussian contributions, marked by coloured peaks, revealing a multipeak structure with varying widths and amplitudes. Guiding lines from **b** are overlaid with the corresponding colour of the Gaussian peaks. The sum of Gaussian contributions in each panel (dashed red lines) shows a good fit. a.u., arbitrary units.

wider than the valence band, but we note that this is because of further excitations (discussed with Fig. 4). Tracing the centre energy of the bands versus momentum (Fig. 2c, dots) shows that the bands are flat to within about 4 meV. Compared with the bands predicted by the non-interacting BM model for $\theta_{TBG} = 1.1°$ (grey lines), we see a substantial difference, demonstrating the central role of interactions in determining their shape.

As well as revealing band energies, the spectral function also carries crucial amplitude information. Notably, we observe a gradual reduction in the d$I$/d$V$ amplitude of the flat bands along the momentum trajectory from $K_T$ to $K_B$ (Fig. 2a). To quantify the total spectral weight of a specific band—independent of its lifetime—we define the peak intensity, $I_{peak} = \int dI/dV \cdot dV_b$, obtained by integrating the area under its corresponding d$I$/d$V$ peak. Plotting $I_{peak}$ versus $\theta_{QTM}$ for the conduction flat band (Fig. 2d, dots) reveals a sharp increase near the left $\Gamma$ point, followed by a gradual decline towards the $K_T$ point and a steeper drop to near zero at $K_B$. Beyond $K_B$, the intensity remains negligible, falling below the measurement noise floor.

The magnitude of d$I$/d$V$ encodes tunnelling matrix element information, offering further insights into the character of individual wavefunctions. Specifically, d$I$/d$V$ is proportional to $A(k, \omega) \cdot P_t(k) \cdot P_s(k)$, in

which $A(k, \omega)$ is the spectral function, $P_t(k)$ is the projection operator onto the top TBG layer and $P_s(k) = \langle \sigma_x + I \rangle_k$ is the sublattice projection operator (Methods section 'The tunnelling matrix elements'). At the magic angle, the BM model predicts that almost all wavefunctions in the mini-BZ have equal population on the top and bottom layers, $P_t(k) \approx 0.5$ (Methods section 'The tunnelling matrix elements'). However, the sublattice projection should exhibit strong momentum dependence, which is highly sensitive to the ratio $w_0/w_1$, which reflects the relative size of AA and AB stacking regions. Comparing our measurements with theoretical predictions[47] for various $w_0/w_1$ ratios (coloured traces), we find excellent agreement for $w_0/w_1 = 0.6$, whereas other values show substantial discrepancies. This value represents a notable deviation from the accepted value of 0.8 and could have marked implications on the physics predicted in MATBG[33,48].

## Evolution of the interacting bands with filling

Figure 3a presents the energy bands at all integer fillings from $v = -4$ to $v = 4$, as well as at $v = 4.6$. At most integer fillings, two sharp and extremely flat bands are observed. At charge neutrality ($v = 0$), these bands are symmetrically positioned about the Fermi energy,

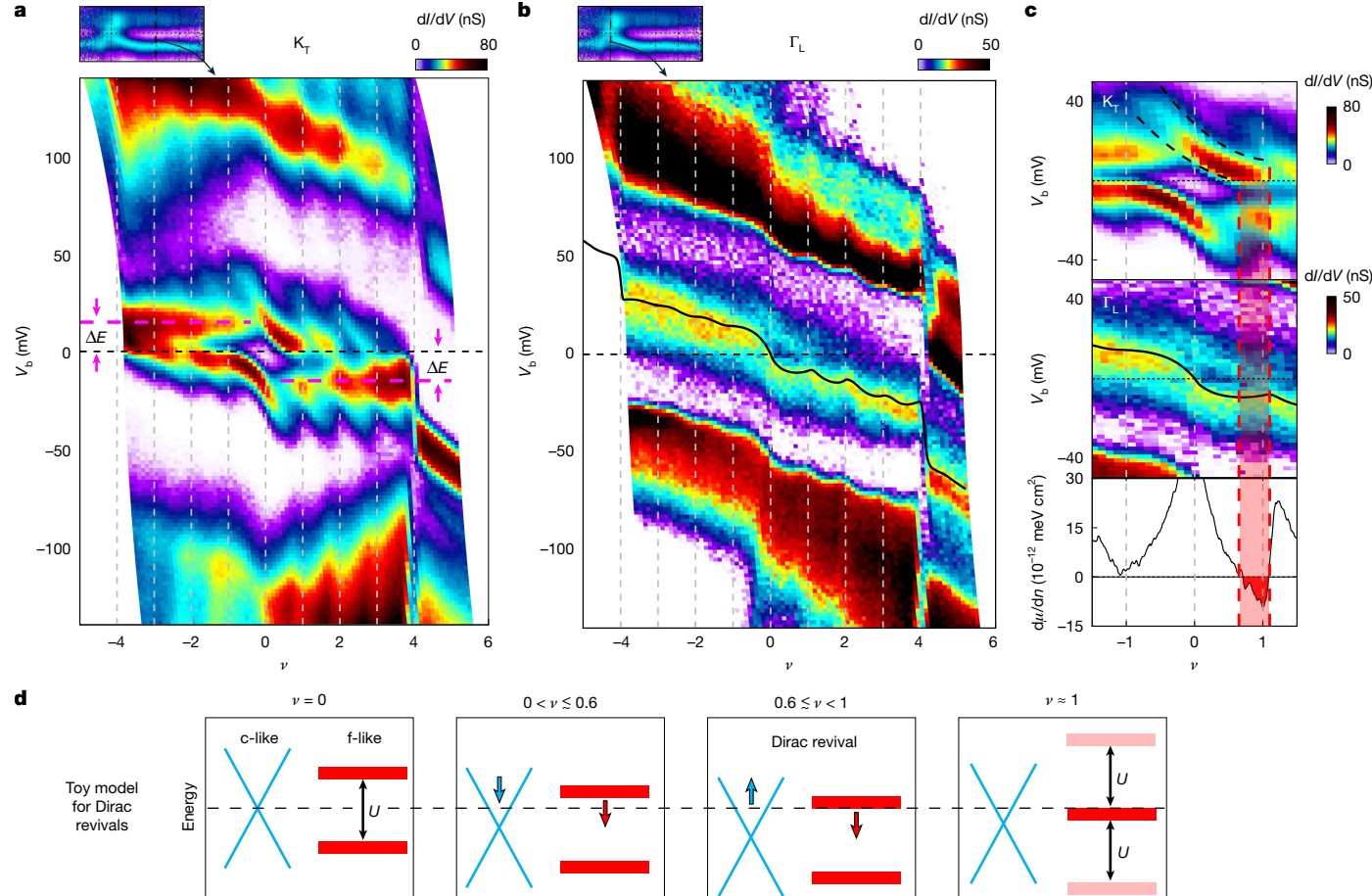

**Fig. 4 | Spectroscopy at fixed momentum points. a**, d$I$/d$V$ versus $\nu$ and $V_b$ at $K_T$ (top inset indicates location on Fig. 2b). The data show clear cascading behaviour of the flat bands, identical across all momenta along the flat bands (Extended Data Fig. 5). Fixed-energy excitations appear at $\Delta E \approx 15$ meV for hole doping and $\Delta E \approx -15$ meV for electron doping (dashed purple lines). **b**, d$I$/d$V$ versus $\nu$ and $V_b$ at the $\Gamma$ point (inset). Overlaid is $-\mu(\nu)$ from compressibility measurements in ref. 17 (black line), showing surprising agreement between the filling dependence of the d$I$/d$V$ peak at $\Gamma_L$ and the filling dependence of the total chemical potential. **c**, Correlated doping evolution of spectral features at $K_T$ (top) and $\Gamma$ (middle), alongside inverse compressibility from ref. 17 (bottom). Top and middle panels are zoom-ins of **a** and **b** near charge neutrality. The flat band at $K_T$ (heavy electrons) shifts downward with increasing $\nu$, reaching the Fermi level near $\nu \approx 0.6$, at which it begins to fill (dashed black lines).

Simultaneously, the Dirac point at $\Gamma$ (light electrons) also shifts downward and then reverses, moving up as the heavy band becomes populated. This reversal correlates with inverse compressibility becoming negative, signalling charge reshuffling from light to heavy electrons. **d**, Schematic toy model of Dirac revivals: light c-like electrons with Dirac dispersion (cyan) and heavy f-like electrons with flat bands (red). c–f hybridization is neglected. At $\nu = 0$, the light-electron Dirac point lies at $E_F$, with the heavy-electron Hubbard bands centred around it. At $0 < \nu \lesssim 0.6$, light electrons fill and heavy-electron bands shift towards $E_F$. For $0.6 \lesssim \nu < 1$, the heavy band crosses $E_F$ and becomes populated, whereas the Dirac point of the light electrons moves back towards $E_F$, driving the observed 'Dirac revivals'[17]. At $\nu \approx 1$, a new heavy band appears at energy $U$ above, restarting the cycle.

$E_F$ ($V_b = 0$ mV). On electron doping, the conduction flat band becomes pinned at $E_F$ from above, whereas the valence flat band lies $\Delta E \approx 15$ meV below $E_F$. The opposite occurs for hole doping, in which the valence flat band becomes pinned to $E_F$ from below and the conduction flat band shifts to $\Delta E \approx 15$ meV above it. By $\nu = 4.6$, $E_F$ has moved well into the remote bands.

Although the bands shift in a mostly rigid manner—maintaining their flatness across most of momentum space—the behaviour near the $\Gamma$ point is distinct. At $\nu = 0$, the $\Gamma$-point states lie symmetrically between the flat bands, but on electron or hole doping, they undergo a pronounced, filling-dependent energy shift: a local peak at $\nu = -4$ continuously transforms into a dip at $\nu = 4$. This evolution—highlighted schematically in Extended Data Fig. 9—produces an effective 'stretching' of the bands around $\Gamma$, reaching energy shifts of about 42 meV at $\nu = 4$ and about 54 meV at $\nu = -4$.

This $\Gamma$-point stretching arises naturally once we consider how different momentum states respond to the Hartree potential that develops with carrier doping. The key point is that the topological nature of

the flat bands dictates that wavefunctions near the $\Gamma$ point are fundamentally different from those elsewhere in the mini-BZ—a feature already present in the single-particle BM model and which underlies the c/f decomposition used in THF models[36–38,49,50]. The flat-band states, which occupy most of the mini-BZ, are highly localized in real space, on the AA regions of the moiré cell. Conversely, the $\Gamma$-point states are extended and mostly avoid AA regions. As filling increases, electrons predominantly occupy these AA-localized states, producing a spatially varying Hartree potential that is strongly peaked at AA. Momentum states with large AA weight therefore experience large Hartree energy shifts, whereas the $\Gamma$-point states—with minimal AA overlap—shift only weakly. When energies are referenced to the Fermi level, this momentum-dependent Hartree response manifests precisely as the $\Gamma$-point stretching observed in Fig. 3a. Extended Data Fig. 9 and its accompanying discussion (Methods section 'Hartree-driven band stretching and its connection to wavefunctions at different momenta within the flat band') provide a step-by-step explanation of this 'Hartree stretching' mechanism.

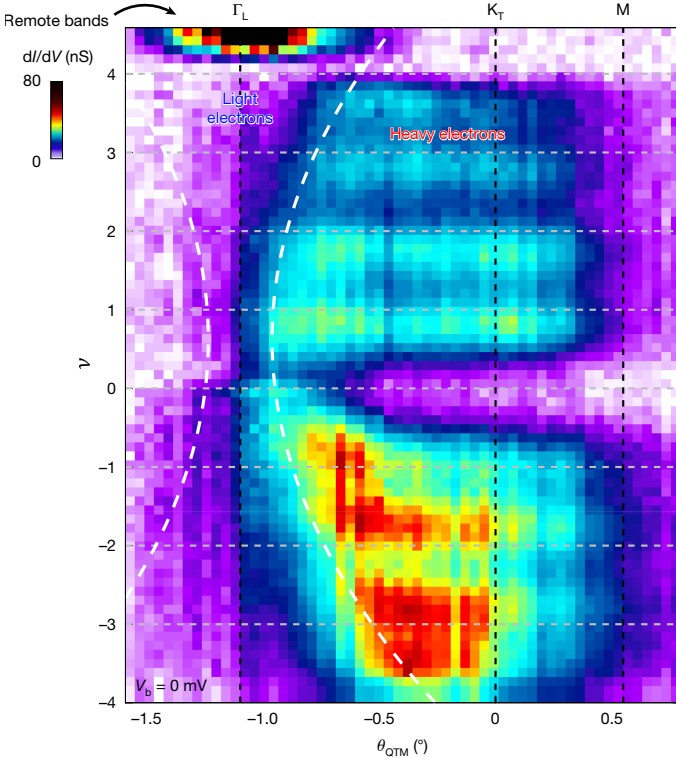

**Fig. 5 | Doping evolution of the relative fraction of light and heavy electrons in momentum space.** d$I$/d$V$ versus $\theta_{QTM}$ and $v$ measured at $V_b$ = 0 mV. This signal is proportional to the spectral weight at the Fermi energy and shows where electronic states reside in momentum space as a function of doping. The momenta $\Gamma_L$, $K_T$ and M are labelled on the top $x$-axis and integer fillings are indicated by horizontal grey lines. We plot the raw d$I$/d$V$ signal, whose magnitude drops sharply left to the $\Gamma_L$ point owing to tunnelling matrix elements. At charge neutrality ($v$ = 0), the spectral weight is concentrated near $\Gamma$ and is absent elsewhere. With doping, horizontal features emerge across a wide momentum range, reflecting the spectral weight of the flat heavy-electron bands, appearing at the Fermi level. This weight oscillates with a periodicity $\Delta v \approx 1$, consistent with the cascades of these bands. Near $\Gamma_L$, the spectral weight disappears owing to the Hartree-induced bending that pushes the light-electron bands below the Fermi energy. The horizontal features get progressively shorter with increased filling, as highlighted by the dashed lines, tracing the momentum at which the d$I$/d$V$ intensity falls to 80% of its value at each $v$ (Methods section 'Fitting the transition between light and heavy electrons'). This boundary is mirrored about the $\Gamma$ point; together, the two dashed lines delineate the region in momentum space occupied by light electrons, with the remaining area populated by heavy electrons.

Further insight into the evolution of the electronic structure emerges when we examine partial moiré fillings. Figure 3b plots this evolution from $v = -0.9$ to $v = 0.9$, with dashed lines following the progression of key spectral features. The dashed red line tracks a flat band that appears initially at $v = -0.9$ at high energy as a diffuse 'plume'—reflecting an incoherent band (spectral peak width comparable with its energy). As filling increases, this band gradually moves towards $E_F$, increasing in spectral weight as it approaches. The dashed blue line follows a band with the opposite trend: it begins as a band with strong spectral weight just below $E_F$ and progressively shifts to large negative energies while becoming increasingly smeared. Notably, in both cases, just before the sharp band vanishes while crossing $E_F$ at $v \approx 1$ or $v \approx -1$, a new diffuse band appears at high energy. This behaviour is most clearly seen in Fig. 3c, which presents the energy spectrum, momentum-averaged over the flat part of the bands (excluding the region near $\Gamma$; Methods section 'Spectroscopy at different momenta: its Gaussians decomposition and spectral function peak and lifetime

analysis versus quasiparticle energy'), along with its decomposition into Gaussian contributions.

## Filling-dependent spectroscopy at key momenta

Recent topological heavy fermion[36–38] and Mott semimetal[39] theories predict that the dual-nature electrons in MATBG arises from the different character of the electronic states at different momenta within the flat bands. To investigate this, we use a key capability of the QTM—performing spectroscopy as a function of filling at a particular momentum.

Figure 4a presents d$I$/d$V$ measured as a function of $V_b$ and $v$ at momentum $K_T$. At charge neutrality, a clear energy gap is observed at this momentum. The evolution of the energy bands with filling—previously shown in Fig. 3b—appears here in the familiar form of electronic cascades[17,18]: near each integer $v$, a faint high-energy band emerges, which gradually shifts with filling to lower energies, becoming stronger and sharper the closer it approaches $E_F$. This band crosses $E_F$ over a finite range of $v$ and, once fully occupied, a new incoherent high-energy band emerges, whereas the previous one persists below $E_F$. This cascading behaviour recurs at all integer fillings and is more pronounced on the electron-doped side. Identical filling-dependent spectral evolution is observed at all other momenta along the flat part of the bands (Methods section 'Spectroscopy at different momenta: its Gaussians decomposition and spectral function peak and lifetime analysis versus quasiparticle energy'). This explains why scanning tunnelling microscopy, which averages over all momenta, captures a similar pattern.

A markedly different behaviour is seen at the $\Gamma$ point (Fig. 4b). Here there is no apparent gap at neutrality nor do we see any sign of cascading bands crossing through $E_F$ with filling. Instead, the d$I$/d$V$ shows a peak at $E_F$ at $v \approx 0$, which gradually shifts to negative energies with filling. The shift is not strictly monotonic but exhibits wiggles that align with integer fillings, more prominently seen on the electron-doped side. Notably, when we overlay the negative of the chemical potential, $-\mu(v)$, obtained from the compressibility measurements of ref. 17 (black line), we find that it closely follows this peak, even capturing its wiggles. Because that measurement was performed in a sample with a slightly larger twist angle ($\theta_{TBG}$ = 1.13°), we do not expect perfect correspondence.

Zooming in further (Fig. 4c), we uncover a strong correlation between the behaviour at the $K_T$ point, the $\Gamma$ point and the inverse compressibility, d$\mu$/d$n(n)$, from ref. 17 (Fig. 4c top, centre and bottom). From $v \approx 0$ to $v \approx 0.6$, the peaks at both $\Gamma$ and $K_T$ shift monotonically to lower energies with increasing filling. At $v \approx 0.6$, as the flat band at $K_T$ reaches $E_F$, the peak at $\Gamma$ reverses direction and begins to increase in energy. Notably, this turning point coincides with a sign change in the total inverse compressibility, which becomes negative. This behaviour continues until $v \approx 1$, at which the flat band at $K_T$ becomes fully occupied, the inverse compressibility sharply rises and the peak at $\Gamma$ reverses direction again.

These observations naturally follow from the coexistence of two classes of electronic states that satisfy two basic conditions: they have light and heavy character and they couple differently to the Hartree potential. Although these conditions are naturally realized by the microscopic BM model 'c' and 'f' orbitals, they can be satisfied much more broadly by any two 'c-like' and 'f-like' components that retain these essential features. This is illustrated by a simple toy model (Fig. 4d and Methods section 'Toy model to understand the essence of the Dirac revivals'), in which the light c-like electrons near $\Gamma$ are described by a Dirac dispersion (cyan) and the heavy f-like electrons by flat bands (red). For simplicity, we neglect c–f hybridization, important in the actual bands but not essential for understanding the non-monotonic filling of the light electrons. In this picture, the measurement at $\Gamma$ tracks the filling-dependent evolution of the Dirac point, whereas the measurement at $K_T$ follows the flat bands. At $v = 0$, the flat bands lie away from $E_F$ and the Dirac point sits at $E_F$. On doping, carriers initially populate the c-like bands, shifting their Dirac point downward

in energy. Around $v \approx 0.6$, a flat band reaches $E_F$ and begins to fill, at which point the Dirac carriers start to depopulate and the Dirac point shifts upward towards $E_F$. This behaviour is identical to 'Dirac revivals' of ref. 17, except that here the reshuffling of charge occurs between the light and heavy electrons rather than between flavours, as originally postulated[17]. The resulting evolution reflects a competition between a c−f Coulomb repulsion term, which drives the revivals and tends to depopulate the c-like electrons (producing upward wiggles), and Hartree terms, which lead to progressive population of the c-like electrons with filling (producing the overall downward trend). This competition naturally explains why the revivals do not fully reset to the Dirac point at integer fillings.

Although some key aspects of our experiments are captured by theoretical models of interacting electrons in MATBG[23,24,27,28,34,35,38,39], one persistent feature remains unexplained: the excitation appearing at $\Delta E \approx 15$ meV for holes and $\Delta E \approx -15$ meV for electrons (purple lines in Figs. 3a and 4a). Similar features were previously observed in non-momentum-resolving scanning tunnelling microscopy measurements[16,18], but our momentum-resolved experiments now show that this excitation exists only in the flat section of the bands (Fig. 4a) and not at the Γ point (Fig. 4b), suggesting its connection with the heavy electrons. Notably, its energy remains fixed with filling, in contrast to the strongly filling-dependent evolution of the heavy-electron bands themselves. The universality of this 15 meV scale—identical across widely separated regions in the sample and insensitive to local strain (Methods section 'Spectroscopy of flat bands at different locations')—may hint at the presence of an intrinsic excitation or collective mode whose microscopic origin remains unknown, but which may play an important role in the correlated and superconducting behaviour of MATBG.

## Doping dependence of the light and heavy electrons

Finally, we present in Fig. 5 the zero-bias $dI/dV$, which reflects the spectral weight at $E_F$. When plotted as a function of $\theta_{QTM}$ and $v$, this quantity maps out where electronic states reside in momentum space as a function of doping. In this plot, the heavy electrons flat bands manifest as horizontal features extending over a wide range in momentum space. Their spectral weight oscillates with filling with a periodicity $\Delta v \approx 1$, consistent with the cascading behaviour of the heavy electrons. The spectral weight decays near the Γ point owing to the Hartree stretching of the bands that pushes the light electrons below the Fermi energy. Visibly, the horizontal features get progressively shorter with filling (highlighted by the dashed white lines), indicating that the region of k-space occupied by heavy electrons shrinks, whereas the region occupied by light electrons expands.

Our measurements resolve a long-standing puzzle in the physics of MATBG: the dual nature of its electrons. By directly visualizing the energy bands, we show that this duality arises from distinct characteristics of the flat bands' wavefunctions in different regions of momentum space. Near the Γ point, the bands are dispersive, consistent with c-like electron behaviour, whereas elsewhere, they remain extremely flat, characteristic of f-like electrons. We directly observe how this leads to stretching of the bands owing to Hartree potential, as well as to electronic cascades and 'Dirac revivals' owing to the reshuffling of charge between these two degrees of freedom. Qualitatively, our findings provide direct evidence for the THF and Mott semimetal descriptions of this system. However, we also uncover a new low-energy excitation, not accounted by any existing models, which may play an important role in the exotic physics of MATBG.

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

## Methods

### Cryogenic QTM

All measurements in this work were performed in a cryogenic QTM system operating at a temperature of 4 K, as described previously[45]. For conductance measurement, voltage biases are applied using a custom-built digital-to-analogue converter array, capable of supplying both d.c. and a.c. signals with 1 µV resolution. Currents were measured using a FEMTO current amplifier, followed by a National Instruments sampler.

### Fabrication and characterization of the QTM tip and van der Waals device

The fabrication of both QTM tips and van der Waals devices on flat substrates were described previously[45,46]. Briefly, we use the standard dry transfer and polymer membrane transfer techniques to fabricate the TBG sample and the QTM tip, respectively. Extended Data Fig. 1a shows an optical image of the TBG heterostructure, comprising bilayer-WSe$_2$/TBG/hexagonal boron nitride (h-BN)/Pt on an approximately 100 µm width and 80 µm tall Si/SiO$_2$ pillar. Extended Data Fig. 1b shows an optical image of the MLG backed by h-BN and graphite placed on a QTM tip.

In Extended Data Fig. 1c, we characterize the TBG sample by conductive AFM scan of the region of magic twist angle carried out at room temperature (Bruker Icon). To compensate for the thermal drift during this AFM scan, we performed a small window scan at rate 10.9 Hz, such that the time for an entire scan is around 20 s. We apply an approximately 1 V d.c. voltage bias between the tip and sample and measure the tunnelling current. The scan image in Extended Data Fig. 1c shows both the moiré lattice and atomic defects in WSe$_2$ (bright spots). To accurately correct the thermal drift, we trace the position of three defects inside the scan window. We then obtain the drift velocity by following the shift of defects positions between consecutive scans and assuming that, within a single 2D scan, the drift velocity is constant. Extended Data Figure 1c,d shows the image and corresponding fast Fourier transform (FFT) after thermal drift correction. The three moiré lattice vectors obtained from this image are: 12.72 nm, 12.73 nm and 13.14 nm. This gives a twist angle of $\theta_{TBG} = 1.1° \pm 0.02°$ and strain of $\epsilon = 0.03\% \pm 0.02\%$.

In Extended Data Fig. 1e, we characterize the QTM tip by in situ imaging of the tip contact area by an atomic defect in the WSe$_2$ barrier as a localized tunnelling channel. At large bias ($V_b \approx -900$ mV, well outside the range used in the actual spectroscopy measurements), the defect level becomes energetically accessible and provides an extra, spatially localized current path. When the QTM tip passes over the defect, we observe a small but measurable increase in current. By scanning the tip across such defects and mapping this current enhancement, we obtain a real-space image of the tip's contact footprint.

Extended Data Fig. 1e presents such a high-bias scan, showing several images of the tip footprint, each produced by a different defect, revealing a contact size of approximately 200 × 50 nm. These real-space dimensions set the momentum-space resolution through the Heisenberg uncertainty relation. The measured footprint corresponds to angular momentum resolutions (in angular units) of roughly $\delta\theta_{QTM} \approx 0.05°$ and $\approx 0.2°$ along the two principal directions. In Extended Data Fig. 2, we show that the sharpest momentum features—obtained in the remote bands—exhibit a full width at half maximum (FWHM) of $\delta\theta_{QTM} \approx 0.1°$, consistent with the resolution expected from the measured tip dimensions.

### Energy and momentum resolution

We determine the energy resolution of our QTM measurements by analysing a sharp spectral feature at $\nu = -0.7$ in Fig. 4a (reproduced in Extended Data Fig. 2a). A linecut through this feature yields a FWHM of $\delta E \approx 10$ meV by fitting a Lorentzian function. This is an upper bound to the energy resolution of the measurements and it may well be coming from the intrinsic lifetime of the energy bands of MATBG at this

temperature. To determine the momentum resolution, we analyse a sharp spectral feature of the remote energy bands of MATBG measured under the same conditions as in Fig. 2a. Extended Data Fig. 2b plots a linecut of d$I$/d$V$ along the dashed white line, cutting through the dispersive bands, yielding an angular FWHM of $\delta\theta = 0.1°$ by fitting a Lorentzian function. This translates into a $\delta k \approx 0.03$ nm$^{-1}$ momentum resolution. For the flat bands, we trace the d$I$/d$V$ peak width versus $\theta_{QTM}$ at zero bias around the Γ point and identify the peak width $\delta\theta \approx 0.5°$. This peak width is composed of two peaks overlapping each other, such that the single peak broadening is $\delta\theta \approx 0.25°$.

### Electrostatics of the QTM junction

The QTM junction is modelled as a three-plate capacitor consisting of: (1) the MLG on the QTM tip; (2) the TBG layer on the sample; and (3) the metallic back gate. Below, to stay consistent with our previous publication, we use in all of the formulas the notation 'top layer' ('T') and bottom layer ('B') to refer to the two layers between which the electrons perform the QTM's momentum-resolved tunnelling, not the top and bottom layers of TBG. Namely, the top layer is the MLG layer on the QTM tip and the bottom layer is the TBG on the sample side.

The QTM tip and TBG are separated by two layers of WSe$_2$ (1.2 nm) and the TBG to the back gate by a 37 nm h-BN layer. The corresponding interlayer geometric capacitances are denoted as $C_g$ (tip–TBG) and $C_{bg}$ (TBG–back gate). A voltage bias $V_b$ is applied between the tip and TBG sample layer. The gate voltage $V_{bg}$ is applied between the TBG sample and the metal back gate. The electrostatic configuration of the measurements can be described by the following[46]:

$$-eV_b = \mu_B - \mu_T - \frac{e^2 n_T}{C_g}, \tag{1}$$

$$eV_{bg} = \mu_B + \frac{e^2(n_B + n_T)}{C_{bg}} \tag{2}$$

Note again that the notation 'top' ('T') refers to the MLG on the QTM tip and the notation 'bottom' ('B') refers to the TBG. $\mu_B$ and $\mu_T$ are the chemical potentials of the bottom TBG and top graphene probe, respectively, and $n_B$ and $n_T$ are their corresponding carrier densities. The back-gate capacitance was experimentally calibrated to be $C_{bg} = 62.3$ nF cm$^{-2}$ based on the gate voltage required to reach filling factor $\nu = 4$ in the magic-angle region (as verified by the conductive AFM measurements; Methods section 'Fabrication and characterization of the QTM tip and van der Waals device').

It is important to note that, under a finite voltage bias, TBG becomes doped as a result of the gating from the tip. This means that, at a fixed $V_{bg}$, the filling factor $\nu$ changes with the tip bias, $V_b$. To account for this and present our data as a function of the TBG's filling factor, we determine the gating effect directly from the data. Extended Data Fig. 3 shows the raw measurement of QTM tunnelling conductance, d$I$/d$V_b$ (for simplicity called d$I$/d$V$ throughout the paper), and the transconductance with respect to the back gate, d$I$/d$V_{bg}$, as functions of $V_b$ and $V_{bg}$. At integer fillings, and in particular at $\nu = \pm 4$, d$I$/d$V_b$ shows a suppression and d$I$/d$V_{bg}$ shows a dipole-like feature. By tracing the $\nu = 4$ line in the d$I$/d$V_{bg}$ measurements (through a polynomial fitting, overlaid on the data), we extract the trajectory of constant carrier density in the $V_b$–$V_{bg}$ plane. We then 'skew' the raw data by plotting it as a function of $\nu$ that is determined by both $V_{bg}$ and $V_b$. The skew-corrected data are shown in Figs. 3 and 4. For Figs. 1 and 2, the scans were taken at a fixed gate voltage without skew correction, so at high bias, the filling factor may deviate from the nominal one.

Along this skewed line, the TBG carrier density $n_B$ and chemical potential $\mu_B$ remain fixed. By using the graphene relation $n_T = \mu_T^2/\pi\nu_f^2$, the electrostatic equation can be further simplified to:

$$-eV_b = \mu_B - eV_{bg}^* \frac{C_{bg}}{C_g} - \sqrt{eV_{bg}^* \frac{C_{bg}\pi\nu_f^2}{e^2}}, \tag{3}$$

in which $eV_{bg}^* = eV_{bg} - \mu_B - e^2 n_B/C_{bg}$, giving for every $V_b$ the corresponding shift in $V_{bg}$ to maintain fixed $n_B$ and $\mu_B$. By fitting the skew lines with this relation, we derive the interlayer capacitance to be $C_g = 2.3\ \mu F\ cm^{-2}$, consistent with our previous tunnelling experiment using two layers of $WSe_2$ (ref. 46).

## The tunnelling matrix elements

The tunnelling current is modelled by Fermi's golden rule assuming momentum-resolved tunnelling between the TBG sample and the graphene probe. As in the previous section, to stay consistent with our previous publication, in all of the formulas below, we use the notation 'top layer' ('T') and 'bottom layer' ('B') to refer to the two layers between which the electrons perform the QTM's momentum-resolved tunnelling, not the top and bottom layers of TBG. Namely, the top layer is the MLG layer on the QTM tip and the bottom layer is the TBG on the sample side.

The calculation is described in detail in refs. 46,47 and here we briefly review the formula and tunnelling matrix elements:

$$I = \frac{2\pi e g_s g_v}{\hbar} \int d\omega \left( f_B(\omega) - f_T(\omega + e\phi) \right)$$
$$\sum_{n,n'} \sum_{\mathbf{k}_T, \mathbf{k}_B} \left| \langle u_{gr, \mathbf{k}_T, n} | T^{\mathbf{G}_T, \mathbf{G}_B} | u_{tbg, \mathbf{k}_B, n'} \rangle \right|^2 \delta_{\mathbf{k}_T + \mathbf{G}_T, \mathbf{k}_B + \mathbf{G}_B} A_{\mathbf{k}_B, \omega} A_{\mathbf{k}_T, \omega + e\phi} \tag{4}$$

$g_{s/v}$ is the spin/valley degeneracy, $\mathbf{k}_{B/T}$ is the electron momentum of the bottom/top layer, $n$ and $n'$ are the band indexes, $f_{B/T}(E)$ is the Fermi–Dirac distribution function and $\phi$ is the electrostatic potential that shifts the relative position of energy bands between TBG and graphene, derived from the electrostatic model in Methods section 'Electrostatics of the QTM junction'. The chemical potential $\mu_{B/T}$ is defined relative to their charge neutrality (Dirac point of TBG and graphene probe). $A_{\mathbf{k}_{B/T}, E}$ is the spectral function of the TBG and the graphene probe. $T^{\mathbf{G}_T, \mathbf{G}_B}$ is the tunnelling matrix in sublattice space between the graphene probe wavefunction $|u_{gr, \mathbf{k}_T}\rangle$ and the TBG wavefunction $|u_{tbg, \mathbf{k}_B}\rangle$ projected on the top layer. $\mathbf{G}_T$ and $\mathbf{G}_B$ are the reciprocal lattice vectors of the graphene probe and the top layer graphene of TBG, at which it satisfied the relation $\mathbf{k}_T + \mathbf{G}_T = \mathbf{k}_B + \mathbf{G}_B$. We assume that the tunnelling happens within the first BZ of graphene probe and it limits the Bragg scattering process of $\{\mathbf{G}_T, \mathbf{G}_B\}$ choices to be three. For example, when $\mathbf{G}_T = \mathbf{G}_B = 0$, the matrix is given by: $T^{\mathbf{G}_T=0, \mathbf{G}_B=0} = I + \sigma_x$, in which $I$ is the unity matrix. The other two Bragg scattering processes are related by $C_3$. Here we focus on $T^{\mathbf{G}_T=0, \mathbf{G}_B=0}$ and the matrix elements are given by:

$$\left| \langle u_{gr, \mathbf{k}_T} | T^{\mathbf{G}_T=0, \mathbf{G}_B=0} | u_{tbg, \mathbf{k}_B} \rangle \right|^2$$
$$= \left| \langle (u_{gr, \mathbf{k}_T, A}^* + u_{gr, \mathbf{k}_T, B}^*) \rangle_{FS} (u_{tbg, \mathbf{k}_B, A} + u_{tbg, \mathbf{k}_B, B}) \right|^2 \tag{5}$$
$$\propto |u_{tbg, \mathbf{k}_B, A} + u_{tbg, \mathbf{k}_B, B}|^2 = |\langle (I + \sigma_x) u_{tbg, \mathbf{k}_B} \rangle|,$$

in which $u_{gr, \mathbf{k}_T, A/B}^*$ is the graphene probe wavefunction with the sublattice component $A/B$. The graphene probe wavefunction is averaged over the Fermi surface and we assume that, within the graphene Fermi surface range, the TBG wavefunctions $u_{tbg, \mathbf{k}_B, A/B}$ remain the same. The above equation shows that the tunnelling current is proportional to the $|\langle (I + \sigma_x) u_{tbg, \mathbf{k}_B}\rangle|$, in which the $|u_{tbg, \mathbf{k}_B}\rangle$ contains both the layer polarization and sublattice information.

In Fig. 2d, we show the matching between the matrix element $|\langle (I + \sigma_x) u_{tbg, \mathbf{k}_B}\rangle|$ and the $dI/dV$ intensity in the flat-band region. It shows an excellent agreement at ratio $w_0/w_1 = 0.6$. Here we further show that the naive layer polarization quantity (Extended Data Fig. 4b,d) is inconsistent with the data, regardless of the $w_0/w_1$ ratios.

For the tunnelling current calculation in Fig. 1, we input the single-particle band structure of TBG at $\theta_{QTM} = 1.2°$, the electrostatics from Methods section 'Electrostatics of the QTM junction' and calculate the tunnelling current based on equation (4).

## Spectroscopy at different momenta: its Gaussians decomposition and spectral function peak and lifetime analysis versus quasiparticle energy

In Extended Data Fig. 5, we show the tunnelling spectroscopy at different momenta along the flat-band region. The momenta at which the spectroscopy images are measured are marked by dashed black lines on a zoom-in of the bands from Fig. 2b, shown in the top panel. The range of momenta covers angles from $\theta_{QTM} = -0.5°$ to $0.8°$. Apart from an overall magnitude, the spectroscopy in all momenta looks very similar. In Fig. 3c, we use this invariance on momentum in the flat bands to make Gaussian fit to the averaged data over the flat-band region.

The Gaussian fitting procedure is performed as follows. We first identify the peak centres directly from the spectroscopy data by tracing the local maxima in the contour representation of Fig. 4a; these trajectories are shown as dashed black lines in Extended Data Fig. 6a. Specifically, we identify two classes of features: (1) the 'Hubbard' bands that exhibit the characteristic cascading behaviour and (2) further excitations whose energy (about ±15 meV) is independent of filling. For each filling, the number of peaks identified in this manner determines the number of Gaussians used in the corresponding spectral fit. The peak positions extracted from the dashed-line trajectories serve as initial guess for the fit, although we allow these positions to vary during the fitting. The free parameters in the fitting routine are the Gaussian centres, widths, amplitudes and the background offset.

In Figs. 3b and 4a, we observed many spectral features originating from the Hubbard bands of heavy electrons. The spectral features typically appear at high energy as smeared 'plumes' and become stronger the closer they approach the Fermi level.

In a simple lifetime-broadening picture, the spectral function peak corresponding to a quasiparticle at an energy $E$ acquires a width inversely proportional to the lifetime and the peak amplitude of the spectral function decreases correspondingly. In the present experiment, several bands coexist within a narrow energy range, making it challenging to extract the linewidths of the bands quantitatively. Instead, we can reliably extract the peak height, $A_{peak}$, which is a robust experimental observable and reflects the combined effects of quasiparticle residue and lifetime. Inspection of Fig. 4 shows a clear and systematic trend: when a band lies far from the Fermi energy, its $A_{peak}$ is small and this maximal spectral weight increases continuously as the band approaches the Fermi level.

To address this more quantitatively, in Extended Data Fig. 6b, we track the maximum of the spectral peak (dashed line) from the highest energy that we can reliably identify a peak to the point at which it reaches the Fermi energy, and the peak becomes less well defined. In panel c, we plot the one over the extracted peak height $1/A_{peak}$ versus peak energy $E$ and compare it with two functional forms: $\frac{1}{A_{peak}} = a(E - E_F) + b$ and $\frac{1}{A_{peak}} = a(E - E_F)^2 + b$. Overall, we see that a linear dependence of $1/A_{peak}$ on $E - E_F$ better fits the observations.

The height of a spectral-function peak reflects both the quasiparticle residue $Z(E - E_F)$ and the lifetime $\tau(E - E_F)$. In principle, both quantities may evolve as a Hubbard band moves to higher energies. If, however, we interpret the dominant trend as arising mainly from lifetime effects, the observed scaling would imply that it is more likely that $\tau(E - E_F) \propto 1/(E - E_F)$ and not the Fermi-liquid expectation $\tau(E - E_F) \propto 1/(E - E_F)^2$. Such dependence is often seen in strongly correlated/Hubbard systems, but because we do not measure this quantity in separation, we cannot make a decisive claim.

## Fitting the transition between light and heavy electrons

Figure 5 shows $dI/dV$ at $V_b = 0$ mV as a function of $\theta_{QTM}$ and filling factor $v$. To determine the boundary between heavy and light electrons, we plot horizontal linecuts of $dI/dV$ versus $\theta_{QTM}$ for each filling and perform a fit of a sigmoid-type function:

$$\frac{dI}{dV} = c_1 + \frac{c_2 - c_1}{1 + e^{-\frac{\theta_{QTM} - \theta_0}{\sigma}}} \qquad (6)$$

On the basis of the fitting, when the heavy-electron spectral intensity decays by 80%, we mark the position $\theta_{hl}(v)$ as a proxy for the transition between heavy and light electrons. The parameter $\sigma$ characterizes the width of this transition. We then fit $\theta_{hl}(v)$ to a third-order polynomial and plot this as the dashed white lines in Fig. 5.

## Determine the larger twist angle of $\theta_{TBG} = 1.2°$

We determine the angle of larger-angle TBG from the measurement in Fig. 1d. By tracing the $dI/dV$ intensity change at the conduction-band edge of flat bands around the Γ point, we performed a step function fitting of the form:

$$y = C + \frac{A}{2}(1 + \tanh(B * (x - x_0))),$$

in which the $x_0$ marks the Γ point position, which is half the value of the tunnelling intensity decay. By measuring the angle distance between the Γ point to the $K_T$ point, at which the Dirac point crosses, we determine the twist angle to be $\theta_{TBG} = 1.2°$.

## Spectroscopy of flat bands at different locations

Extended Data Fig. 7a–f presents tunnelling spectroscopy at a momentum point within the flat part of the bands ($K_T$) measured at several locations across the sample, separated by several microns. These separations are large enough such that each region effectively behaves as independent devices. In all cases, we observe excited states whose energy is independent of filling, similar to the measurement in Fig. 4a.

For each location, we use room-temperature conductive AFM to image the real-space moiré structure (Methods section 'Fabrication and characterization of the QTM tip and van der Waals device') and plot its corresponding FFT. From the latter, we extract the local twist angle, $\theta_{TBG}$, and heterostrain, $\epsilon_{TBG}$, of the moiré. From the spectroscopy, we determine the excitation energy, $\Delta E$. These three locally measured quantities are indicated above each panel. The centre panel shows a real-space map of the measurement locations, with the colour of each point representing the corresponding $\Delta E$ (see colour bar).

Although we cannot definitively rule out heterostrain as the origin of the observed approximately 15 meV excitation, the data strongly indicate that strain alone cannot account for this feature, for several reasons:

1. Universality across widely separated regions of the same device. We observe nearly identical excitation energies (13–16 meV) at locations separated by several microns. It is highly unlikely that the same level of strain would persist uniformly over such large distances.

2. Universality across different samples. Similar excitation energies are measured in two distinct MATBG devices, measured in separate cooldowns, indicating that the feature is neither device-specific nor related to a particular thermal cycle.

3. Lack of correlation with independently measured strain. The heterostrain extracted from the moiré FFT varies by a factor of 3 across the measured positions (from $\varepsilon = 0.02\%$ to $\varepsilon = 0.06\%$). Despite this substantial variation, the excitation energy remains essentially unchanged.

4. Quantitative inconsistency with theoretical expectations. For the least-strained region ($\varepsilon = 0.02\%$), the observed excitation is about 15 meV. Existing theoretical estimates predict a strain-induced level splitting of about 10 meV per 0.1% strain[51], which would correspond to only about 2 meV splitting at $\varepsilon = 0.02\%$, an order of magnitude smaller than what we observe.

## QTM probing of the K and K' valleys

In Extended Data Fig. 8, we explain why the time-reversal symmetry of the QTM probe dictates that it measures the combined contribution of the K and K' valleys.

Panel a shows the inequivalent Fermi surfaces of the K and K' valleys (red and blue) folded into the same mini-BZ. For illustration, we depict triangularly warped Fermi surfaces to emphasize that the two valleys can, in principle, be different.

To understand what the QTM actually probes, it is useful to unfold the picture back to the full BZ (panel b). There the K and K' Fermi surfaces sit at opposite corners. In the experiment, the Dirac points of the probe (purple) move along a circular trajectory (dashed line) and current is detected whenever the Dirac point of the probe intersects a Fermi surface. Crucially, if both the probe and the sample obey time-reversal symmetry, the K and K' valleys produce identical spectra as a function of the QTM angle $\theta_{QTM}$ (panel c). If we take the illustration in panel b as an example, although the two triangular Fermi surfaces are mirror reflections, the rotating Dirac points intersect them in exactly the same sequence—first the flat edge and then the triangle tip—and therefore yield identical spectroscopic measurement.

We might expect that spontaneous valley symmetry breaking (for example, a valley-polarized Chern insulator) would manifest as clear energy splitting between the K and K' bands and that, consequently, it should be easy to identify flavour symmetry breaking from this band splitting. However, this expectation is incorrect—recent dynamical mean-field theory calculations performed in both the symmetric and flavour-symmetry-broken[34,35] states show that very similar band splittings appear in both situations. These splittings originate primarily from the formation of Hubbard bands, which occur regardless of whether flavour symmetry is broken. As a result, the spectroscopic signatures of flavour symmetry breaking are subtle and cannot be identified simply by looking for energy-split bands.

## Hartree-driven band stretching and its connection to wavefunctions at different momenta within the flat band

In this section, we provide a systematic explanation of the momentum-dependent band stretching observed near the Γ point. To highlight the effect, we plot in Extended Data Fig. 9h,i the measured bands from Fig. 3a alongside schematic guides to the eye marking the stretched areas indicated by arrows. Specifically, the stretching appears as a peak at Γ at $v = -4$, which continuously evolves to a dip at Γ at $v = 4$.

This stretching originates from the momentum-dependent response of the electronic wavefunctions to the Hartree potential that builds up with increasing carrier density. Early theoretical works (for example, refs. 21,22) already noted that, although moiré-scale electrostatic potentials are negligible at charge neutrality, a spatially varying Hartree potential $V_H(r)$ develops as electrons or holes are added. The energy of a state at momentum $k$ then shifts according to its real-space overlap with this potential,

$$\delta E_H(k) = \int d^2 r\, V_H(r) |\psi_k(r)|^2,$$

and because different $k$-states have distinct real-space charge distributions, they experience different Hartree shifts. This naturally produces a stretching of the bands, even in the absence of strong correlations. Although indirect signatures of this effect have been reported previously (for example, through Landau-level spectroscopy in ref. 16), Fig. 3 presents the first, to our knowledge, direct observation of this Hartree-driven band stretching.

Notably, our measurements do not only show the overall magnitude of the band stretching but they directly provide the momentum-dependent energy shifts $\delta E_k$. From the expression above, it is evident that these shifts encode detailed information about the real-space structure of the wavefunctions at different $k$. This is especially valuable in MATBG, in which the flat bands are known to have strongly momentum-dependent wavefunctions arising from their nontrivial topology.

In fact, one of the most notable theoretical predictions for TBG, already present in the BM single-particle model, is that the flat-band

wavefunctions change qualitatively across the mini-BZ. This effect does not need electronic correlation and happens also when the bands are not extremely flat—for example, for TBG with twist angles larger than the magic angle. In the BM continuum model, more than 90% of the momentum states in the band have a localized wavefunction in the AA regions of the moiré cell. However, near the Γ point, this structure is reversed: the states become more extended and feature a suppression of charge density at the AA sites.

Although this momentum-dependent real-space structure was used as the basis for formulating the c-electron and f-electron decomposition in the THF model, it is not specific to that framework. Rather, it is a robust and very general feature of the non-interacting flat bands themselves. Our ability to measure $\delta E_H(k)$ with momentum resolution therefore provides direct access to these underlying wavefunction characteristics, offering a powerful new experimental probe of the topology-driven structure of the MATBG bands.

Extended Data Fig. 9a–g provides a step-by-step illustration of how the observed Hartree stretching arises and how it is directly connected to the wavefunction structure of the flat bands. Panel a sketches the real-space Hartree potential $V_H(r)$ that develops on electron filling. Because more than 90% of the flat-band wavefunctions in the BM model reside on the AA sites and have an f-electron Wannier function, the added carriers generate a Hartree potential that is strongly peaked at these sites and therefore highly repulsive for f-electrons. We emphasize that here we are discussing single-particle wavefunctions, so the c/f decomposition is fully valid in this context. As shown in panel b, the shape of $V_H(r)$ remains fixed as a function of filling, whereas its overall amplitude grows approximately linearly with $\nu$ (ref. 21).

Panel c plots the BM charge densities $|\Psi_f(\mathbf{r})|^2$ (red) and $|\Psi_c(\mathbf{r})|^2$ (blue). The f-electron density has far greater overlap with $V_H(r)$ than the c-electron density and thus experiences a substantially larger Hartree shift. This is captured schematically in panel d, in which both c-electron and f-electron energies increase linearly with $\nu$ but with a much steeper slope for the f-electrons.

Panel e translates these energy shifts into momentum space. States near the Γ point—dominated by c-character—shift only weakly with filling, whereas most of the states in the rest of the band—dominated by f-character—shift strongly. Because most of the flat-band states are f-like, the Fermi energy $E_F$ (dashed line) remains effectively pinned to the flatter regions of the band as $\nu$ varies.

Our experiment, however, measures energies relative to $E_F$. Referencing to $E_F$ has the effect shown schematically in panel f: the strongly shifting f-electron states remain near zero energy, whereas the weakly shifting c-electron states acquire an apparent downward slope with filling. This behaviour matches precisely the trend observed in Fig. 4 (here we focus only on the Hartree component; further interaction terms responsible for Hubbard-like band splitting and cascades appear on top of this effect in the experiment).

Finally, panel g shows how the $\nu$-dependent bands of panel e appear once referenced to $E_F$: the bands exhibit a clear stretching near Γ— exactly as observed experimentally in Fig. 3a. This stretching provides compelling evidence for the distinct wavefunction character of states near Γ, a key prediction of the topological structure of the BM bands and a foundational ingredient of the THF model.

## Toy model to understand the essence of the Dirac revivals
In Extended Data Fig. 10, we present a minimal toy model that separates the low-energy spectrum into two sectors: (1) light carriers with a Dirac-like dispersion ('c-like' electrons) and (2) heavy, nearly flat bands ('f-like' electrons). The aim is to capture, in a transparent way, the key experimental features, namely the cascade phenomenology and the Dirac revivals—without committing to microscopic details.

The components/assumption of the model are as follows:
1. We assume two decoupled electronic components, which we call 'c-like' and 'f-like'. They are not necessarily the original c-electrons and f-electrons but they do carry their two important aspects: light/ heavy, different Hartree couplings.
2. Specifically, we model the f-like electrons with a simplified quantum dot model, which assumes completely flat bands but with a phenomenological 'lifetime' broadening that gives a finite width to the energy bands. We compute the f-electron spectral function to capture the Hubbard bands filling evolution.
3. We assume that there is a finite Coulomb coupling between c-like and f-like electrons, $W$, and that $W < U$.

The f-electron quantum dot model Hamiltonian is: $H = U n_f (n_f - 1)/2$, in which $U$ is the effective charging energy and $n_f$ is the f-electron occupation. This quantum dot has eight flavours such that $0 \le n_f \le 8$. We compute the spectral function given by this Hamiltonian: $A(\omega) = Z^{-1} \sum_{n,m} \delta(\omega - \varepsilon_n + \varepsilon_m)(e^{-\beta\varepsilon_m} + e^{-\beta\varepsilon_n}) < l|c_{\alpha,f}^{\dagger}|m>|^2$, in which $Z = \mathrm{Tr}(e^{-\beta(H-\mu n)})$ is the partition function, $\beta = 1/k_B T$ and $\varepsilon_m$ is the energy of the $|m>$ eigenstates. We compute the spectral function $A(\omega, n_f)$ and then add smearing in energy to mimic the lifetime effect seeing in the measurement. From the smeared spectral function, we compute the relation $\mu_f(n_f)$.

As well as the quantum dot, we add Dirac (c-like) electrons. Let $n_f$ and $n_c$ denote the f and c fillings ($n = n_f + n_c$ is fixed externally). The interaction term can be written as $W n_f n_c = W n_f (n - n_f)$. The total energy is:

$$E_{tot}(n_f; n) = E_f(n_f) + E_c(n_c) + W n_f(n - n_f),$$

and the stationarity condition

$$\frac{\mathrm{d}E_{tot}}{\mathrm{d}n_f} = 0$$

gives a simple self-consistency equation:

$$\mu_f(n_f) - \mu_c(n_c) + W[n - 2n_f] = 0,$$

with $\mu_{f/c} \equiv \mathrm{d}E_{f/c}/\mathrm{d}n_{f/c}$. We solve this equation graphically to obtain $(n_f, n_c)$ at each total filling $n$ and then compute the corresponding spectral functions in momentum space for both sectors.

Extended Data Fig. 10a,b presents the calculated spectral functions for doping range from $n = 0$ to $n = 1.5$. Panels a and b show the f and c spectral functions in momentum space versus filling, reproducing: (1) the Hubbard-band evolution of heavy f-electrons and (2) the Dirac revival of light c-electrons. Specifically, in panel b, we track the Dirac point evolution: it first shifts downward with increasing $\nu$ owing to the Hartree potential and then when f states start to populate, it shifts upward (solid white guideline); this is precisely the 'Dirac revival' phenomenology and, within this model, we can see that it is driven by the $W$ Coulomb term. The schematic in Fig. 4 is based on this calculation.

The above model does not include c–f hybridization. To show that the Dirac revivals physics is robust and appears also in the case of strong hybridization ($\gamma/U > 1$), we solve a more realistic THF model that also includes hybridization (details in the next section). Extended Data Fig. 10c plots the calculated filling factor dependence of the spectral function. The black line traces the position of the c-electrons Dirac point, clearly demonstrating that the Dirac revivals phenomenology, observed in the experiment, appears also in the presence of hybridization.

## THF model reproducing the Dirac revivals
We consider a THF model[38] that treats $U$, the on-site f–f interaction, non-perturbatively within the Hubbard-I approximation, while treating all other interactions within the self-consistent Hartree approximation. We use the following parameters for the calculations: $\upsilon_\star = -1$ eV nm; $\upsilon'_\star = 366$ meV nm; $\gamma = -70$ meV; $M = 3.7$ meV; $U = 60$ meV; $W = 45$ meV; $Y = 45$ meV.

To calculate the Green's function of f-electrons and c-electrons within Hubbard-I, we first consider the non-hybridized f-electrons propagator:

$$G_{f,\eta}^0(\mathbf{k}, \omega) = \left( \frac{\frac{1}{2} + \frac{v_f}{N}}{\omega + \delta\mu + \frac{U}{2} + i\tau^{-1}} + \frac{\frac{1}{2} - \frac{v_f}{N}}{\omega + \delta\mu - \frac{U}{2} + i\tau^{-1}} \right) I_{2\times2}.$$

$v_f = \text{sign}(v)\lfloor|v|\rfloor$ is the integral filling of each f site relative to charge neutrality, within the non-hybridized theory, taken to be the total filling rounded towards zero. $\delta\mu = \mu_f - v \cdot U$ sets the asymmetry between the Mott-band energies. $N = 8$ is the number of electronic states (flavours × orbitals) per f site and $\mu_f = \mu - W \cdot \langle\delta n_c\rangle$ is the effective electro-chemical potential felt by the f site owing to repulsion from c-electrons, with $\delta n_c$ measured relative to charge neutrality. $\tau$ is a finite quasiparticle lifetime introduced by hand for numerical stability. We take $\tau^{-1} = 1$ meV.

The non-hybridized c-electrons propagator is given by

$$G_{c,\eta}^0(\mathbf{k}, \omega) = [\omega + \mu_c - H^{(c,\eta)}(\mathbf{k}) + i\tau^{-1}]^{-1},$$

in which $H^{(c,\eta)}(\mathbf{k})$ is the single-particle Hamiltonian of c-electrons in valley $\eta$ as given by Song and Bernevig[38] and $\mu_c = \mu - W \cdot \langle\delta n_f\rangle - V \cdot \langle\delta n_c\rangle$ is the effective electro-chemical potential felt by the c-electron, with $\delta n_f$ measured relative to charge neutrality. Finally, the Hubbard-I propagator is given by

$$\begin{bmatrix} G_{c,\eta} & G_{cf,\eta} \\ G_{fc,\eta} & G_{f,\eta} \end{bmatrix}(\mathbf{k}, \omega) = \begin{bmatrix} (G_{c,\eta}^0(\mathbf{k}, \omega))^{-1} & (H^{(fc,\eta)}(\mathbf{k}))^\dagger \\ H^{(fc,\eta)}(\mathbf{k}) & (G_{f,\eta}^0(\mathbf{k}, \omega))^{-1} \end{bmatrix}^{-1},$$

with $H^{(fc,\eta)}(\mathbf{k})$ the f−c single-particle hybridization term as given by Song and Bernevig[38].

The densities $\langle\delta n_f\rangle$ and $\langle\delta n_c\rangle$ are calculated by integrating the spectral functions over negative frequencies and subtracting their values at charge neutrality:

$$\langle\delta n_\alpha\rangle = 4 \int_{-\infty}^0 d\omega \int \frac{d^2\mathbf{k}}{A_{BZ}} \mathcal{A}_\alpha(\mathbf{k}, \omega) - n_\alpha^0.$$

The spectral function is given by

$$\mathcal{A}_\alpha(\mathbf{k}, \omega) = -\frac{1}{\pi} \text{Im} \text{Tr}[G_{\alpha,\eta}(\mathbf{k}, \omega)],$$

with $\alpha = f,c$, $n_\alpha^0$ is the density at charge neutrality and the factor 4 corresponds to spin and valley degeneracy, for which we use the fact that the state is flavour symmetric by assumption. For each value of the total density, we solve for $\langle\delta n_f\rangle$, $\langle\delta n_c\rangle$ and the chemical potential $\mu$ self-consistently.

### Comparing the energy spectrum at partially full and completely full flat bands

In Extended Data Fig. 11, we plot the spectroscopy data as in Fig. 4a and compare linecuts A and B, taken when the chemical potential lies inside and outside the flat bands, respectively (dashed lines in panel a). For partial filling of the flat bands, we observe a characteristic feature at $\Delta E = -15$ meV, as indicated by the dashed black lines in linecut A at $v = 1.5$. By contrast, when the flat bands are fully filled, we instead find a single broad peak, as shown in linecut B at $v = 4.3$.

One possible interpretation of the $\Delta E$ feature is a single-particle splitting induced by strain. However, the strain in the regions in which we perform the measurements is generally small (Methods section 'Spectroscopy of flat bands at different locations'), making it unlikely to account for the observed 15 meV splitting. Moreover, a single-particle splitting should also be present when the flat bands are fully filled. In

panel c, we attempt to fit the broad peak in linecut B with two symmetric Gaussians constrained to have the same lifetime as at partial filling (FWHM of about 10 meV, comparable with the energy resolution) and a fixed splitting of 15 meV and obtain a poor fit. This further suggests that simple single-particle strain-induced splitting is unlikely to be the origin of the $\Delta E$ feature.

### Discussions of possible ground states at integer fillings

At charge neutrality, we observed a semimetallic phase with states at Γ at the Fermi level. There are two physically distinct mechanisms to produce a semimetal: the thermally disordered Mott semimetal[27,34,35,39] and the strain-induced or spontaneously C3 symmetry-broken semimetal[29,52,53]. The two states give very similar spectra, with Hubbard-like nearly flat bands away from the Γ point and a band touching at or near Γ. There should be small differences in the spectra between the two states—for example, the Dirac points of the strain-induced semimetal are generally not precisely at Γ and do not have to occur at the same energy—but we believe that these differences may be too small for us to observe within experimental resolution. The main factor limiting the resolution is probably the tip size, limiting the resolution in momentum space to discern the fine details of the dispersion near Γ.

At non-zero integer fillings, our experiment is not directly sensitive to symmetry breaking, as the tunnelling from the QTM tip is not sensitive to the electron spin and valley and we do not have real-space sensitivity that can identify the spatial modulation of the spectrum that occurs in intervalley coherent or Kekulé spiral states. Experiments do not show any obvious signatures of symmetry breaking in the momentum-resolved electronic spectrum. In particular, there is no clear gap in the electronic spectrum at the Fermi level at any density (Fig. 3a). Our results are incompatible with a gap-opening broken-symmetry state at integer fillings at the temperature of the experiment.

Transport experiments typically see quantum oscillations emanating from $v = 2$ and $-2$. Our measurements, on the other hand, do not show any signatures of Fermi surface at these filling factors. It is important, however, to keep in mind the experimental conditions. Our measurements are performed at $T = 4$ K and zero magnetic field. By contrast, quantum oscillations are necessarily measured at finite magnetic fields and it is therefore difficult to exclude the possibility that the oscillatory features observed at finite field reflect a field-stabilized ground state that differs from the zero-field state examined here.

## Data availability

The data shown in this paper are provided with the paper. Further data that support the plots and other analysis in this work are available from the corresponding author on request.

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

**Acknowledgements** We thank E. Andrei, L. Bascones, A. Bernevig, L. Glazman, E. Khalf, P. Guinea, P. Jarillo-Herrero, M. J. Calderon, A. MacDonald, P. Ledwith, F. von Oppen, A. Stern, R. Valenti, A. Vishwanath and A. Yazdani for fruitful discussions. This work was supported by the Leona M. and Harry B. Helmsley Charitable Trust grant, the Rosa and Emilio Segre Research Award, the ERC-AdG grant (QTM, no. 101097125), the Deutsche Forschungsgemeinschaft (DFG) funded project number 277101999 – CRC 183 (C02), SNF Sinergia project number CRSII_222792/1, BSF grant (2020260) and ISF Quantum grant (1621/24). A.I. was supported by the Azrieli Fellows Program. E.B. was supported by the European Research Council (ERC) under grant HQMAT (grant agreement no. 817799) and CRC 183 of the DFG (project C02).

**Author contributions** J.X., A.I., J.B. and S.I. designed the experiment. J.X., A.I. and J.B. built the cryogenic QTM microscope, fabricated the devices with N.G. and Y.Z. and performed the experiments. J.X., A.I., J.B., Y.V., E.B. and S.I. analysed the data. K.W. and T.T. supplied the h-BN crystals. J.X., A.I., J.B. and S.I. wrote the manuscript, with input from the other authors.

**Competing interests** The authors declare no competing interests.

**Additional information**
**Correspondence and requests for materials** should be addressed to S. Ilani.

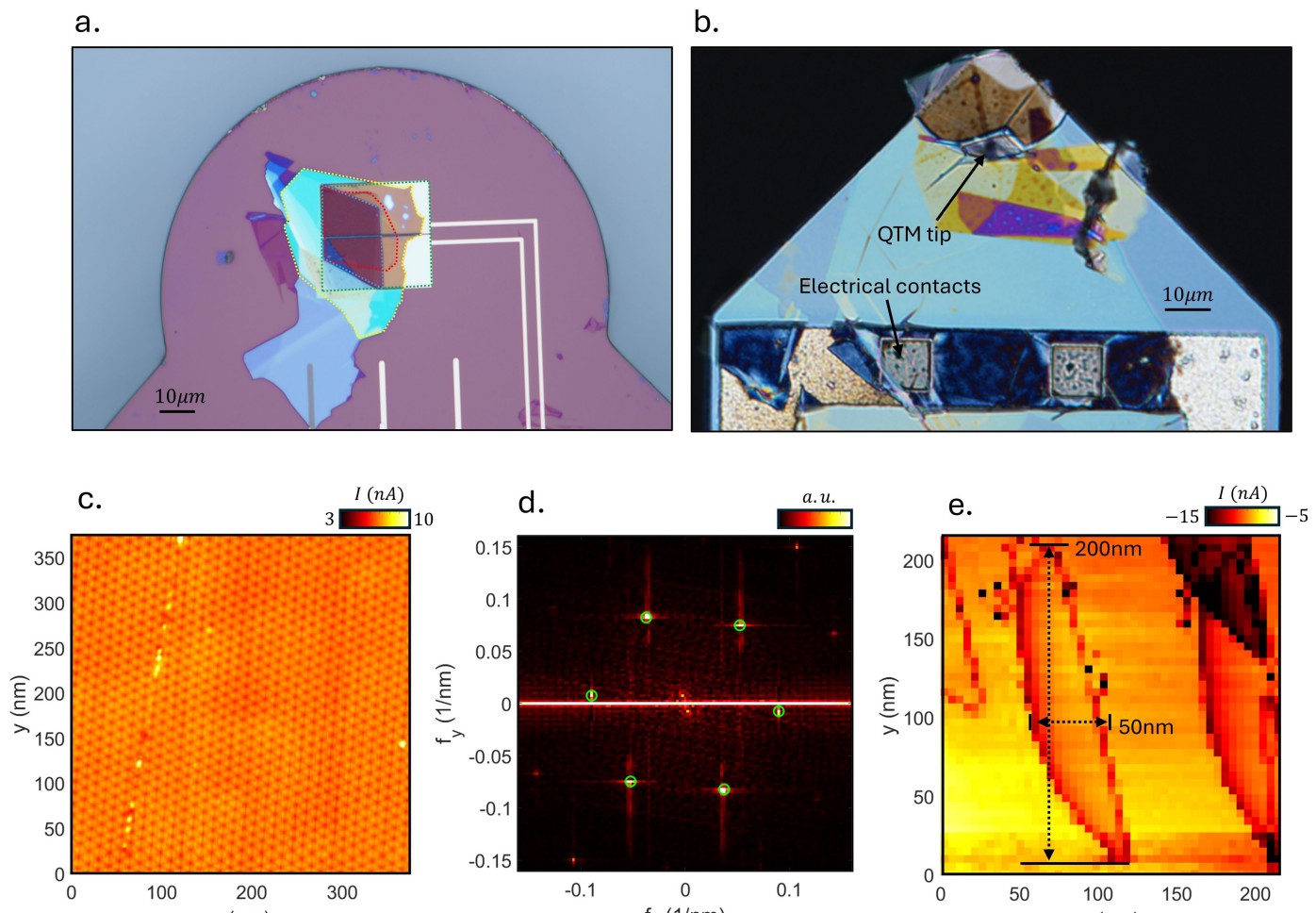

**Extended Data Fig. 1 | MATBG device and QTM-tip characterization. a**, The TBG sample used in the experiments. The stack is composed of bilayer WSe$_2$ (blue dashed)/TBG (red dashed)/37-nm-thick h-BN (yellow dashed) placed on top of a metallic back gate split into two pads (green dashed). **b**, The QTM tip used in the experiments, composed of MLG placed over h-BN and graphite layers. Electrical contacts to a conducting line on the cantilever are made with a bridging graphite flake. **c**, Room-temperature conductive AFM image of the magic-angle region in the sample. This AFM image is obtained after the thermal drift correction by using the atomic defects in WSe$_2$ (bright spots), as described in Methods section 'Fabrication and characterization of the QTM tip and van der Waals device'. **d**, FFT of this AFM image, from which we obtain the moiré lattice vectors to be 12.72 nm, 12.73 nm and 13.14 nm. It translates the twist angle and strain to be 1.10° ± 0.02° and 0.03% ± 0.02%. **e**, Tunnelling current map as a function of real-space coordinates $x$ and $y$, acquired under a large bias $V_b = -900$ mV. When the QTM tip passes over atomic defects in the WSe$_2$ tunnelling barrier, the local tunnelling current increases, thereby revealing the spatial footprint of the tip–sample contact. The elongated region of enhanced current indicates a QTM tip footprint of approximately 50 × 200 nm, as marked on the image.

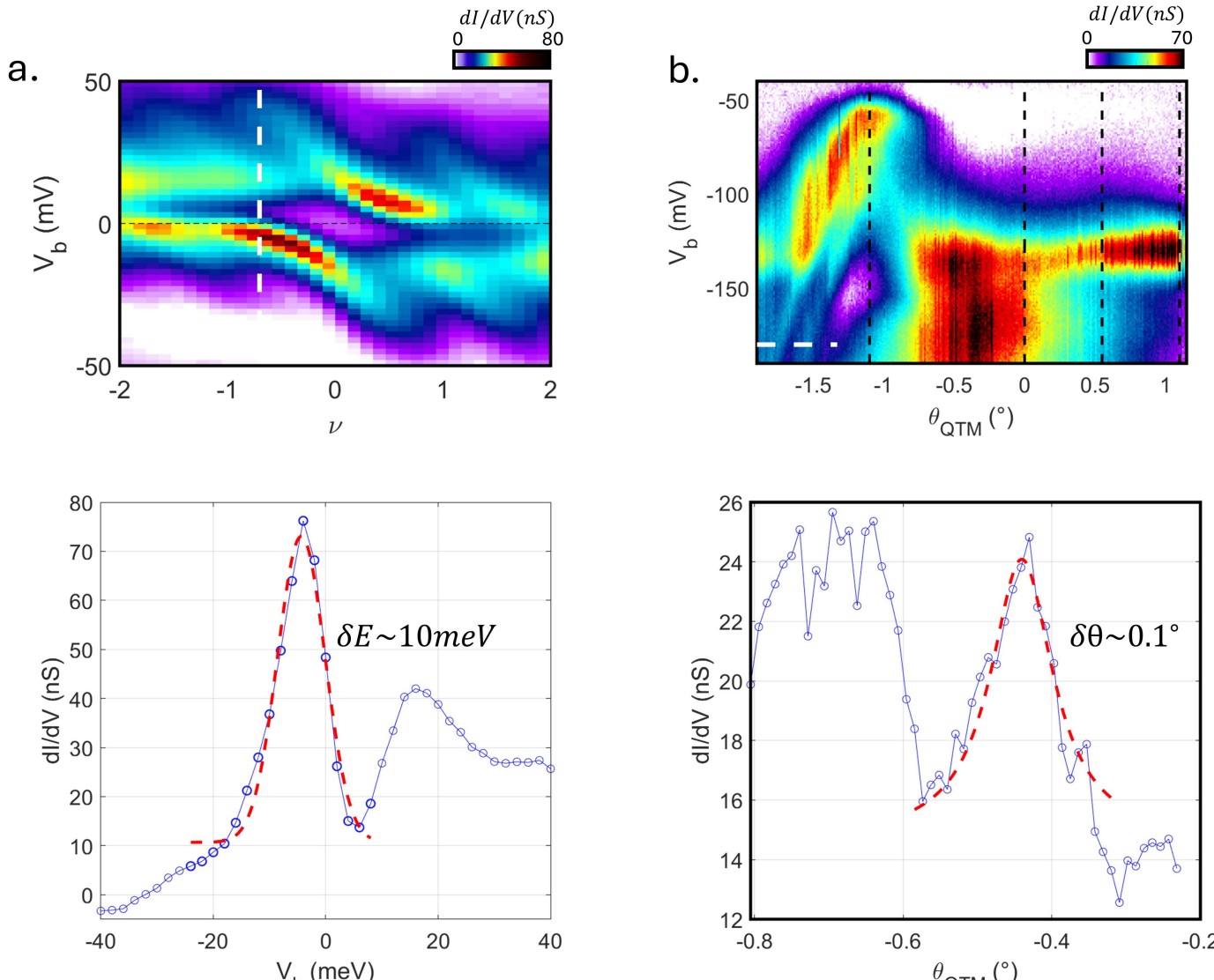

**Extended Data Fig. 2 | Energy and momentum resolution. a**, Zoom-in on the measurement of d$I$/d$V$ versus $V_b$ and $\nu$ in Fig. 4a. To estimate the energy resolution, we present in the bottom panel a linecut along the dashed white line at filling factor $\nu = -0.7$. Along this cut, we see a narrow peak centred at around $V_b \approx -4$ meV, whose width gives an upper bound on the energy resolution of our measurement. The FWHM of a Lorentzian fit to this peak (dashed red line) gives

$\delta E \approx 10$ meV. **b**, Measurement of d$I$/d$V$ versus $\theta$ and $V_b$ focusing on the remote dispersive bands at large negative energies. A linecut along the dashed white line cuts through a narrow peak as a function of $\theta_{QTM}$ (narrow in momentum). A Lorentzian fit to this peak (dashed red line) gives a FWHM of $\delta\theta_{QTM} \approx 0.1°$, corresponding to a momentum resolution of $\delta k \approx 0.03$ nm$^{-1}$.

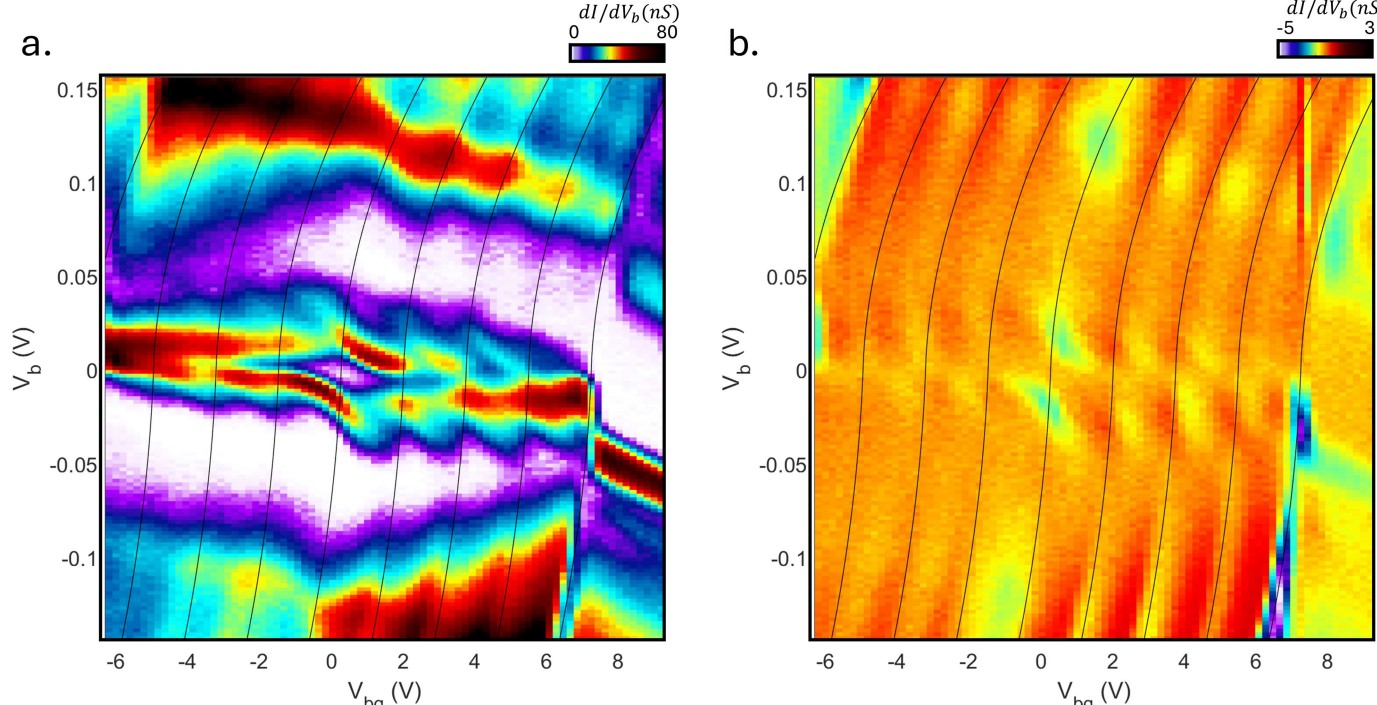

**Extended Data Fig. 3 | Correcting for the doping by the tip at finite biases.**
**a**,**b**, The spectroscopy, d$I$/d$V_b$ and the transconductance with respect to the back gate, d$I$/d$V_{bg}$, measured as a function of $V_b$ and $V_{bg}$ with $\theta_{QTM}$ within the flat-band region. At each integer filling, d$I$/d$V_b$ shows a suppression of conduction and d$I$/d$V_{bg}$ shows a dipole-like feature. This feature is more prominent at $v = \pm4$ owing to the gaps to the remote bands. We can see that the position of these features in $V_{bg}$ shift with increasing magnitude of $V_b$, making them follow an S-like trajectory. This shift reflects the fact that, because of the parallel capacitor formed by the QTM tip and TBG sample, a relative bias

between them leads to doping of the TBG. To correct this, we trace the $v = 4$ line by a polynomial fitting and overlay it at each integer filling (black lines). On the basis of this fitting line, we can skew the measurements such that their $x$-axis becomes the true filling factor (accounting for doping by the back gate as well as the tip) instead of just $V_{bg}$. The data presented in Figs. 3 and 4 have been skewed with the above method. The data in Figs. 1 and 2 are measured at a fixed back-gate voltage and can therefore have some deviations with respect to the nominal filling factor at large biases. Figure 5 is measured at zero bias and therefore it does not have any effect of doping by the tip.

## Conduction Band, fitting: $P_t \cdot P_s$

a.

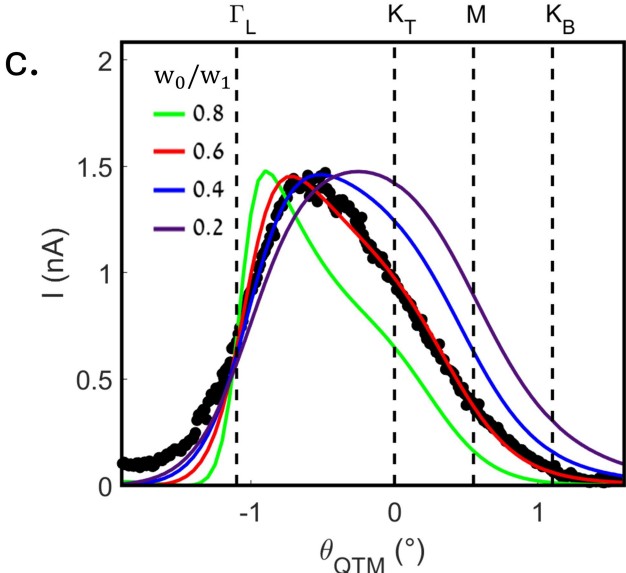

## Conduction Band, fitting: $P_t$

b.

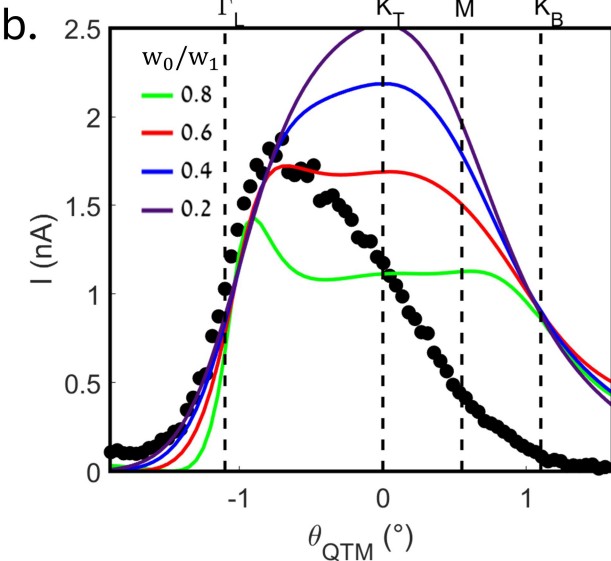

## Valence Band, fitting: $P_t \cdot P_s$

c.

## Valence Band, fitting: $P_t$

d.

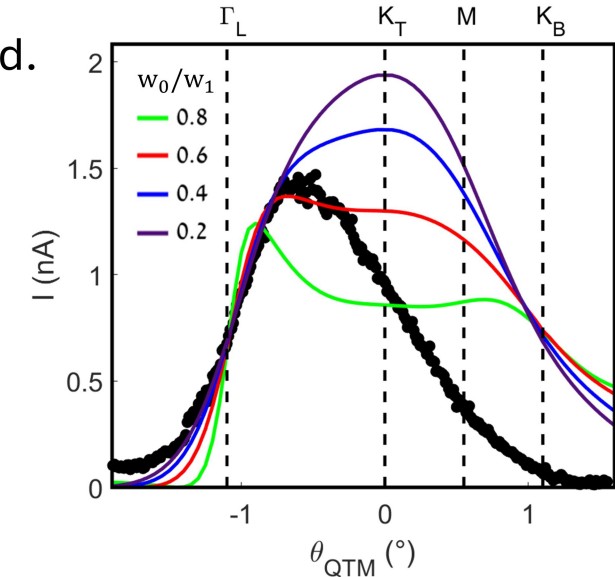

**Extended Data Fig. 4 | The tunnelling wavefunction examined by QTM and comparison with theory. a,c,** The peak intensity, $I_{peak} = \int dI/dV \cdot dV_b$, obtained by integrating the area under the $dI/dV$ peak in the conduction and valence flat band in Fig. 2a, plotted as a function of momenta (determined from $\theta_{QTM}$), in which we overlay the tunnelling matrix elements $P_k = P_t \cdot P_s$ for different $w_0/w_1$ ratios at the top. $w_0/w_1 = 0.6$ gives the best fit, consistent for both conduction and valence band sides. **b,d,** The integrated $I_{peak}$ on the conduction and valence band sides, overlaid with the naive layer polarization $P_t$, which is inconsistent for different $w_0/w_1$ ratios.

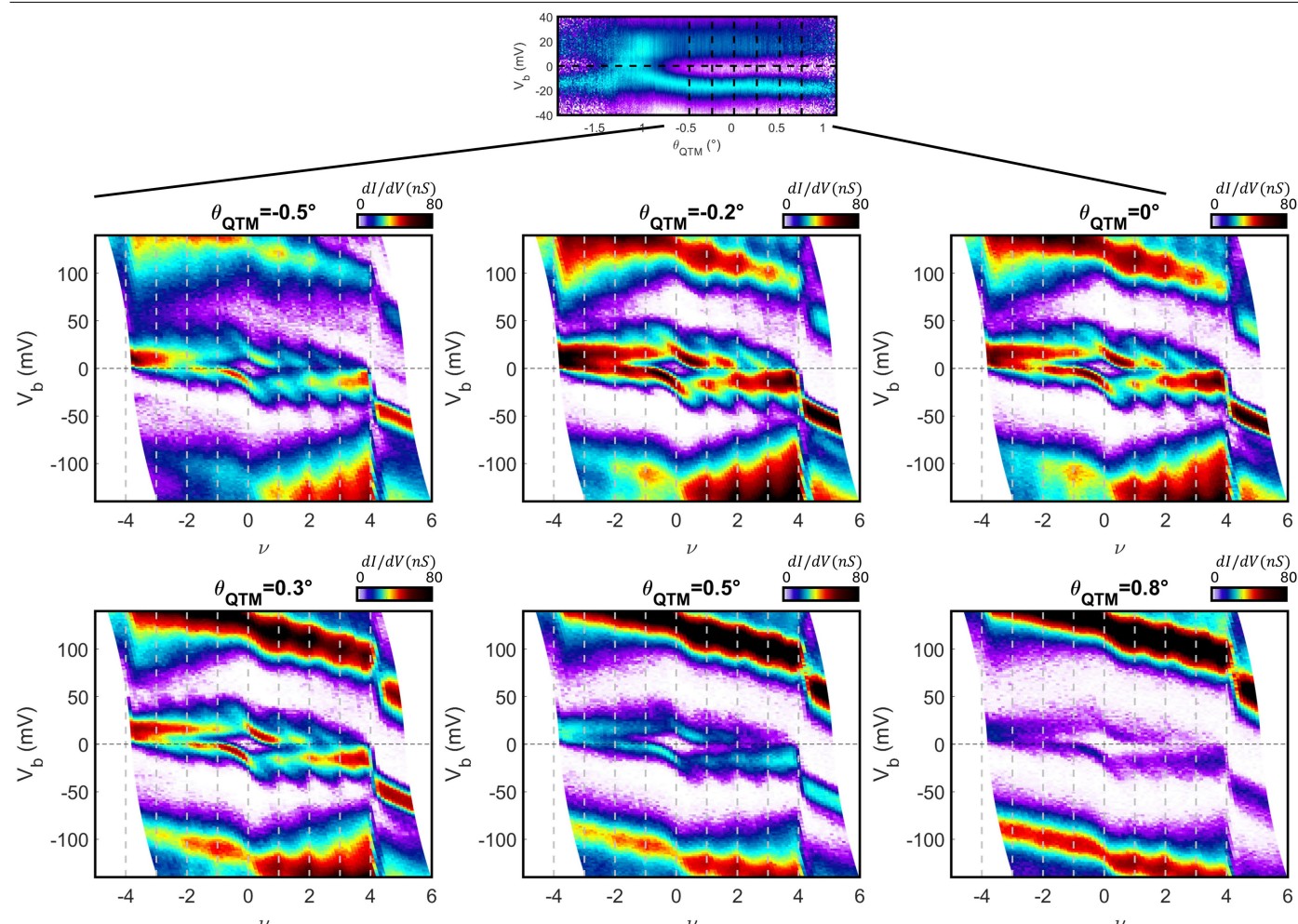

**Extended Data Fig. 5 | Spectroscopy at different momenta along the flat-band region.** d$I$/d$V$ spectroscopy versus voltage bias $V_b$ and filling factor $\nu$ at different $\theta_{QTM}$ in the flat-band region (marked by the dashed black line in the top panel).

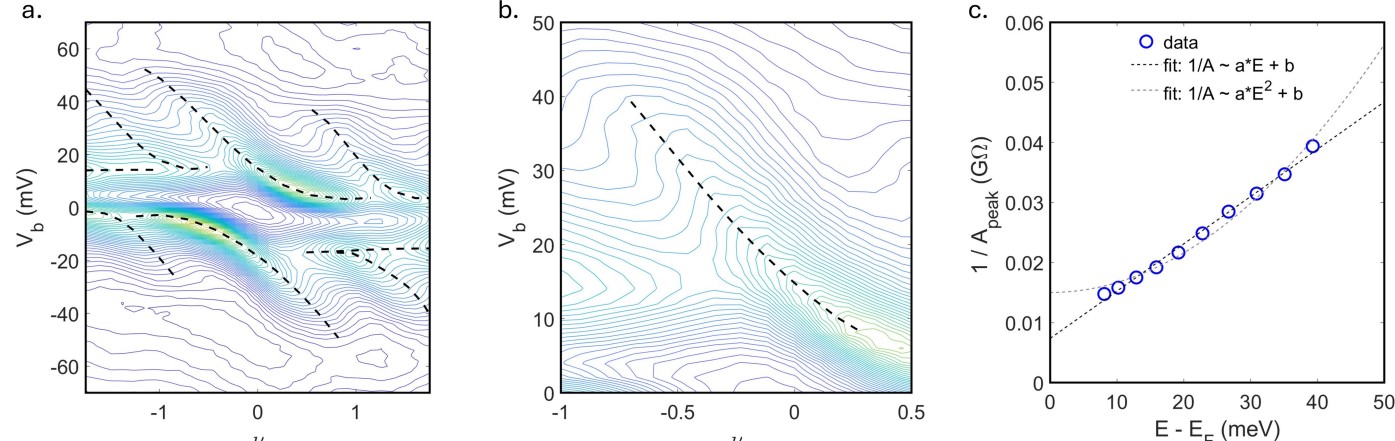

**Extended Data Fig. 6 | Gaussian peak fitting and spectral peak height versus quasiparticle energy. a**, Here we plot the contour representation of Fig. 4a. Peak centres are extracted by following local maxima in the contour lines. The resulting trajectories track both the Hubbard-like bands and excitations at energies of about ±15 meV, which are independent of doping, are marked by dashed black curves and used as initial guesses for the Gaussian fits in Fig. 3c. **b**, The zoom-in contour plot of the upper Hubbard band between $\nu = -1$ and 1, which is traced by the dashed black line that follows the local maximum of the spectral peak. **c**, Along this trajectory, we extract the spectral amplitude $A_{peak}$ as a function of quasiparticle energy $E - E_F$ and plot $1/A_{peak}$ versus $E - E_F$. The data are compared with two functional forms, $1/A_{peak} \approx a(E - E_F) + b$ and $1/A_{peak} \approx a(E - E_F)^2 + b$. Linear dependence provides a noticeably better fit.

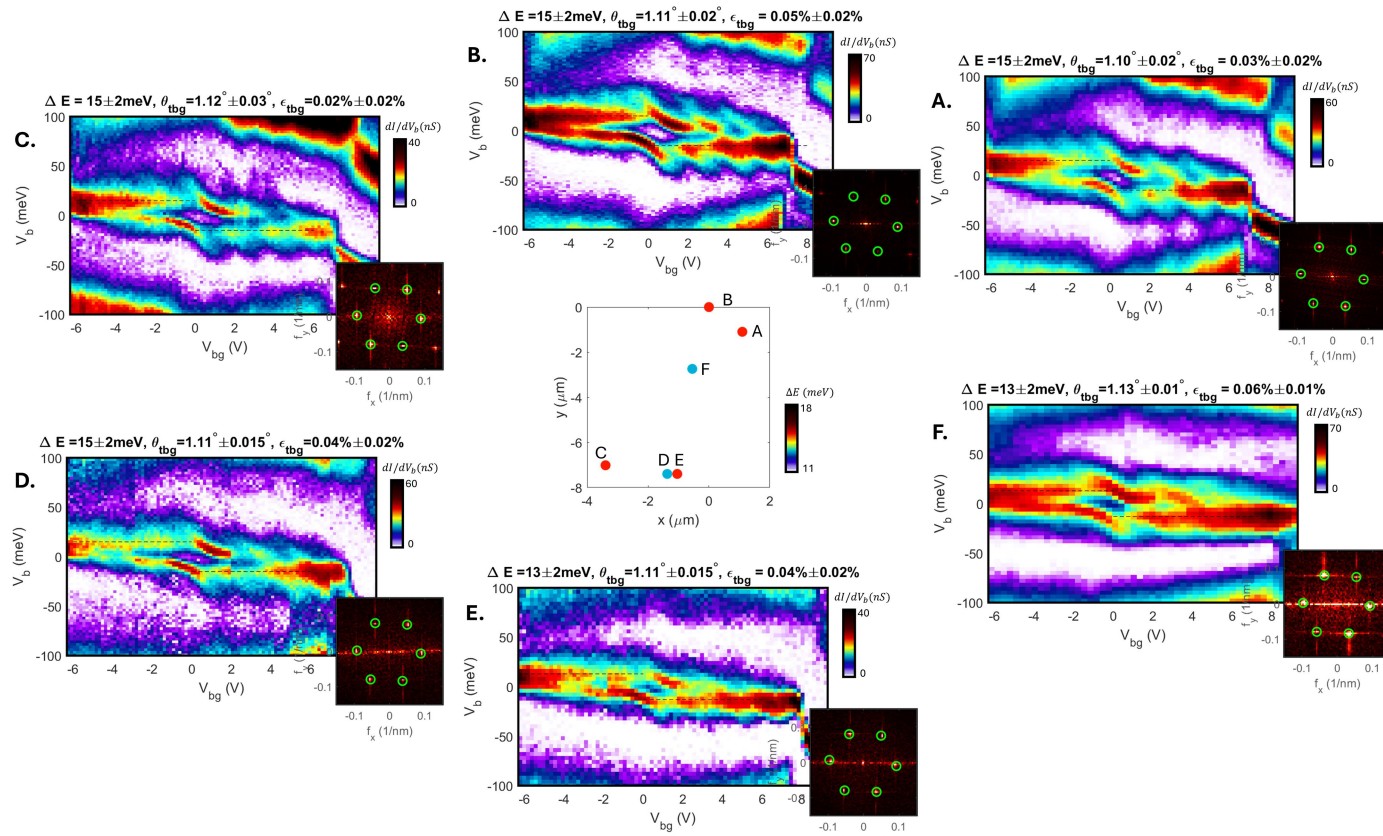

**Extended Data Fig. 7 | Spectroscopy of the flat bands at far-separated locations along the sample.** d$I$/d$V$ spectroscopy versus voltage bias $V_b$ and filling factor $\nu$, measured at the $K_T$ momentum (flat-band region) at six widely separated locations across the sample. For each panel, the measurement position, the energy scale of the filling-dependent excitation $\Delta E$, the local twist angle $\theta_{tbg}$ and heterostrain $\varepsilon_{tbg}$ are indicated at the top. The last two quantities are extracted from FFT analysis of room-temperature conductive AFM measurements of the moiré lattice at the corresponding location, shown in the insets (bottom right of each panel). The energy $\Delta E$ is marked by a dashed black line in each spectrum. The central panel shows a real-space map of all measurement locations; the colour of each point corresponds to the locally extracted $\Delta E$ value (colour scale shown on the right).

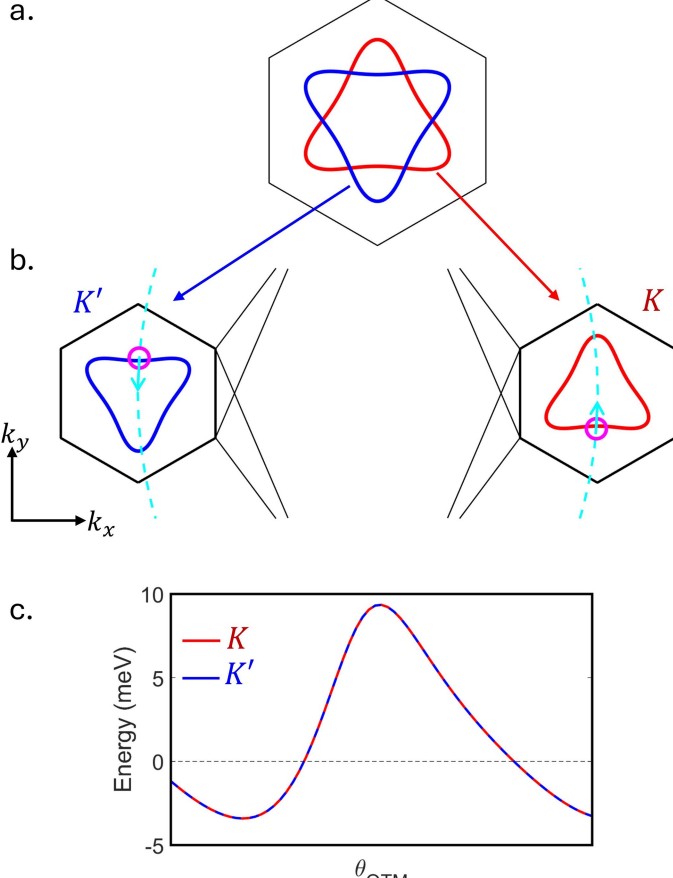

**a.**

**b.**

$K'$ $K$

$k_y$

$k_x$

**c.**

**Extended Data Fig. 8 | How a time-reversal-symmetric QTM tip examines inequivalent K and K′ valleys. a**, Schematic of inequivalent TBG Fermi surfaces in the K valley (red) and K′ valley (blue) folded into the same mini-BZ. **b**, View in the full BZ, in which the two mini-BZs appear at opposite corners and are related by time-reversal symmetry. Also shown are the graphene–probe Fermi surfaces (purple) and the circular trajectory traced by the probe Dirac points as the tip is rotated (cyan). **c**, Expected QTM signal versus $\theta_{QTM}$. When both the probe and the TBG sample preserve time-reversal symmetry, the K and K′ valleys yield identical spectral functions; the measured spectral function therefore represents their superposed contribution.

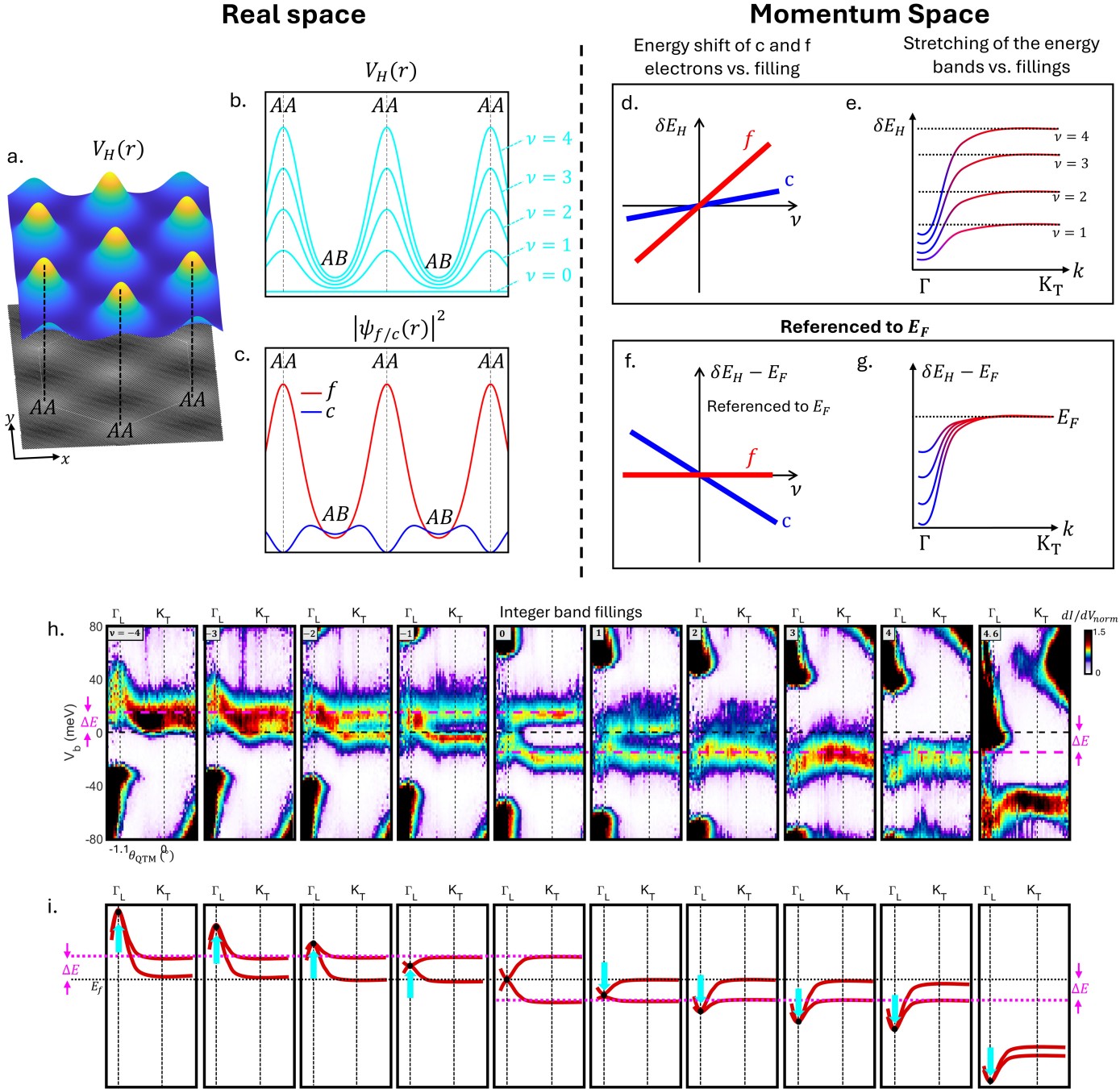

**Real space**

**Momentum Space**

**Extended Data Fig. 9 | Illustration of how filling-dependent Hartree band stretching arises from the real-space structure of the wavefunctions at different momenta. a**, Schematic real-space Hartree potential $V_H(r)$ that develops on doping, with the AA stacking regions of the moiré unit cell indicated. **b**, Filling-dependent $V_H(r)$ plotted along a linecut through the centre of the moiré unit cell. The potential peaks at the AA sites; its shape remains approximately constant with filling, whereas its overall amplitude increases roughly linearly. **c**, Schematic charge densities of the c-electron and f-electron states near Γ and away from Γ, respectively. Because of their real-space structure, the f-electron states have much larger overlap with $V_H(r)$. **d**, Hartree energy shifts $\delta E_H$ for the c-electron and f-electron states as a function of filling. Both

shift roughly linearly but the f-electron states shift more strongly owing to their larger spatial overlap with $V_H(r)$. **e**, Resulting momentum-space band evolution with filling. Dashed lines denote the Fermi energy at different fillings. **f,g**, Same as **d,e**, respectively but plotted relative to the Fermi energy. **h**, Experimental dispersions reproduced from Fig. 3a, showing the evolution of the flat-band structure as a function of filling. **i**, Schematic guides to the eye corresponding to the data in **h**, with arrows indicating the progressive stretching of the bands near the Γ point—from a pronounced peak at $v = -4$ to a clear dip at $v = 4$. This evolution reflects the momentum-dependent Hartree shifts discussed in the main text.

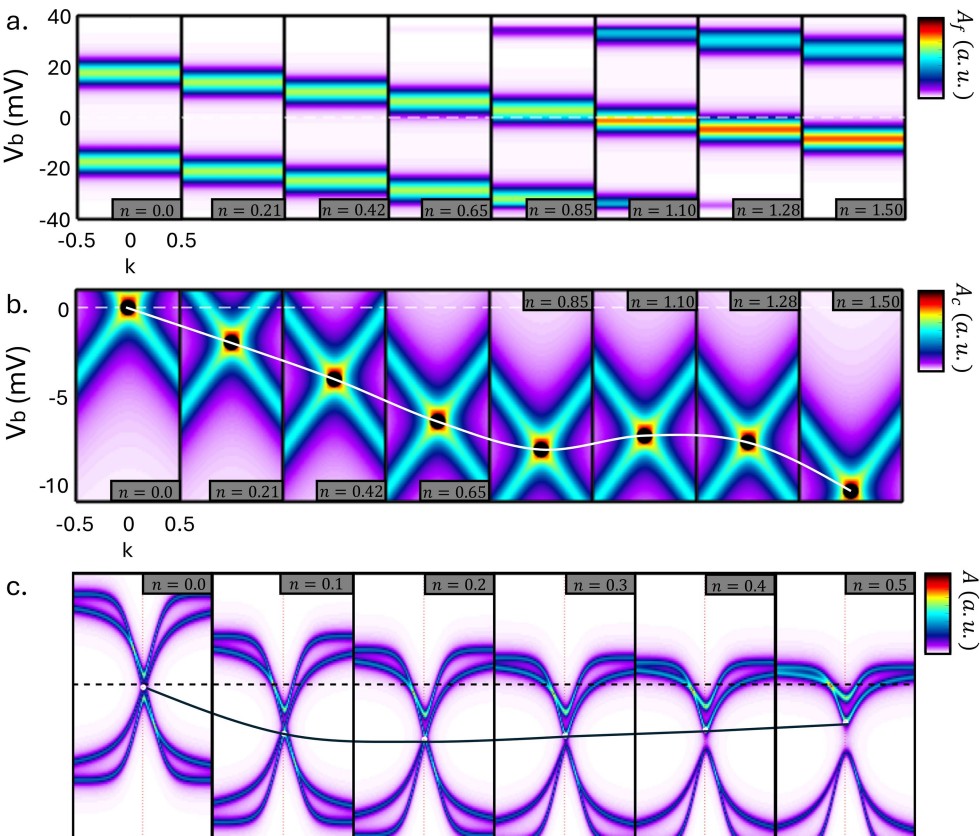

**Extended Data Fig. 10 | Spectral functions calculated with a toy model and a THF model with strong c–f hybridization, capturing the Dirac revival phenomenology.** The toy model includes a heavy-electrons sector, modelled by the energy spectrum of a quantum dot with phenomenological lifetime, a light-electrons sector, represented by Dirac dispersion, and a density–density repulsion between heavy and light carriers. For simplicity, the model neglects the hybridization between the two sectors. **a**, Spectral function of the flat (Hubbard-like) bands of heavy carriers, for total filling of $n_{tot} = 0$ to 1.5. At charge neutrality, the Hubbard bands are separated by the effective charging energy in the model, $U$. With electron doping, one band is pushed towards Fermi level and another band is pushed away from it and gradually fades.

**b**, Spectral function of the light carriers as a function of filling. The Dirac point, traced by the solid white line, shows that the Dirac electrons are initially populated and then slightly depopulated as one Hubbard band crosses the Fermi level. Once a Hubbard band is filled, the Dirac electrons continue to fill, demonstrating the reshuffling of charge between the light and heavy sectors that is at the heart of the observed Dirac revivals. **c**, Filling factor dependence of the spectral function calculated using a THF model that includes strong c–f hybridization. The black line traces the position of the c-electrons Dirac point, clearly demonstrating that the Dirac revivals phenomenology observed in the experiment also appears in the presence of hybridization.

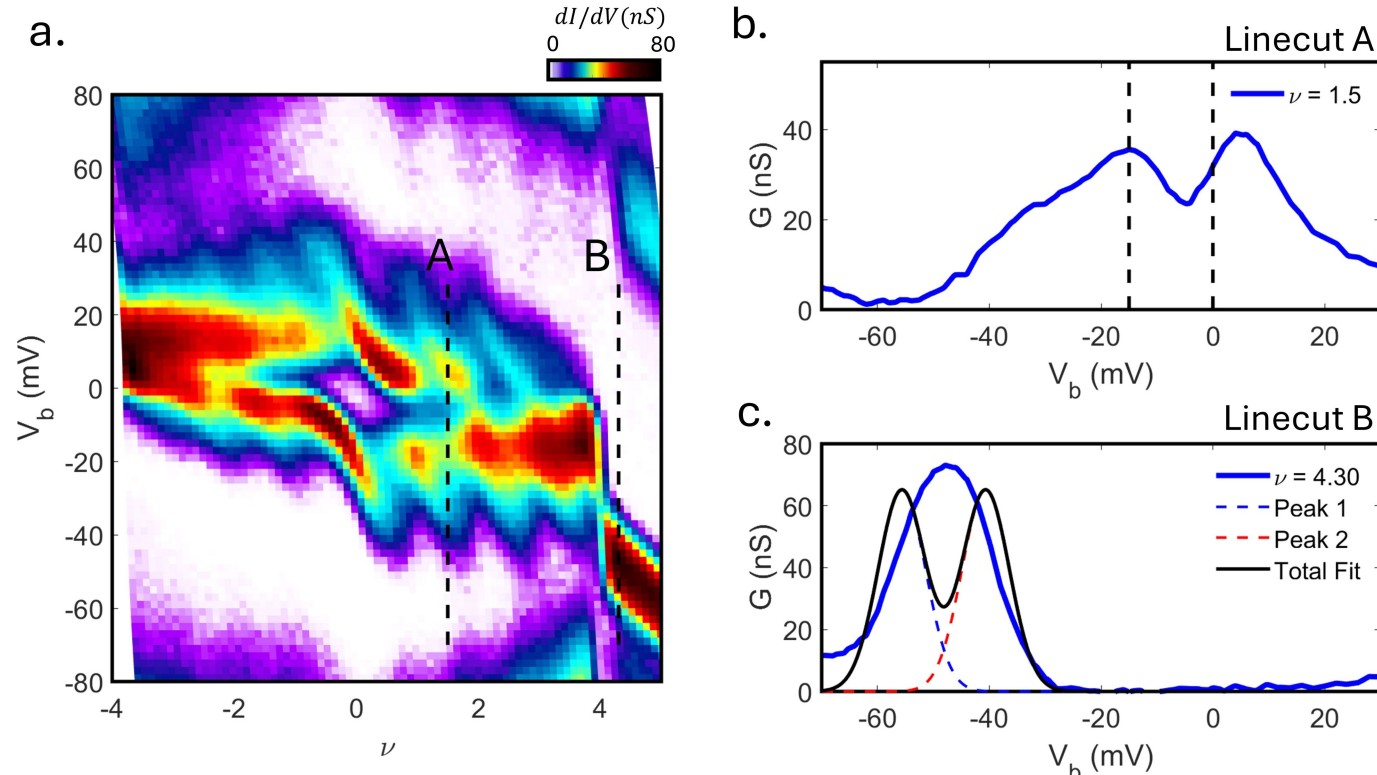

**Extended Data Fig. 11 | Spectral feature ΔE = 15 meV inside and outside the flat bands. a**, d$I$/d$V$ spectroscopy at the flat-band region (same dataset as Fig. 4a). **b,c**, Linecuts A and B at filling factors $\nu$ = 1.5 and 4.3. When the chemical potential lies inside the flat bands (linecut A), we see the ΔE = −15 meV feature, marked by the dashed black line relative to the Fermi energy. When the chemical potential lies outside the flat bands (linecut B), the spectrum shows a single broad peak. In panel **c**, we fit linecut B using a decomposition into two symmetric Gaussian peaks (labelled as peaks 1 and 2). We constrain the linewidth (FWHM ≈ 10 meV, roughly energy resolution) and the peak splitting to 15 meV. Under these constraints, we get a poor fit to the data, suggesting that a single particle splitting is less likely the origin of the ΔE = −15 meV feature.