## [Peer Review file · Nature]

Quantum Twisting Microscopy of Interacting Flat Bands in Magic-Angle Graphene

Corresponding Author: Professor Shahal Ilani

Version 0:

Reviewer comments:

Referee #1

(Remarks to the Author)

Xiao et al. report the visualization of interacting energy bands in magic-angle twisted bilayer graphene (TBG) using the quantum twisting microscope (QTM) technique developed by their group. The flat-band electrons and their interactions are central to the correlated phenomena in TBG, and accessing their momentum-resolved structure provides valuable insight. The results presented are technically solid and scientifically interesting, particularly in light of the high energy and momentum resolution achieved. I would recommend publication after the authors address the following questions.

1. I was curious about the possible effects of the tunneling barrier materials used in QTM. Have the authors tried alternative barrier materials for the measurements? Specifically, how might the choice of WSe_2 as the tunneling barrier influence the intrinsic electronic behavior of TBG?
2. What is the spatial resolution of the measurement? Is it limited to ~ 200 nm due to the size of the probe tip?
3. With the achieved high momentum resolution, can the authors resolve any differences between the inequivalent K and K' valleys in the pristine TBG and during doping/filling?
4. What is the origin of the four bands resolved in Figure 3c? Are they related to a specific symmetry breaking or interaction effect?
5. The authors report a 15 meV excitation feature. Could they provide a possible explanation or hypothesis regarding its origin?
6. According to the principle of QTM, the measured spectrum arises from a specific momentum arc. How can the authors rule out the possibility that the observed spectral weight from the c- and f-electrons originates from other momenta?
7. Regarding the relative fraction and spatial distribution of the c- and f-electrons, is it possible to provide direct real-space evidence through complementary STM measurements?

Minor suggestion: From Figure 2 to Figure 4, part of labels and guiding lines are difficult to recognize.

Referee #2

(Remarks to the Author)

Xiao et al. report energy- and momentum-resolved spectroscopy of the interacting electronic band structure of magic angle twisted bilayer graphene (MATBG) using a quantum twisting microscope. They observe contrasting electronic behavior at different momenta. Near the mini Brillouin Zone Gamma point, the bands are gapless and dispersive, leading to light carriers (c-electrons), while the bands are very flat and exhibit repeating cascades elsewhere, consistent with heavy carriers (f-electrons). The results are interpreted as support for the topological heavy fermion and Mott semimetal models of MATBG. In addition, the measurements provide information about the tunneling ratio w_0/w_1 , which has been poorly constrained, and reveal a 15 meV gap which is not understood. The work is extremely impressive, and it provides important new insight into the electronic structure of MATBG - it goes beyond traditional scanning tunneling spectroscopy by providing momentum resolution, and it has better energy and spatial resolution than nanoARPES. I therefore recommend eventual publication in Nature. However, I do have some technical comments and questions that the authors should address.

1. One set of questions is related to the energy resolution and the broadening of the bands in MATBG. The authors report ~ 10 meV energy resolution on the basis of a fit to a feature near the Fermi level. However, many spectral features further away appear much more broadened. Is this consistent with expectations from lifetime broadening when tunneling far from the Fermi level, and what lifetime would this imply for the MATBG states?

Addressing this could help to quantify the extent to which bands are broadened by interaction effects, if at all. This question is also relevant to some of the conclusions in the manuscript. For example, the manuscript notes that a diffuse “plume” reflecting an incoherent band with short lifetime. It is later stated that this band becomes stronger and sharper as it gradually shifts with filling to lower energies toward the Fermi level. Can this behavior be accounted for by lifetime effects?

Lastly, does the word ‘incoherent’ have a technical meaning? If so, this should be defined. If not, it should be in quotes or that should be stated explicitly.

2. There is a paragraph which describes a pronounced “stretching” around the Gamma point and states that it shifts in energy. I do not understand these comments nor what they refer to. Do the authors mean that there is a peak near Gamma at $\nu = -4$ in Fig. 3a, left panel? And a similar dip at $\nu = 4$ in the second to the right panel? Is this a signature of changes in band shape as carrier are populated, e.g. as proposed in Ref. 53? Do the data distinguish between such an explanation and more recent theory such as the topological heavy fermion model?

3. In Fig. 3c, there are several Gaussian fits to the data of different colors, but not all datasets are fit with the same number of Gaussians. How is it determined how many Gaussians to fit, and what is the explanation for all the different spectral features and the filling-dependence on the number of features? Do the authors have an understanding of the apparent changes in spectral weight? These questions require more discussion in the manuscript.

Referee #3

(Remarks to the Author)

This work applies the technique of quantum twisting microscope, recently developed by Ilani’s group, to address one of the outstanding theoretical puzzles in the field of moire materials: the nature of the electronic carriers in twisted bilayer graphene. The results beautifully and conclusively illustrates the dual nature of the interacting bands, showing distinct behavior close to Gamma and away from Gamma. The data is of very high quality and the results are likely to have a big impact on understanding correlation in flat bands and how they relate to band topology. However, I have some concerns about the way the authors interpret some of their measurements that need to be addressed before I can recommend for publication:

1. The authors employ the terminology of “c” and “f” electrons in somewhat ambiguous ways that make certain aspects of the interpretation unclear and could be misleading to the reader. Sometimes the author refer to “f electrons” as the “localized” states, the “heavy” states, the states away from Gamma, or the states with a particular orbital character (as in the THF model). Conversely the “c” electrons are the “itinerant” states, the “light” states, the states near Gamma, or the states with a different type of orbital character. While the decoupled limit of the topological heavy fermion model provides a case where these properties can all coincide, the experiment does not appear to be in that limit. Instead, the “f” and “c” orbitals strongly hybridize at generic momenta. They could further be strongly renormalized by interactions, e.g. c-electrons could be dressed by particle hole pairs of f-electrons or vice versa. These effects do not change the overall picture of a dual character associated with light/heavy carriers close to Gamma/away from Gamma and are consistent with all experimental observation without being tied to specific microscopic orbitals. For example, Is it clear that the orbitals near Gamma have mostly c-like orbital character, just because they are light? The orbital character itself is not measured in the current experiment. Furthermore, the spectrum at the Fermi energy in Fig. 5 seems to suggest a strongly doping dependent spectral weight near the Fermi energy, especially upon electron doping $\nu = 2$. This suggests that quasiparticle dressing could be significant. Also, as the Fermi energy moves further and further away from Gamma, it becomes increasingly unclear whether the mostly c-character electrons are expanding in phase space (due to mixing with remote bands?) or the mostly f-character electrons are gaining more dispersion. Thus, more care should be taken regarding what is meant by “f” and “c” because the various defining characteristics given do not generally coincide. I suggest the authors adopt a more agnostic language when referring to the observation to avoid misleading the reader, and possibly discuss the caveats associated with any given theory interpretation.

2. In STM measurements, the 15 meV feature is still observed as a flat-band splitting when the chemical potential is in the remote bands. Do the authors also see the 15 meV feature when the chemical potential is in the remote bands? If the 15 meV feature is active in the remote bands it suggests it should be attributed to a single particle effect, rather than a new unaccounted for excitations as the authors propose (the remote bands are expected to be very weakly correlated).

3. I found the Dirac revival description rather confusing and I believe the zero c-f hybridization limit employed in that discussion is inapplicable to the parameters relevant to the experiment. For one, the decoupled limit contains double the number of c-electrons. It is the c-f hybridization which is responsible to sending half of the Gamma c-electrons to the remote bands while the other half forms the near-Gamma part of the flat bands. An interpretation of Fig 4 would then suggest that both remote band and flat band c-character electrons are filled equally, as the Dirac cone corresponds to a hybridization between these. This seems surprising because the remote bands are rather far in energy in practice. It is also not clear why the c-electron dispersion would move in an opposite manner due to the f-electron band filling. More f-electrons should mean a larger Hartree potential, pushing the c-electrons down even further below the f-band. They can furthermore be continuously doped at an energy U until the band is full, thereby pinning the chemical energy to this value. I thus cannot see how a downward shift would be possible in the picture the authors put forward. One can get downward shifts, in the sense the authors describe, through flavor polarization transitions as in the old Dirac revival story. But this is because there are

correlations between the doped heavy electrons, such that they wish to be collectively doped so that their flavors can align. While calculations on the decoupled THF model have shown these features, this is because they assume the f-electron occupation is the same integer at every site thereby necessitating doping every f-electron at once.

4. In addition to the 'resets' identified close to integer fillings, there seem to also be changes in the quasiparticle weight e.g. the weight at Gamma at the chemical potential gets progressively weaker as we move away from neutrality. This seems to suggest significant quasiparticle dressing inconsistent with the simple model employed by the authors. Can the authors comment on that and discuss possible caveats in their interpretation?

5. It's not clear how the extraction procedure for "s" relates to the definition in Ref. 42. Indeed, the extraction appears to depend mostly on the strength of the Hartree dispersion. This gradually makes the band more and more dispersive, such that more and more electrons do not qualify as "f"-character anymore. However, it seems to also be likely for the electrons with mostly f-like orbital character to gain more dispersion through a stronger potential magnifying a fixed, smaller, c-component. In the latter case, the orbital proportions of the total flat band does not change and "s", as defined in Ref. 42, would correspondingly also not change.

6. There are at least two distinct semimetals possible at charge neutrality. One is a symmetry broken semimetal stabilized by strain and another is a thermally disordered "Mott" semimetal. Similarly, there are three distinct kinds of insulators proposed at non-zero integer fillings: symmetry-broken intervalley coherent states, kekule spiral states, and thermally disordered insulators. Can the authors comment on whether any of these best fit the data or whether it is insufficient to make this conclusion?

Some other smaller points

- It is not clear what the authors mean when they refer to the f-electrons as "localized." At least in the beginning of the manuscript, the f-electrons referred to states with momenta away from Gamma. But states in momentum space are never localized; they are of course Bloch waves. Perhaps the authors mean that wavepackets constructed from these parts of momentum space (excluding Gamma) are tightly localized in some sense? It would be helpful if the authors specified what they mean here.

- A spin incoherent Luttinger liquid (SILL) is an example of carriers with the "dual nature" the authors refer to. The carriers at the Fermi points are both itinerant and active in transport while also carrying $\sim \log(2)$ entropy due to their spin. It would be interesting for the authors to compare and contrast their observations of dual nature with that of the SILL. Could similar highly-dressed quasiparticles, which are both itinerant and carry the entropy, be active in the TBG case?

Referee #4

(Remarks to the Author)

I co-reviewed this manuscript with one of the reviewers who provided the listed reports.

Version 1:

Reviewer comments:

Referee #1

(Remarks to the Author)

The authors have sufficiently addressed most comments from the reviewers. They have provided more experimental results and calculations to support their conclusions. I therefore support its publication on Nature.

Referee #2

(Remarks to the Author)

The authors have fully addressed my questions in their revised manuscript, and in my opinion, have also provided adequate answers to the questions from the other reviewers. I recommend this manuscript for publication in Nature in its present form.

Referee #3

(Remarks to the Author)

The authors have responded thoroughly to our questions. We recommend for publication, but would like the authors to address the comments below for further clarity.

1. While the authors present a convincing evidence that the 15meV feature is unlikely to be due to strain, it is still puzzling whether it is a single particle effect or not. In particular, a very similar 15meV splitting was observed in [Wong et al Nature volume 582, pages 198–202 (2020)]. In this case, the 15meV splitting continuously connects to a corresponding splitting in the remote bands. Do the authors believe that the splitting they observe is of a different nature, as it is not present in the remote bands? Relatedly, could the authors comment on the very large lifetime of flat band electrons while the chemical potential is in the remote bands? I would have thought that the system is mostly non-interacting once the chemical potential is in the remote bands, such that the flat bands should have a relatively sharp quasiparticle peak.

2. A widely anticipated application of the QTM to TBG is a better understanding of the superconducting normal state. Transport experiments typically see quantum oscillations emanating from $\nu=2$ and $\nu=-2$, pointed away from charge neutrality. These have been argued to suggest that superconductivity emerges from a small Fermi surface of light carriers. Fig. 5 however does not seem consistent with a Fermi surface of c-like electrons. Instead there is a small spectral weight that

is spread out across the heavy/f-like parts of the Brillouin Zone. Notably, there does not seem to be any sign of a Fermi surface peak. It would be good to comment on this tension and its implications for how we should understand the superconducting normal state.

Referee #4

(Remarks to the Author)

I co-reviewed this manuscript with one of the reviewers who provided the listed reports.

Response to referees

We first wish to thank all the referees for finding our work to be "extremely impressive", "address one of the outstanding theoretical puzzles in the field", that "the data is of very high quality", and "likely to have a big impact on understanding correlation in flat bands and how they relate to band topology", as well as finding our paper suitable for publication in *Nature*. The referees have also asked several important questions, particularly regarding the Hartree driven band stretching, its connection to momentum-dependent wave function structure, how it informs the interpretation of light and heavy carriers, and the possible origins of the $\sim 15\text{meV}$ excitation.

To fully address these points, we have added new experimental data and substantially expanded the supporting information, including ten new Extended Data figures and nine new Methods sections with detailed explanations. Below, we provide point-by-point response to all referee questions. We believe these additions significantly strengthen the manuscript, and we hope the referees will find that all their questions have been fully addressed and that the paper is now suitable for publication in *Nature*.

Referee#1

Xiao et al. report the visualization of interacting energy bands in magic-angle twisted bilayer graphene (TBG) using the quantum twisting microscope (QTM) technique developed by their group. The flat-band electrons and their interactions are central to the correlated phenomena in TBG, and accessing their momentum-resolved structure provides valuable insight. The results presented are technically solid and scientifically interesting, particularly in light of the high energy and momentum resolution achieved. I would recommend publication after the authors address the following questions.

We thank Referee #1 for their review and supportive comments. Below we address each of the questions in detail.

1. I was curious about the possible effects of the tunneling barrier materials used in QTM. Have the authors tried alternative barrier materials for the measurements? Specifically, how might the choice of WSe₂ as the tunneling barrier influence the intrinsic electronic behavior of TBG?

We thank the referee for raising this question. Our experiments were performed only with a WSe₂ barrier. We note that there is a significant number of experiments done with MATBG in contact with WSe₂ [1-4]. At T=4K (the temperature of our experiments), these studies consistently report the same electronic phenomenology as in devices encapsulated in hBN, indicating that WSe₂ does not alter the intrinsic MATBG band structure at these temperatures.

Two effects associated with WSe₂ have been reported in prior works, but both occur only at much lower energy/temperature scales than those probed here:

1. Proximity-induced spin–orbit coupling, observed only at $T \ll 4$ K and with a characteristic energy splitting below our spectral resolution [3].
2. An enhancement of the superconducting transition temperature, which appears at $T < 1$ K [1].

Given that these effects emerge only at temperatures well below 4K, they are irrelevant in our experiment.

We also note that STM studies performed without any capping layer (e.g., Refs. 8 and 20 of the manuscript) exhibit spectral features fully consistent with our measurements, providing further evidence that the presence of WSe₂ does not modify the intrinsic interacting bands we observe.

[1] H. S. Arora, R. Polski, Y. Zhang, A. Thomson, Y. Choi, H. Kim, Z. Lin, I. Z. Wilson, X. Xu, J.-H. Chu, K. Watanabe, T. Taniguchi, J. Alicea, S. Nadj-Perge, Superconductivity in metallic twisted bilayer graphene stabilized by WSe₂, *Nature* 583, 379–384 (2020).

[2] R. Polski, Y. Zhang, Y. Peng, H. S. Arora, Y. Choi, H. Kim, K. Watanabe, T. Taniguchi, G. Refael, F. von Oppen, S. Nadj-Perge, Hierarchy of Symmetry Breaking Correlated Phases in Twisted Bilayer Graphene, arXiv:2205.05225 (2022).

[3] J.-X. Lin, Y.-H. Zhang, E. Morissette, Z. Wang, S. Liu, D. Rhodes, K. Watanabe, T. Taniguchi, J. Hone, J. I. A. Li, Spin-orbit–driven ferromagnetism at half moiré filling in magic-angle twisted bilayer graphene, *Science* 375, 437–441 (2022).

[4] M. He, J. Cai, H. Zheng, E. Seewald, T. Taniguchi, K. Watanabe, J. Yan, M. Yankowitz, A. Pasupathy, W. Yao, X. Xu, Dynamically tunable moiré exciton Rydberg states in a monolayer semiconductor on twisted bilayer graphene, *Nature Materials* 23, 224–229 (2024).

[5] M. He, X. Wang, J. Cai, J. Herzog-Arbeitman, R. Peng, T. Taniguchi, K. Watanabe, A. Stern, B. A. Bernevig, M. Yankowitz, O. Vafek, X. Xu, Strongly Interacting Hofstadter States in Magic-Angle Twisted Bilayer Graphene, *Nature Physics* 21, 1380–1386 (2025).

2. What is the spatial resolution of the measurement? Is it limited to ~200 nm due to the size of the probe tip?

Indeed, the real-space resolution of these measurements is set by the size of the QTM tip footprint (/contact area). We determine this footprint size *in-situ* by using an atomic defect in the WSe₂ barrier as a localized tunnelling channel. At large bias ($V_b \approx -900$ mV, well outside the range used in the actual spectroscopy measurements), the defect level becomes energetically accessible and provides an additional, spatially localized current path. When the QTM tip passes over the defect, we observe a small but measurable increase in current. By scanning the tip across such defects and mapping this current enhancement, we obtain a real-space image of the tip’s contact footprint.

The newly added **Extended Data Fig. 9** (shown below) presents such a high-bias scan, showing multiple images of the tip footprint, each produced by a different defect, revealing a contact size of approximately $200 \text{ nm} \times 50 \text{ nm}$.

These real-space dimensions set the ultimate momentum-space resolution through the Heisenberg uncertainty relation. The measured footprint corresponds to angular momentum resolutions (in angular units) of roughly $\delta\theta_{QTM} \sim 0.05^\circ$ and $\sim 0.2^\circ$ along the two principal directions.

In Extended Data Fig. 2 we show that the sharpest momentum features – obtained in the remote bands - exhibit FWHM of $\delta\theta_{QTM} \sim 0.1^\circ$, consistent with the resolution expected from the measured tip dimensions.

Extended Data Figure 9: In-situ imaging of the QTM tip contact footprint.

Changes to the manuscript: We added new Extended Data Fig. 9 and the corresponding discussion in Methods M11.

3. With the achieved high momentum resolution, can the authors resolve any differences between the inequivalent K and K' valleys in the pristine TBG and during doping/filling?

As illustrated in the figure below, the time-reversal symmetry of the QTM probe dictates that it measures the *combined* contribution of the K and K' valleys.

Panel **a** shows the inequivalent Fermi surfaces of the K and K' valleys (red and blue) folded into the same mini-BZ. For illustration, we depict triangularly warped Fermi surfaces to emphasize that the two valleys can in principle be different.

To understand what the QTM actually probes, it is useful to unfold the picture back to the full Brillouin zone (panel **b**). There, the K and K' Fermi surfaces sit at opposite corners. In the experiment, the Dirac points of the probe (purple) move along a circular trajectory (dashed line), and current is detected whenever the probe's Dirac point intersects a Fermi surface. Crucially, if both the probe and the sample obey time-reversal symmetry, the K and K' valleys produce *identical* spectra as a function of the QTM angle θ_{QTM} (panel **c**). If we take the illustration in panel b as an example, although the two triangular Fermi surfaces are mirror reflections, the rotating Dirac points intersect them in exactly the same sequence—first the flat edge, then the tips—and therefore yield identical spectroscopic measurement.

One might expect that spontaneous valley symmetry breaking (e.g., a valley-polarized Chern insulator) would manifest as a clear energy splitting between the K and K' bands and that consequently, it should be easy to identify flavour symmetry breaking from this band splitting. However, this expectation is incorrect – recent DFMT calculations performed in both the symmetric and flavour-symmetry-broken states show that very similar band splittings appear in both situations. These splittings originate primarily from the formation of Hubbard bands, which occur regardless of whether valley symmetry is broken. As a result, the spectroscopic signatures of flavour symmetry breaking are subtle and cannot be identified simply by looking for energy-split bands.

Redacted

Extended Data Figure 10. How a time-reversal-symmetric QTM tip probes inequivalent K and K' valleys.

Changes to the manuscript: We added new Extended Data Fig. 10 and the corresponding discussion in Methods M12.

4. What is the origin of the four bands resolved in Figure 3c? Are they related to a specific symmetry breaking or interaction effect?

The multiple features observed in Fig. 3c arise naturally from interaction effects in the charge sector and do not require any breaking of flavour (spin/valley) symmetry. They reflect the emergence of localized heavy-electron orbitals and the corresponding formation of their Hubbard bands.

The red / blue peaks correspond to the lower and upper Hubbard bands, which move toward or away from the Fermi energy as the filling is varied. The dark red/blue peaks are higher Hubbard bands that for a short range of filling-factors can be seen together with the main Hubbard bands. Qualitatively similar bands and filling-factor dependence is seen DMFT calculations (Ref. 37 and 38), which do not break flavour symmetry. DMFT calculations done with and without flavour symmetry breaking show that the effect of the latter on the bands is rather subtle (opening of small gaps) but the overall multi (Hubbard) band structure is a more generic feature coming from the charge sector and is associated with the emergence of (quasi-) localized f-like electrons.

The grey peaks at ~ 15 meV correspond to the additional "mystery" excitation that we highlighted in the manuscript. This feature is not captured by current DMFT models and remains an open question.

5. The authors report a 15 meV excitation feature. Could they provide a possible explanation or hypothesis regarding its origin?

The physical origin of the 15 meV excitation remains an open question. Several theory groups are currently exploring possible explanations, but at present there is still no compelling theoretical mechanism. The initial hypothesis, shared by several groups that are trying to interpret our data - is that this feature may arise from strain in MATBG. While we cannot definitively rule out strain, we believe that we have accumulated experimental evidence that makes this explanation unlikely.

To present this important information, we have now added a new Extended Data section (Methods 10 and Extended Data Fig. 8) to the revised manuscript, including new measurements, not present in the original submission. This section summarizes the evidence that strain alone is unlikely to account for the observed 15 meV feature and is intended to help guide future efforts to resolve this puzzle, which we suspect may have broader implications for understanding MATBG.

Evidence that strain is unlikely to be the origin of the 15 meV excitation:

1. Universality across widely separated regions of the same device.

We observe the same excitation energy at multiple locations within the sample, separated by several microns - distances large enough that these regions effectively behave as independent devices. We have added new Extended Data Fig. 8 to the revised manuscript (reproduced below), showing the spectroscopy measured at different locations (individual panels) together with a map of the measurement locations (center panel). and the corresponding spectra.

2. Universality across different samples.

We detect similar excitations (13–15 meV) in two distinct MATBG devices, measured in separate cooldowns, indicating that the feature is not sample-specific or related to a particular thermal cycle.

3. No correlation with independently measured strain.

We have used room-temperature conductive-AFM to image the moiré lattice in each of the measured locations (small insets in Extended Data Fig. 8). From these images we extract the hetero-strain, which varies by a factor of ~ 2.5 across the measured positions (from $\varepsilon = 0.02\%$ to $\varepsilon = 0.06\%$). Despite this substantial variation, the excitation energy remains essentially constant ($\approx 13\text{--}15$ meV).

4. Quantitative inconsistency with theoretical expectations.

For the least-strained region ($\varepsilon = 0.02\%$), the observed excitation energy is ~ 15 meV. Existing theoretical models predict a strain-induced level splitting of ~ 10 meV per 0.1% strain [1], which would imply a splitting of only ~ 2 meV for $\varepsilon = 0.02\%$ - an order of magnitude smaller than what we measure.

[1] J. Herzog-Arbeitman, J. Yu, D. Călugăru, H. Hu, N. Regnault, O. Vafek, J. Kang, B. A. Bernevig, Topological Heavy Fermion Model as an Efficient Representation of Atomistic Strain and Relaxation in Twisted Bilayer Graphene, Physical Review B 112, 125128 (2025).

Extended Data Figure 8. Spectroscopy of the flat bands at far-separated locations along the sample.

Changes to the manuscript: We added new Extended Data Fig. 8 and the corresponding discussions in Methods M10 to the revised manuscript.

6. According to the principle of QTM, the measured spectrum arises from a specific momentum arc. How can the authors rule out the possibility that the observed spectral weight from the c- and f-electrons originates from other momenta?

Two mechanisms could, in principle, lead to spectral weight arising from momenta other than those selected by the QTM trajectory. We address both below:

1. Non-momentum-conserving tunneling

In QTM, the tunneling current is overwhelmingly dominated by momentum-conserving processes. In our earlier work (ref. [48] in main text), we quantified this explicitly and showed that non-momentum-conserving channels contribute a current that is six orders of magnitude smaller than the momentum-conserving one (see Fig. 1c of Ref. [48]). This extreme selectivity ensures that any contribution from non-conserving tunneling is negligible in the present experiment.

2. Distortions of the QTM trajectory due to strain

Strain in either the sample or the tip could, in principle, distort the QTM trajectory so that the arc traced by the probe's Dirac point no longer passes precisely through the MATBG $\Gamma - K_B - K_T - \Gamma$ path. We can rule out both possibilities in the current experiment:

(a) Sample strain: As shown above, we directly measured the local strain at multiple sample locations using room-temperature conductive-AFM imaging of the moiré lattice. In all cases the strain was found to be extremely small, $\varepsilon = 0.06\% - 0.02\%$, far below the threshold required to shift the Γ or K points by more than our measured momentum resolution. Thus, sample strain cannot significantly distort the QTM trajectory.

(b) Tip strain: Strain in the QTM tip could shift the probe's Dirac point such that the QTM arc *misses* the true Γ or K points of the MATBG mini-BZ. This would manifest as an **artificial gap** even if a system is gapless.

Redacted

This indicates that the tip strain is negligible and that these measurements cut cleanly through both the K and Γ points.

7. Regarding the relative fraction and spatial distribution of the c- and f-electrons, is it possible to provide direct real-space evidence through complementary STM measurements?

Our experimental setup does not allow us to perform STM measurements, and therefore we cannot map the wavefunctions directly in real space. However, our experiment *does* provide key information about the real-space structure of the wavefunctions at different momenta through the **momentum-resolved Hartree energy shifts**. In retrospect, we realize that we did not explain this point clearly enough in the original manuscript. In the revised manuscript we have added new Extended Data Figure. 11 and 12 and new methods section M13 with detailed, step-by-step explanation of how the observed filling-dependent stretching of the bands encodes information about the wavefunctions at different momenta.

For completeness, we provide a version of this explanation below:

The experimental observation that provides information about the nature of the wavefunctions is the filling-factor-dependent stretching of the energy bands, as seen in

Fig. 3a in the main text. By stretching we mean that the bands have a peak at Γ for $\nu = -4$, which continuously evolves into a dip at Γ at $\nu = 4$. To highlight this evolution better, we have added a new Extended Data Fig. 11 that shows the measured bands alongside schematic guides to the eye and uses arrows to highlight the stretching.

Extended Data Figure 11. Highlighting the filling-dependent stretching of the bands.

This stretching originates from the momentum-dependent response of the electronic wavefunctions to the Hartree potential that builds up with increasing carrier density. Early theoretical works (e.g., Refs. [23], [24]) already noted that while moiré-scale electrostatic potentials are negligible at charge neutrality, a spatially varying Hartree potential $V_H(\mathbf{r})$ develops as electrons or holes are added. The energy of a state at momentum k then shifts according to its real-space overlap with this potential,

$$\delta E_H(k) = \int d^2r V_H(r) |\psi_k(r)|^2$$

because different k -states have distinct real-space charge distributions, they experience different Hartree shifts. This naturally produces a stretching of the bands, even in the absence of strong correlations. While indirect signatures of this effect have been reported previously (e.g., via Landau level spectroscopy in Ref. [17]), Fig. 3 of our manuscript presents the first **direct** observation of this Hartree-driven band stretching.

Importantly, our measurements do not merely reveal the overall *magnitude* of the band stretching; they directly provide the **momentum-dependent energy shifts** δE_k . From the expression above, it is evident that these shifts encode detailed information about the real-space structure of the wavefunctions at different k . This is especially valuable in MATBG, where the flat bands are known to have strongly momentum-dependent wavefunctions arising from their nontrivial topology.

In fact, one of the most striking theoretical predictions for TBG, already present Bistritzer–MacDonald (BM) single-particle model, is that the flat-band wavefunctions change qualitatively across the mini-Brillouin zone. This effect does not need electronic

correlation and happens also when the bands are not extremely flat - for example, for TBG with twist angles larger than the magic angle. In the BM continuum mode, more than 90% of the momentum states in the band around neutrality have a localized wavefunction at the AA regions of the moiré cell. However, near the Γ point this structure is reversed: the states become more extended and feature a suppression of charge density at the AA sites.

Although this momentum-dependent real-space structure was used as the forms the basis for formulating the c - and f -electron decomposition in the strongly interacting topological heavy fermion (THF) models, it is not specific to that framework. Rather, it is a robust and very general feature of the non-interacting flat bands themselves. Our ability to measure $\delta E_H(k)$ with momentum resolution therefore provides direct access to these underlying wavefunction characteristics, offering a powerful new experimental probe of the topology-driven structure of the MATBG bands.

The new Extended Data Fig. 12 (shown below) provides a step-by-step illustration of how the observed Hartree stretching arises and how it is directly connected to the wavefunction structure of the flat bands. Panel a sketches the real-space Hartree potential $V_H(\mathbf{r})$ that develops upon *electron* filling. Because more than 90% of the flat-band wavefunctions in the BM model resides on the AA sites and have a f -electron Wannier function, the added carriers generate a Hartree potential that is strongly peaked at these sites and therefore highly repulsive for f -electrons. We emphasize that here we are discussing single particle wavefunctions, so the c/f decomposition is fully valid in this context. As shown in panel b, the *shape* of $V_H(\mathbf{r})$ remains practically fixed as a function of filling, while its overall amplitude grows approximately linearly with ν [1].

Panel c plots the BM charge densities $|\psi_f(\mathbf{r})|^2$ (red) and $|\psi_c(\mathbf{r})|^2$ (blue). The f -electron density has far greater overlap with $V_H(\mathbf{r})$ than the c -electron density and thus experiences a substantially larger Hartree shift. This is captured schematically in panel d, where both c - and f -electron energies increase linearly with ν , but with a much steeper slope for the f -electrons.

Panel e translates these energy shifts into momentum space. States near the Γ point - dominated by c -character - shift only weakly with filling, whereas the vast majority of states in the rest of the band - dominated by f -character - shift strongly. Because most of the flat-band states are f -like, the Fermi energy E_F (dashed line) remains effectively pinned to the flatter regions of the band as ν varies.

Our experiment, however, measures energies **relative to E_F** . Referencing to E_F has the effect shown schematically in panel f: the strongly shifting f -electron states remain near zero energy, while the weakly shifting c -electron states acquire an apparent *downward* slope with filling. This behaviour matches precisely the trend observed in Fig. 4 of the main text (Here we focus solely on the Hartree component; additional interaction terms responsible for Hubbard-like band splittings and cascades appear on top of this effect in the experiment.)

Finally, panel g shows how the ν -dependent bands of panel e appear once referenced to E_F : the bands exhibit a clear *stretching* near Γ - exactly as observed experimentally in Fig. 3a. This stretching provides compelling evidence for the distinct wavefunction character of states near Γ , a key prediction of the topological structure of the BM bands and a foundational ingredient of the THF model.

Extended Data Figure 12. Illustration of how filling-dependent Hartree band stretching arises from the real-space structure of the wavefunctions at different momenta.

[1] F. Guinea & N.R. Walet, Electrostatic effects, band distortions, and superconductivity in twisted graphene bilayers, *Proc. Natl. Acad. Sci. U.S.A.* 115 (52) 13174-13179.

Changes to the manuscript: We added new Extended Data Fig. 11 and Extended Data Fig. 12 and the corresponding discussions in Methods M13 to the revised manuscript.

Minor suggestion: From Figure2 to Figure 4, part of labels and guiding lines are difficult to recognize.

Changes to manuscript: we fixed the labels and guiding lines in Figs. 2 and 4

Referee #2

Xiao et al. report energy- and momentum-resolved spectroscopy of the interacting electronic band structure of magic angle twisted bilayer graphene (MATBG) using a quantum twisting microscope. They observe contrasting electronic behavior at different momenta. Near the mini Brillouin Zone Gamma point, the bands are gapless and dispersive, leading to light carriers (c-electrons), while the bands are very flat and exhibit repeating cascades elsewhere, consistent with heavy carriers (f-electrons). The results are interpreted as support for the topological heavy fermion and Mott semimetal models of MATBG. In addition, the measurements provide information about the

tunneling ratio w_0/w_1 , which has been poorly constrained, and reveal a 15 meV gap which is not understood. The work is extremely impressive, and it provides important new insight into the electronic structure of MATBG - it goes beyond traditional scanning tunneling spectroscopy by providing momentum resolution, and it has better energy and spatial resolution than nanoARPES. I therefore recommend eventual publication in Nature. However, I do have some technical comments and questions that the authors should address.

We thank the referee for their positive appraisal of our work.

1. One set of questions is related to the energy resolution and the broadening of the bands in MATBG. The authors report ~ 10 meV energy resolution on the basis of a fit to a feature near the Fermi level. However, many spectral features further away appear much more broadened. Is this consistent with expectations from lifetime broadening when tunneling far from the Fermi level, and what lifetime would this imply for the MATBG states? Addressing this could help to quantify the extent to which bands are broadened by interaction effects, if at all. This question is also relevant to some of the conclusions in the manuscript. For example, the manuscript notes that a diffuse “plume” reflecting an incoherent band with short lifetime. It is later stated that this band becomes stronger and sharper as it gradually shifts with filling to lower energies toward the Fermi level. Can this behavior be accounted for by lifetime effects?

We agree with the referee that understanding the origin of spectral broadening is important. We believe that the ~ 10 meV linewidth observed in the current measurements reflects intrinsic physics rather than technical limitations of the QTM.

Redacted

Within a simple lifetime-broadening picture, the spectral function peak corresponding to a quasiparticle at an energy E acquires a width inversely proportional to the lifetime and the spectral function's peak amplitude decreases correspondingly. In the current experiment, multiple bands coexist within a narrow energy range, making it challenging to extract the linewidths of the bands quantitatively. Instead, we can reliably extract the *peak height*, A_{peak} , which is a robust experimental observable and reflects the combined effects of quasiparticle residue and lifetime.

Inspection of Fig. 4 in the main text shows a clear and systematic trend: when a band lies far from the Fermi energy, its A_{peak} is small, and this maximal spectral weight increases continuously as the band approaches the Fermi level. This behaviour is qualitatively consistent with a lifetime that decreases with increasing excitation energy.

To address this more quantitatively, we added a **new Extended Data Fig. 13** to the revised manuscript, reproduced below.

- **Panel a** tracks the maximum of the spectral peak (dashed line) from where we can reliably see a peak at high energy to the point where it reaches the Fermi energy and the peak becomes less well defined.
- **Panel b** plots the one over the extracted peak height $1/A_{\text{peak}}$ versus peak energy E and compares two functional forms: $\frac{1}{A_{\text{peak}}} \propto (E - E_F) + b$ and $\frac{1}{A_{\text{peak}}} \propto a \cdot (E - E_F)^2 + b$. Overall, we see that a linear dependence of $\frac{1}{A_{\text{peak}}}$ on $E - E_F$ fits better the observations.

The height of a spectral-function peak reflects both the quasiparticle residue $Z(E - E_F)$ and the lifetime $\tau(E - E_F)$. In principle, both quantities may evolve as a Hubbard band moves to higher energies. If, however, we interpret the dominant trend as arising primarily from lifetime effects, the observed scaling would imply that it is more likely that $\tau(E - E_F) \propto 1/(E - E_F)$, and not the Fermi-liquid expectation $\tau(E - E_F) \propto 1/(E - E_F)^2$. Such dependence is often seen in strongly correlated / Hubbard systems, but we want to be careful and not make a strong claim based on this data.

Extended Data Figure 13. Spectral peak height vs quasiparticle energy.

Changes to the manuscript: We added new Extended Data Fig. 13 and the corresponding discussions in Methods M14 to the revised manuscript.

Lastly, does the word ‘incoherent’ have a technical meaning? If so, this should be defined. If not, it should be in quotes or that should be stated explicitly.

By ‘incoherent’ we mean that the observed spectral peak has a width which is comparable or larger than its energy. We have clarified it in the text.

2. There is a paragraph which describes a pronounced “stretching” around the Gamma point and states that it shifts in energy. I do not understand these comments nor what they refer to. Do the authors mean that there is a peak near Gamma at $\nu=-4$ in Fig. 3a, left panel? And a similar dip at $\nu=4$ in the second to the right panel? Is this a signature of changes in band shape as carrier are populated, e.g. as proposed in Ref. 53? Do the

data distinguish between such an explanation and more recent theory such as the topological heavy fermion model?

We thank the referee for this question, which made us realize that our original explanation of this feature was not sufficiently clear. In the revised manuscript we have added a more systematic description, which we summarize below.

The referee is absolutely right - by stretching we mean that there is a peak at Γ at $\nu = -4$, which continuously evolves to a dip at Γ at $\nu = 4$. To highlight better this evolution, we have added a new Extended Data Fig. 11 that shows the measured bands alongside schematic guides to the eye and uses arrows to highlight the stretching.

Extended Data Figure 11. Highlighting the filling-dependent stretching of the bands.

This stretching reflects a change in the band shape as carriers are added. However, the effect arises from a mechanism that is both simpler and more general than the one proposed in Ref. 53. Specifically, it originates from the momentum-dependent response of the electronic wavefunctions to the Hartree potential that builds up with increasing carrier density. Early theoretical works (e.g., Refs. [23], [24]) already noted that while moiré-scale electrostatic potentials are negligible at charge neutrality, a spatially varying Hartree potential $V_H(\mathbf{r})$ develops as electrons or holes are added. The energy of a state at momentum k then shifts according to its real-space overlap with this potential,

$$\delta E_H(k) = \int d^2r V_H(r) |\psi_k(r)|^2$$

because different k -states have distinct real-space charge distributions, they experience different Hartree shifts. This naturally produces a stretching of the bands, even in the absence of strong correlations. While indirect signatures of this effect have been reported previously (e.g., via Landau level spectroscopy in Ref. [17]), Fig. 3 of our manuscript presents the first **direct** observation of this Hartree-driven band stretching.

Importantly, our measurements do not merely reveal the overall *magnitude* of the band stretching; they directly provide the **momentum-dependent energy shifts** δE_k . From the expression above, it is evident that these shifts encode detailed information about the real-space structure of the wavefunctions at different k . This is especially valuable in MATBG, where the flat bands are known to have strongly momentum-dependent wavefunctions arising from their nontrivial topology.

In fact, one of the most striking theoretical predictions for TBG, already present Bistritzer–MacDonald (BM) single-particle model, is that the flat-band wavefunctions change qualitatively across the mini-Brillouin zone. This effect does not need electronic correlation and happens also when the bands are not extremely flat—for example, for TBG with twist angles larger than the magic angle. In the BM continuum mode, more than 90% of the momentum states in the band around neutrality have a localized wavefunction at the AA regions of the moiré cell. However, near the Γ point this structure is reversed: the states become more extended and feature a suppression of charge density at the AA sites.

Although this momentum-dependent real-space structure was used as the basis for formulating the c - and f -electron decomposition in the strongly interacting topological heavy fermion (THF) models, is not specific to that framework. Rather, it is a robust and very general feature of the non-interacting flat bands themselves. Our ability to measure $\delta E_H(k)$ with momentum resolution therefore provides direct access to these underlying wavefunction characteristics, offering a powerful new experimental probe of the topology-driven structure of the MATBG bands.

The new Extended Data Fig. 12 (shown below) provides a step-by-step illustration of how the observed Hartree stretching arises and how it is directly connected to the wavefunction structure of the flat bands. Panel a sketches the real-space Hartree potential $V_H(\mathbf{r})$ that develops upon *electron* filling. Because more than 90% of the flat-band wavefunctions in the BM model resides on the AA sites and have a f -electron Wannier function, the added carriers generate a Hartree potential that is strongly peaked at these sites and therefore highly repulsive for f -electrons. We emphasize that here we are discussing single particle wavefunctions, so the c/f decomposition is fully valid in this context. As shown in panel b, the *shape* of $V_H(\mathbf{r})$ remains fixed as a function of filling, while its overall amplitude grows approximately linearly with ν [1].

Panel c plots the BM charge densities $|\psi_f(\mathbf{r})|^2$ (red) and $|\psi_c(\mathbf{r})|^2$ (blue). The f -electron density has far greater overlap with $V_H(\mathbf{r})$ than the c -electron density and thus experiences a substantially larger Hartree shift. This is captured schematically in panel d, where both c - and f -electron energies increase linearly with ν , but with a much steeper slope for the f -electrons.

Panel e translates these energy shifts into momentum space. States near the Γ point - dominated by c -character - shift only weakly with filling, whereas the vast majority of states in the rest of the band - dominated by f -character - shift strongly. Because most

of the flat-band states are f -like, the Fermi energy E_F (dashed line) remains effectively pinned to the flatter regions of the band as ν varies.

Our experiment, however, measures energies **relative to E_F** . Referencing to E_F has the effect shown schematically in panel f: the strongly shifting f -electron states remain near zero energy, while the weakly shifting c -electron states acquire an apparent *downward* slope with filling. This behaviour matches precisely the trend observed in Fig. 4 of the main text (Here we focus solely on the Hartree component; additional interaction terms responsible for Hubbard-like band splittings and cascades appear on top of this effect in the experiment.)

Finally, panel g shows how the ν -dependent bands of panel e appear once referenced to E_F : the bands exhibit a clear *stretching* near Γ - exactly as observed experimentally in Fig. 3a. This stretching provides compelling evidence for the distinct wavefunction character of states near Γ , a key prediction of the topological structure of the BM bands and a foundational ingredient of the THF model.

Extended Data Figure 12. Illustration of how filling-dependent Hartree band stretching arises from the real-space structure of the wavefunctions at different momenta.

[1] F. Guinea & N.R. Walet, Electrostatic effects, band distortions, and superconductivity in twisted graphene bilayers, *Proc. Natl. Acad. Sci. U.S.A.* 115 (52) 13174-13179.

Changes to the manuscript: We added new Extended Data Fig. 11 and Extended Data Fig. 12 and the corresponding discussion in Methods M13.

3. In Fig. 3c, there are several Gaussian fits to the data of different colors, but not all datasets are fit with the same number of Gaussians. How is it determined how many Gaussians to fit, and what is the explanation for all the different spectral features and the filling-dependence on the number of features? Do the authors have an understanding

of the apparent changes in spectral weight? These questions require more discussion in the manuscript.

We thank the referee for this question. Indeed, we did not write explicitly our fitting procedure. We now added to the revised manuscript, Methods M7 and Extended Data Fig. 7 that explain this in detail. We repeat the explanation briefly below:

The Gaussian fitting procedure is performed as follows. We first identify the peak centers directly from the spectroscopy data by tracing the local maxima in the contour representation of Fig. 4a; these trajectories are shown as dashed black lines in Extended Data Fig. 7. This analysis reveals two classes of features: (a) the “Hubbard” bands that exhibit the characteristic cascading behavior, and (b) additional excitations that remain fixed near ± 15 meV. For each filling, the number of peaks identified in this manner determines the number of Gaussians used in the corresponding spectral fit. The peak positions extracted from the dashed-line trajectories serve as initial guesses for the fit, though they are allowed to vary. The free parameters in the fitting routine are the Gaussian center, width, amplitude, and the background offset.

We discussed the spectral weight changes with energy in response to a previous question by the referee, and this discussion is also included in the revised manuscript.

Extended Data Figure 7. Gaussian peak fitting procedure.

Changes to the manuscript: We have added to the revised manuscript Extended Data Fig. 7 and a corresponding discussion in Methods section M7.

Referee #3

This work applies the technique of quantum twisting microscope, recently developed by Ilani’s group, to address one of the outstanding theoretical puzzles in the field of moire materials: the nature of the electronic carriers in twisted bilayer graphene. The results beautifully and conclusively illustrates the dual nature of the interacting bands, showing distinct behavior close to Gamma and away from Gamma. The data is of very high quality and the results are likely to have a big impact on understanding correlation

in flat bands and how they relate to band topology. However, I have some concerns about the way the authors interpret some of their measurements that need to be addressed before I can recommend for publication:

We thank the referee for their positive appraisal of our work.

1. The authors employ the terminology of “c” and “f” electrons in somewhat ambiguous ways that make certain aspects of the interpretation unclear and could be misleading to the reader. Sometimes the author refer to “f electrons” as the “localized” states, the “heavy” states, the states away from Gamma, or the states with a particular orbital character (as in the THF model). Conversely the “c” electrons are the “itinerant” states, the “light” states, the states near Gamma, or the states with a different type of orbital character. While the decoupled limit of the topological heavy fermion model provides a case where these properties can all coincide, the experiment does not appear to be in that limit. Instead, the “f” and “c” orbitals strongly hybridize at generic momenta. They could further be strongly renormalized by interactions, e.g. c-electrons could be dressed by particle hole pairs of f-electrons or vice versa. These effects do not change the overall picture of a dual character associated with light/heavy carriers close to Gamma/away from Gamma and are consistent with all experimental observation without being tied to specific microscopic orbitals. For example, Is it clear that the orbitals near Gamma have mostly c-like orbital character, just because they are light? The orbital character itself is not measured in the current experiment. Furthermore, the spectrum at the Fermi energy in Fig. 5 seems to suggest a strongly doping dependent spectral weight near the Fermi energy, especially upon electron doping $\nu=2$. This suggests that quasiparticle dressing could be significant. Also, as the Fermi energy moves further and further away from Gamma, it becomes increasingly unclear whether the mostly c-character electrons are expanding in phase space (due to mixing with remote bands?) or the mostly f-character electrons are gaining more dispersion. Thus, more care should be taken regarding what is meant by “f” and “c” because the various defining characteristics given do not generally coincide. I suggest the authors adopt a more agnostic language when referring to the observation to avoid misleading the reader, and possibly discuss the caveats associated with any given theory interpretation.

We thank the referee for this valuable and insightful comment and fully agree with the points raised. Indeed, our experimental findings are robust and stand on their own without the need to commit to any particular theoretical framework. Following the referee’s advice, we have carefully revised the manuscript to adopt a more agnostic language when discussing our observations – referring to “light” and “heavy” electrons for states near and away from Γ - and limiting the use of “c” and “f” terminology only to contexts where it is appropriate. Specifically, we retain the *c/f* terminology in two cases: (a) when discussing single-particle properties such as the topology-driven real-space structure of the BM flat-band wavefunctions, and (b) when introducing or comparing with the THF model, where this language is built into the theoretical framework.

The referee raises several thoughtful points, all of which we address in detail in our responses to the following questions. Here, we would like to comment specifically on one important statement: that that **"the orbital character itself is not measured in the current experiment."**

While it is true that our spectroscopy does not directly measure orbital character, we emphasize that our experiment *does* provide key information about the real-space structure of the wavefunctions through the **momentum-resolved Hartree energy shifts**. In retrospect, we realize that we did not explain this point clearly enough in the original manuscript. In the revised version we therefore added a detailed, step-by-step explanation of how the observed filling-dependent stretching of the bands encodes information about the wavefunctions at different momenta. In particular, we show that the Hartree-driven band stretching demonstrates that the wavefunctions near and away from Γ have markedly different real-space structures, in line with what is expected from the topological nature of the flat bands.

We stress that this topological structure is a **single-particle** property: it does not require strong correlations and already appears in the single-particle BM description of the flat bands. THF-like models build on this fact - namely, that the flat-band wavefunctions can be conveniently written in a "c/f" basis.

To clarify these points, the revised manuscript includes (i) a dedicated Extended Data section with two new figures that illustrate the effect step by step, and (ii) a rewritten discussion in the main text, where we emphasize that the Hartree stretching probes the momentum dependence of the wavefunctions structure - a feature arising from the topology of the flat bands and far more general than any specific THF interpretation.

For completeness, we provide a version of this explanation below:

The experimental observation that provides information about the nature of the wavefunctions is the filling-factor-dependent stretching of the energy bands, as seen in Fig. 3a in the main text. By stretching we mean that the bands have a peak at Γ for $\nu = -4$, which continuously evolves into a dip at Γ at $\nu = 4$. To highlight this evolution better, we have added a new Extended Data Fig. 11 that shows the measured bands alongside schematic guides to the eye and uses arrows to highlight the stretching.

Extended Data Figure 11. Highlighting the filling-dependent stretching of the bands.

This stretching originates from the momentum-dependent response of the electronic wavefunctions to the Hartree potential that builds up with increasing carrier density. Early theoretical works (e.g., Refs. [23], [24]) already noted that while moiré-scale electrostatic potentials are negligible at charge neutrality, a spatially varying Hartree potential $V_H(\mathbf{r})$ develops as electrons or holes are added. The energy of a state at momentum k then shifts according to its real-space overlap with this potential,

$$\delta E_H(k) = \int d^2r V_H(r) |\psi_k(r)|^2$$

because different k -states have distinct real-space charge distributions, they experience different Hartree shifts. This naturally produces a stretching of the bands, even in the absence of strong correlations. While indirect signatures of this effect have been reported previously (e.g., via Landau level spectroscopy in Ref. [17]), Fig. 3 of our manuscript presents the first **direct** observation of this Hartree-driven band stretching.

Importantly, our measurements do not merely reveal the overall *magnitude* of the band stretching; they directly provide the **momentum-dependent energy shifts** δE_k . From the expression above, it is evident that these shifts encode detailed information about the real-space structure of the wavefunctions at different k . This is especially valuable in MATBG, where the flat bands are known to have strongly momentum-dependent wavefunctions arising from their nontrivial topology.

In fact, one of the most striking theoretical predictions for TBG, already present Bistritzer–MacDonald (BM) single-particle model, is that the flat-band wavefunctions change qualitatively across the mini-Brillouin zone. This effect does not need electronic correlation and happens also when the bands are not extremely flat - for example, for TBG with twist angles larger than the magic angle. In the BM continuum mode, more than 90% of the momentum states in the band around neutrality have a localized wavefunction at the AA regions of the moiré cell. However, near the Γ point this

structure is reversed: the states become more extended and feature a suppression of charge density at the AA sites.

Although this momentum-dependent real-space structure was used as the basis for formulating the c - and f -electron decomposition in the strongly interacting topological heavy fermion (THF) models, it is not specific to that framework. Rather, it is a robust and very general feature of the non-interacting flat bands themselves. Our ability to measure $\delta E_H(k)$ with momentum resolution therefore provides direct access to these underlying wavefunction characteristics, offering a powerful new experimental probe of the topology-driven structure of the MATBG bands.

The new Extended Data Fig. 12 (shown below) provides a step-by-step illustration of how the observed Hartree stretching arises and how it is directly connected to the wavefunction structure of the flat bands. Panel a sketches the real-space Hartree potential $V_H(\mathbf{r})$ that develops upon *electron* filling. Because more than 90% of the flat-band wavefunctions in the BM model resides on the AA sites and have a f -electron Wannier function, the added carriers generate a Hartree potential that is strongly peaked at these sites and therefore highly repulsive for f -electrons. We emphasize that here we are discussing single particle wavefunctions, so the c/f decomposition is fully valid in this context. As shown in panel b, the *shape* of $V_H(\mathbf{r})$ remains practically fixed as a function of filling, while its overall amplitude grows approximately linearly with ν [1].

Panel c plots the BM charge densities $|\psi_f(\mathbf{r})|^2$ (red) and $|\psi_c(\mathbf{r})|^2$ (blue). The f -electron density has far greater overlap with $V_H(\mathbf{r})$ than the c -electron density and thus experiences a substantially larger Hartree shift. This is captured schematically in panel d, where both c - and f -electron energies increase linearly with ν , but with a much steeper slope for the f -electrons.

Panel e translates these energy shifts into momentum space. States near the Γ point - dominated by c -character - shift only weakly with filling, whereas the vast majority of states in the rest of the band - dominated by f -character - shift strongly. Because most of the flat-band states are f -like, the Fermi energy E_F (dashed line) remains effectively pinned to the flatter regions of the band as ν varies.

Our experiment, however, measures energies **relative to E_F** . Referencing to E_F has the effect shown schematically in panel f: the strongly shifting f -electron states remain near zero energy, while the weakly shifting c -electron states acquire an apparent *downward* slope with filling. This behaviour matches precisely the trend observed in Fig. 4 of the main text (Here we focus solely on the Hartree component; additional interaction terms responsible for Hubbard-like band splittings and cascades appear on top of this effect in the experiment.)

Finally, panel g shows how the ν -dependent bands of panel e appear once referenced to E_F : the bands exhibit a clear *stretching* near Γ - exactly as observed experimentally in Fig. 3a. This stretching provides compelling evidence for the distinct wavefunction

character of states near Γ , a key prediction of the topological structure of the BM bands and a foundational ingredient of the THF model.

Extended Data Figure 12. Illustration of how filling-dependent Hartree band stretching arises from the real-space structure of the wavefunctions at different momenta.

[1] F. Guinea & N.R. Walet, Electrostatic effects, band distortions, and superconductivity in twisted graphene bilayers, *Proc. Natl. Acad. Sci. U.S.A.* 115 (52) 13174-13179.

Changes to the manuscript: We added new Extended Data Fig. 11 and Extended Data Fig. 12 and the corresponding discussion in Methods M13.

2. In STM measurements, the 15 meV feature is still observed as a flat-band splitting when the chemical potential is in the remote bands. Do the authors also see the 15 meV feature when the chemical potential is in the remote bands? If the 15 meV feature is active in the remote bands it suggests it should be attributed to a single particle effect, rather than a new unaccounted for excitations as the authors propose (the remote bands are expected to be very weakly correlated).

This is an excellent point. Indeed, observing similar splitting when the Fermi level is in the remote bands and the flat bands are completely full would point to a single particle origin of this energy scale.

In our experiments, when the flat bands are below the Fermi energy ($\nu > 4$), we do not resolve two distinct maxima; instead, we observe a single broader peak. Moreover, we see a single peak at all momenta. This observation is consistent over many measurements done at different locations and in two independent samples. Very rarely we do see such splitting, but this can be that at these specific locations there is a local strain that causes such single-particle splitting. But the generic case is a single, broader peak.

Extended Data Figure 16. Spectral feature $\Delta E = 15\text{meV}$ inside and outside the flat bands.

To demonstrate this more quantitatively, we added Extended Data Fig. 16 (reproduced above) which compares two spectroscopic linecuts (dashed lines, panel a): at $\nu = 1.5$ (panel b) at $\nu = 4.3$ (panel c). At $\nu = 1.5$, we can clearly see an excitation at $\Delta E \approx -15\text{meV}$ which is clearly resolved from the heavy band peak just above the Fermi energy. In contrast, the spectroscopy at $\nu = 4.3$, shows only a single broader peak (FWHM $\sim 21\text{meV}$). Fitting it with two gaussians with the same lifetime we measure at partial filling ($\sim 10\text{meV}$) and a splitting of 15meV we get a poor fit (panel d).

Indeed, the simplest scenario would have been if there was a single-particle splitting that explains the 15meV excitation. Specifically, several theory groups are now suggesting that this excitation may arise from strain in MATBG. While we cannot definitively rule out strain, we believe that we have accumulated experimental evidence that makes this explanation unlikely.

We present this in new Extended Data section (Methods 10 and Extended Data Fig. 8) which includes new data that was not present in the original submission. This section summarizes the evidence that strain alone is unlikely to account for the observed 15meV feature and is intended to help guide future efforts to resolve this puzzle, which we suspect may have broader implications for understanding MATBG.

Evidence that strain is unlikely to be the origin of the 15meV excitation:

1. Universality across widely separated regions of the same device.

We observe the same excitation energy at multiple locations within the sample, separated by several microns - distances large enough that these regions effectively behave as independent devices. We have added new Extended Data Fig. 8 to the revised manuscript (reproduced below), showing the spectroscopy measured at different locations (individual panels) together with a map of the measurement locations (center panel). and the corresponding spectra.

2. Universality across different samples.

We detect similar excitations (13–15 meV) in two distinct MATBG devices, measured in separate cooldowns, indicating that the feature is not sample-specific or related to a particular thermal cycle.

3. No correlation with independently measured strain.

We have used room-temperature conductive-AFM to image the moiré lattice in each of the measured locations (small insets in Extended Data Fig. 8). From these images we extract the hetero-strain, which varies by a factor of ~ 2.5 across the measured positions (from $\varepsilon = 0.02\%$ to $\varepsilon = 0.06\%$). Despite this substantial variation, the excitation energy remains essentially constant (≈ 13 – 15 meV).

4. Quantitative inconsistency with theoretical expectations.

For the least-strained region ($\varepsilon = 0.02\%$), the observed excitation energy is ~ 15 meV. Existing theoretical models predict a strain-induced level splitting of ~ 10 meV per 0.1% strain [1], which would imply a splitting of only ~ 2 meV for $\varepsilon = 0.02\%$ - an order of magnitude smaller than what we measure.

[1] J. Herzog-Arbeitman, J. Yu, D. Călugăru, H. Hu, N. Regnault, O. Vafek, J. Kang, B. A. Bernevig, Topological Heavy Fermion Model as an Efficient Representation of Atomistic Strain and Relaxation in Twisted Bilayer Graphene, *Physical Review B* 112, 125128 (2025).

Extended Data Figure 8. Spectroscopy of the flat bands at far-separated locations along the sample.

Changes to the manuscript: We have added Extended Data Fig. 8 and Extended Data Fig. 16, and corresponding discussions in Methods M10 and M17 to the revised manuscript.

I found the Dirac revival description rather confusing and I believe the zero c-f hybridization limit employed in that discussion is inapplicable to the parameters

relevant to the experiment. For one, the decoupled limit contains double the number of c-electrons. It is the c-f hybridization which is responsible to sending half of the Gamma c-electrons to the remote bands while the other half forms the near-Gamma part of the flat bands. An interpretation of Fig 4 would then suggest that both remote band and flat band c-character electrons are filled equally, as the Dirac cone corresponds to a hybridization between these. This seems surprising because the remote bands are rather far in energy in practice.

It is also not clear why the c-electron dispersion would move in an opposite manner due to the f-electron band filling. More f-electrons should mean a larger Hartree potential, pushing the c-electrons down even further below the f-band. They can furthermore be continuously doped at an energy U until the band is full, thereby pinning the chemical energy to this value. I thus cannot see how a downward shift would be possible in the picture the authors put forward. One can get downward shifts, in the sense the authors describe, through flavor polarization transitions as in the old Dirac revival story. But this is because there are correlations between the doped heavy electrons, such that they wish to be collectively doped so that their flavors can align. While calculations on the decoupled THF model have shown these features, this is because they assume the f-electron occupation is the same integer at every site thereby necessitating doping every f-electron at once.

We thank the referee for these thoughtful comments. We fully agree that our experiment directly demonstrates the importance of **c-f hybridization** in the MATBG flat bands. Indeed, our measured dispersions show that the bands are only properly described in the regime of *strong* hybridization, $\gamma/U > 1$.

That said, several robust features of our data – including the physics of the **Dirac revivals** – follow quite generally from the coexistence of **two classes of electronic states** that satisfy two basic conditions:

1. **They have light and heavy characters.**
2. **They have different real-space wavefunctions and therefore couple differently to the Hartree potential.**

While these conditions are naturally realized by the microscopic BM “c” and “f” orbitals, they can be satisfied much more broadly by any two “c-like” and “f-like” components that retain these two essential features. As discussed in response to earlier questions, our data directly shows that states near and away from the Γ point indeed satisfy these two conditions.

In particular, the band stretching observed in Fig. 3 already demonstrates that these two electronic sectors experience different Hartree shifts. The Dirac revivals represent a second, equally important experimental signature that follows from the same ingredients. We recognize that our original manuscript failed to explain this important physics clearly enough. We have therefore added a dedicated Extended Data section

providing a step-by-step account of the mechanism. Before presenting the full explanation, we address the referee’s specific questions:

- (i) **“It is not clear why the c-electron dispersion would move in an opposite manner due to the f-electron band filling. More f-electrons should mean a larger Hartree potential, pushing the c-electrons down even further below the f-band”**

We completely agree with the referee that f electrons experience larger Hartree potential. Indeed, in the zoom-out view of our measurements the c electrons are pushed down with filling due to the Hartree effect.

However, a closer, zoom-in view of the spectra reveals a non-monotonic behaviour: just before each integer filling, the Γ -point spectral peak shifts slightly *upwards*, signalling the depopulation of the light electrons. This aligns with the observed negative compressibility in our earlier work (Zondiner et al., Ref. 19). Remarkably, if superpose $-\mu(\mathbf{v})$, from that experiment onto the Γ -point spectral peak measured here, the correspondence is extremely close – including in the regions of negative compressibility where μ decreases with filling. We emphasize that while in Zondiner *et al.* the Dirac revivals were observed in the *global compressibility*, here we see the physics at a far more microscopic level – in the non-monotonic evolution of the spectral peak at Γ -point. In addition, this is also seen in the non-monotonic evolution of the bottom and top of the remote band edges, also at Γ (Fig. 4a). This momentum-resolved resets now provides direct, microscopic evidence for the Dirac revival mechanism.

As we show below, this behavior is a generic consequence of having light and heavy electronic species with different couplings to the Hartree potential.

Under these conditions, populating the heavy electrons generically **depopulates** the light electrons – i.e., the light-electron states shift *toward* the Dirac point as the heavy states are populated. This is precisely the “Dirac revival” phenomenon.

The underlying physics is deeply connected to the phenomenon of **population inversion** observed in quantum dots and explained by Silvestrov & Imry [Phys. Rev. Lett. **85**, 2565 (2000)] and subsequent work [Rev. B 71, 201308R 2005 and Phys. Rev. B 72, 125316]. There is a dot with two energy levels – wide and narrow – which couple differently to the gate, exhibit a non-monotonic filling sequence: the wide level first fills, then empties when the narrow level becomes occupied, and then refills. This behaviour maps directly onto the heavy/light electron structure in the current system; the “resets” identified at that work correspond exactly to the reshuffling of charge in the Dirac revivals observed here.

- (ii) **“One can get downward shifts, in the sense the authors describe, through flavor polarization transitions as in the old Dirac revival story”**

This is correct, but **not necessary**. As we explicitly show using a toy model in the new Extended Data section, the same non-monotonic energy shifts and charge reshuffling

arise *without* invoking flavor polarization. The phenomenon appears purely from the coexistence of light and heavy electronic sectors with different Hartree couplings.

(iii) **“Decoupled THF model have shown these features, because they assume the f-electron occupation is the same integer at every site thereby necessitating doping every f-electron at once”**

We agree that enforcing all f electrons to be doped at once is an unrealistic scenario for MATBG. Importantly, however, this assumption is *not required* to obtain Dirac revivals. In our toy model the heavy-electron manifold has a **finite bandwidth** - it is *not* filled all at once - and yet the Dirac revivals appear clearly.

For conceptual clarity, we treat the heavy and light sectors as decoupled in this simplified model. They do not need to be the microscopic BM c and f electrons; they simply need to inherit the two experimentally motivated features: light vs. heavy character, and different Hartree couplings.

In this model the magnitude of the revival is proportional to the direct Coulomb repulsion between the two species. Hybridization may smooth the evolution but will not eliminate the effect. In a sense, the reader can use the fact that in the experiments we do see these revivals directly to appreciate that these ingredients are explicitly present.

To clarify all these points in the revised manuscript we have:

(a) added a new Experimental Data section presenting a transparent toy model that captures the essence of the Dirac revivals.

(b) Updated figure 4 to emphasize that the cartoons in this figure are the result of this toy model and explicitly name the two components "c-like" and "f-like", reflecting their phenomenological roles rather than their microscopic origin.

For completeness we provide below the full explanation of the toy model used in this new Extended Data section.

The components/assumption of the model are as follows:

1. We assume two decoupled electronic components, which we call "c-like" and "f-like". They are not necessarily the original c and f electrons, but they do carry their two important aspects: light/heavy, different Hartree couplings
2. Specifically, we model the f-like electrons with a simplified quantum dot model, which assumes completely flat bands but with a phenomenological "lifetime" broadening that gives a finite width to the energy bands. We compute the f-electron spectral function to capture the Hubbard bands filling evolution.
3. We assume that is a finite direct Coulomb coupling between c-like and f-like electrons, W , and that $W < U$.

The f-electron quantum dot model Hamiltonian is: $H = Un_f(n_f - 1)/2$, where U is the effective charging energy and n_f is the f-electron occupation. This quantum dot has eight flavors such that $0 \leq n_f \leq 8$. We compute the spectral function given by this Hamiltonian: $A(\omega) = Z^{-1} \sum_{n,m} \delta(\omega - \varepsilon_n + \varepsilon_m) (e^{-\beta\varepsilon_m} + e^{-\beta\varepsilon_n}) \langle l|c_{\alpha,f}^\dagger|m\rangle|^2$, where $Z = Tr(e^{-\beta(H-\mu n)})$ is the partition function. $\beta = 1/k_B T$, and ε_m is the energy of the $|m\rangle$ eigenstates. We compute the spectral function $A(\omega, n_f)$ and then add smearing in energy to mimic the lifetime effect seen the measurement. From the smeared spectral function, we compute the relation $\mu_f(n_f)$.

In addition to the quantum dot, we add Dirac (c-like) electrons. Let n_f and n_c denote the f and c fillings ($n = n_f + n_c$ is fixed externally). The interaction term can be written as $Wn_f n_c = Wn_f(n - n_f)$. The total energy is

$$E_{\text{tot}}(n_f; n) = E_f(n_f) + E_c(n_c) + Wn_f(n - n_f),$$

and the stationarity condition:

$$\frac{dE_{\text{tot}}}{dn_f} = 0$$

gives a simple self-consistency equation:

$$\mu_f(n_f) - \mu_c(n_c) + W[n - 2n_f] = 0,$$

with $\mu_{f/c} \equiv dE_{f/c}/dn_{f/c}$. We solve this equation graphically to obtain (n_f, n_c) at each total filling n , and then compute the corresponding spectral functions in momentum space for both sectors.

We now include Extended Data Fig. 14 showing the model results over $n = 0$ to 1.5. Panels a and b display the f and c spectral functions in momentum space versus filling, reproducing: (i) the Hubbard-band evolution of heavy f electrons; and (ii) the Dirac revival of light c electrons. Specifically, in Panel b, we track the Dirac point evolution: it first shifts downward with increasing ν due to the Hartree potential, and then when f states start to populate it shift upward (black solid guideline); This is precisely the "Dirac revival" phenomenology, and within this model we can see that it is driven by the W Coulomb term. The schematic in Fig. 4 of the main text is based on this calculation.

Extended Data Figure 14. Spectral functions calculated with a Toy model, capturing the Dirac revival phenomenology.

In our toy model we do not force instantaneous doping of f-electrons at every site. Instead, after solving the quantum dot model, we add smearing to the chemical potential to mimic finite f bandwidth, such that f electrons fill gradually. In this case, if we don't add the f and c electrons repulsion term, we do not get the depopulation of c electrons, and the system will behave as the referee described: c electrons get filling until the energy scale $\sim U$ and then pinned when f electrons start populating. Only when we $Wn_f n_c$ we observe the charge reshuffling behaviour.

The above model does not include c-f hybridization. To show that the Dirac revivals physics is robust and appears also in the case of strong hybridization $\gamma/U > 1$, we solve a more realistic THF model that also includes hybridization, solved within Hubbard-I approximation (detailed in Method M16). New Extended Data Fig. 15 (shown below) plots the calculated filling factor dependence of the spectral function. The black line traces the position of the c electrons Dirac point, clearly demonstrating that the Dirac revivals phenomenology, observed in the experiment, appears also in the presence of hybridization

Finally, we note that light electrons depopulation was also observed in recent DMFT calculations in the limit $\gamma/U > 1$, where it was similarly interpreted as the reshuffling between f and c electrons [1].

[1] M. J. Calderón, A. Camjayi, A. Datta, E. Bascones, Cascades in Transport and Optical Conductivity of Twisted Bilayer Graphene, Physical Review B 112, L041126 (2025).

Extended Data Figure 15. Filling factor dependence of the spectral function calculated using a THF model that includes strong c-f hybridization.

Changes to the manuscript: We have added Extended Data Figs. 14 and 15 and the corresponding discussions in a new Methods section M15 and M16.

4. In addition to the ‘resets’ identified close to integer fillings, there seem to also be changes in the quasiparticle weight e.g. the weight at Gamma at the chemical potential gets progressively weaker as we move away from neutrality. This seems to suggest significant quasiparticle dressing inconsistent with the simple model employed by the authors. Can the authors comment on that and discuss possible caveats in their interpretation?

The apparent reduction of spectral weight at the Fermi energy near the Γ point is *not* due to a decrease in quasiparticle weight. Instead, it is a direct consequence of the filling-dependent Hartree shifts, which modify the band dispersion and push the light-electron Dirac point progressively farther below the Fermi energy as doping increases. As this shift grows with filling, the Γ -point states simply move out of the energy window probed at E_F , giving the appearance of vanishing spectral weight. This mechanism also explains the increasing fraction of light-electron character within the populated band upon doping, as shown in the revised Fig. 5.

Following the referee’s advice, we have updated this figure to avoid referring to “c” and “f” electrons and instead label the experimentally observed boundary between *light* and *heavy* carriers, which is the quantity directly measured in our experiment.

Changes to the manuscript: We revised Fig. 5 and corresponding text.

5. It’s not clear how the extraction procedure for “s” relates to the definition in Ref. 42. Indeed, the extraction appears to depend mostly on the strength of the Hartree dispersion. This gradually makes the band more and more dispersive, such that more and more electrons do not qualify as “f”-character anymore. However, it seems to also be likely for the electrons with mostly f-like orbital character to gain more dispersion through a stronger potential magnifying a fixed, smaller, c-component. In the latter

case, the orbital proportions of the total flat band does not change and “s”, as defined in Ref. 42, would correspondingly also not change.

We accept this comment. Since the experiment does not probe "s" directly we decided to remove the corresponding discussion/speculations from the manuscript.

6. There are at least two distinct semimetals possible at charge neutrality. One is a symmetry broken semimetal stabilized by strain and another is a thermally disordered “Mott” semimetal. Similarly, there are three distinct kinds of insulators proposed at non-zero integer fillings: symmetry-broken intervalley coherent states, kekule spiral states, and thermally disordered insulators. Can the authors comment on whether any of these best fit the data or whether it is insufficient to make this conclusion?

We thank the referee for this important question. We address the two regimes the referee asks about:

Charge neutrality:

Indeed, as the referee says, there are two physically distinct mechanisms to produce a semi-metal at charge neutrality: the Mott semimetal [Ref. 30, 37, 38, 42 in the manuscript] and the strain-induced or spontaneously C3 symmetry broken semimetal [1-3]. The two states give very similar spectra, with Hubbard-like nearly flat bands away from the Γ point and a band touching at or near Γ . There should be small differences in the spectra between the two states – e.g., the two Dirac points of the strain-induced semi-metal are generically not precisely at Γ , and do not have to occur at the same energy – but we believe that these differences may be too small for us to observe within experimental resolution. The main factor limiting the resolution is likely to be the tip size, limiting the resolution in momentum space, and preventing us from discerning the fine details of the dispersion near Γ that would allow us to distinguish the two types of semi-metals.

Non-zero integer fillings:

Our experiment is not directly sensitive to symmetry breaking, as the tunnelling from the QTM tip is not sensitive to the electron spin and valley, and we do not have real space sensitivity that can identify the spatial modulation of the spectrum that occurs in inter-valley coherent or Kekulé spiral states. We can say that we have not observed any obvious signatures of symmetry breaking in the momentum-resolved electronic spectrum. In particular, we do not observe a clear gap in the electronic spectrum at the Fermi level at any density (see Fig. 3a). Therefore, we argue that our results are incompatible a gap-opening broken symmetry state at integer filling at the temperature of the experiment.

We have added a brief discussion of the interpretation of our results according to the points mentioned above to the concluding section of the paper.

[1] S. Liu, E. Khalaf, J. Y. Lee, A. Vishwanath, Nematic Topological Semimetal and Insulator in Magic-Angle Bilayer Graphene at Charge Neutrality, *Physical Review Research* 3, 013033 (2021).

[2] D. E. Parker, T. Soejima, J. Hauschild, M. P. Zaletel, N. Bultinck, Strain-Induced Quantum Phase Transitions in Magic-Angle Graphene, *Physical Review Letters* 127, 027601 (2021).

[3] F. Xie, A. Cowsik, Z.-D. Song, B. Lian, B. A. Bernevig, N. Regnault, Twisted Bilayer Graphene. VI. An Exact Diagonalization Study at Nonzero Integer Filling, *Physical Review B* 103, 205416 (2021).

Changes to the manuscript: We have added corresponding discussions in Methods M18 to the revised manuscript.

7. Some other smaller points

- It is not clear what the authors mean when they refer to the f-electrons as “localized.” At least in the beginning of the manuscript, the f-electrons referred to states with momenta away from Gamma. But states in momentum space are never localized; they are of course Bloch waves. Perhaps the authors mean that wavepackets constructed from these parts of momentum space (excluding Gamma) are tightly localized in some sense? It would be helpful if the authors specified what they mean here.

We realized that the f and c electron language is confusing, so we avoided it. As regarding to localized – we changed to say tightly localized – since most of the band is flat.

- A spin incoherent Luttinger liquid (SILL) is an example of carriers with the “dual nature” the authors refer to. The carriers at the Fermi points are both itinerant and active in transport while also carrying $\sim \log(2)$ entropy due to their spin. It would be interesting for the authors to compare and contrast their observations of dual nature with that of the SILL. Could similar highly-dressed quasiparticles, which are both itinerant and carry the entropy, be active in the TBG case?

We thank the referee for this insightful comment. Indeed, the spin-incoherent Luttinger liquid (SILL) provides a classic example of electronic degrees of freedom exhibiting a “dual nature”: charge carriers remain itinerant and contribute to transport, while their spin sector is highly fluctuating and carries a large entropy $\sim \log(2)$ per particle.

Analogous phenomenology has also been discussed in higher dimensions - most notably in the context of liquid ^3He , which has been argued to form a “nearly localized Fermi liquid” (D. Vollhardt, *Rev. Mod. Phys.* **56**, 99, 1984). In such cases, strong short-range correlations produce “crystalline-like” local physics despite an overall fluid metallic state. However, unlike in the 1D SILL, the theoretical understanding in $d > 1$ is less controlled.

For MATBG, an additional element is present: the moiré lattice and its associated narrow-band structure. The large gap between the narrow and remote bands indicates that the lattice plays a crucial role. Our findings of nearly flat bands throughout most

of the Brillouin zone seem broadly consistent with the “topological heavy-fermion” perspective, in which much of the narrow band Hilbert space can be represented in terms of spatially localized orbitals, which coexist with more itinerant, Dirac-like carriers near the Γ point.

Whether this scenario can be continuously connected to a higher-dimensional spin-incoherent-like regime - where the same electrons simultaneously carry current and large spin entropy - is an interesting open question.

Referee #4

I co-reviewed this manuscript with one of the reviewers who provided the listed reports.

The authors have responded thoroughly to our questions. We recommend for publication, but would like the authors to address the comments below for further clarity.

1. While the authors present a convincing evidence that the 15meV feature is unlikely to be due to strain, it is still puzzling whether it is a single particle effect or not. In particular, a very similar 15meV splitting was observed in [Wong et al Nature volume 582, pages 198–202 (2020)]. In this case, the 15meV splitting continuously connects to a corresponding splitting in the remote bands. Do the authors believe that the splitting they observe is of a different nature, as it is not present in the remote bands? Relatedly, could the authors comment on the very large lifetime of flat band electrons while the chemical potential is in the remote bands? I would have thought that the system is mostly non-interacting once the chemical potential is in the remote bands, such that the flat bands should have a relatively sharp quasiparticle peak.

Regarding the 15mV peak – from the existing the data we cannot answer this definitively. In the previous review round we have provided all the experimental data and explanations that we have regarding this question. We can add only one more comment in reply to the question – if the excitation was purely of single particle origin then one would expect it to have in Fig. 4a that same slope as a function of filling factor as the band that is crossing the Fermi energy. However we see that the excitation feature appears at an energy that is filling factor independent. This is hard to explain with single particle physics alone.

Experimentally, when the flat bands lie below the Fermi energy ($\nu > 4$), we do not resolve two distinct maxima; rather, we observe a single, broader peak. A single peak is seen across all momenta. This behavior is robust across many measurements taken at different locations and is reproduced in two independent samples.

This is what the experiments demonstrate clearly. Indeed, it is not expected from simple reasoning, but this is a solid experimental observation, which future theories will have to explain.

2. A widely anticipated application of the QTM to TBG is a better understanding of the superconducting normal state. Transport experiments typically see quantum oscillations emanating from $\nu=2$ and $\nu=-2$, pointed away from charge neutrality. These have been argued to suggest that superconductivity emerges from a small Fermi surface of light carriers. Fig. 5 however does not seem consistent with a Fermi surface of c-like electrons. Instead there is a small spectral weight that is spread out across the heavy/f-like parts of the Brillouin Zone. Notably, there does not seem to be any sign of a Fermi surface peak. It would be good to comment on this tension and its implications for how we should understand the superconducting normal state.

Indeed, this is the experimental observation: there is no signature of a fermi surface between -2 and -3 or between 2 and 3. It is important, however, to keep in mind the experimental conditions. Our measurements are performed at $T = 4K$ and zero magnetic field. By contrast, quantum oscillations are necessarily measured at finite magnetic fields, and it is therefore difficult to exclude the possibility that the oscillatory features observed at finite field reflect a field-stabilized ground state that differs from the zero-field state probed here. In this sense, the apparent tension may partly reflect the different regimes accessed by the two experimental approaches.

For completeness, we included the answer we provided here in the revised manuscript,
Methods section 15